# Combination-of-Experts with Knowledge Sharing for Cross-Task Vehicle Routing Problems

**Zikang Yu**[1] , **Jinbiao Chen**[2] , **Jiahai Wang**[1]*
[1]School of Computer Science and Engineering, Sun Yat-sen University, P.R. China
[2]Department of Industrial Systems Engineering and Management, National University of Singapore
`yuzk6@mail2.sysu.edu.cn, bill.cjb@nus.edu.sg,`
`wangjiah@mail.sysu.edu.cn`

## Abstract

Recent neural methods have shown promise in generalizing across various vehicle routing problems (VRPs). These methods adopt either a fully-shared dense model across all VRP tasks (i.e., variants) or a mixture-of-experts model that assigns node embeddings within each task instance to different experts. However, they both struggle to generalize from training tasks with basic constraints to out-of-distribution (OOD) tasks involving unseen constraint combinations and new basic constraints, as they overlook the fact that each VRP task is defined by a combination of multiple basic constraints. To address this, this paper proposes a novel model, combination-of-experts with knowledge sharing (CoEKS), which leverages the structural characteristic of VRP tasks. CoEKS enhances generalization to constraint combinations via two complementary components: a combination-of-experts architecture enabling flexible combinations via prior assignment of constraint-specific experts, and a knowledge sharing strategy strengthening generalization via automatic learning of transferable general knowledge across constraints. Moreover, CoEKS allows new experts to be plugged into the trained model for rapid adaptation to new constraints. Experiments demonstrate that Co-EKS outperforms state-of-the-art methods on in-distribution tasks and delivers greater gains on OOD tasks, including unseen constraint combinations (relative improvement of 12% over SOTA) and new constraints (25% improvement).

## 1 Introduction

Combinatorial Optimization (CO) plays a pivotal role in numerous real-world applications, such as logistics (Zong et al., 2022), transportation (Fu et al., 2025), supply chain management (Tirkolaee et al., 2020), and resource allocation (Heydaribeni et al., 2024). The vehicle routing problem (VRP) stands as one of the most fundamental yet challenging CO problems, requiring the determination of optimal routes for a vehicle fleet serving a set of customers while satisfying multiple operational constraints. Despite decades of algorithmic progress, traditional methods face significant limitations: exact approaches are computationally infeasible for large-scale instances due to the NP-hard nature of VRP (Wu et al., 2024), while heuristic approaches heavily rely on handcrafted expert knowledge and time-consuming iterative search from scratch for each new instance (Bogyrbayeva et al., 2024).

Recent advances in deep learning have introduced neural methods for VRPs, which autonomously learn heuristic policies from massive data end-to-end. These methods not only circumvent the dependency on expert domain knowledge but also produce high-quality solutions within short solving time (Chen et al., 2025a; Li et al., 2025b; Goh et al., 2024; Chen et al., 2023b; Zhang et al., 2023; Liao et al., 2025; Fang et al., 2026; Xiao et al., 2024). However, despite the promising performance, most existing methods adopt a task-specific learning paradigm, necessitating a separate neural model for each VRP task. This lack of cross-task generalization capability incurs costly retraining and deployment overhead when adapting to new tasks.

---

*Corresponding author.

More recently, some efforts have focused on developing unified models for cross-task VRPs. Despite demonstrating feasibility, current methods still perform suboptimally, especially under out-of-distribution (OOD) generalization scenarios: 1) tasks with unseen combinations of basic constraints, and 2) tasks involving new basic constraints. According to their model architectures, they fall into two categories: a *task-shared dense model* and a *node-level mixture-of-experts (MoE) model*. The task-shared dense model (Liu et al., 2024b; Berto et al., 2024; Li et al., 2025a), whose parameters are fully-shared across all tasks, overemphasizes coupled representations but neglects task-specific ones, resulting in negative transfer among tasks. This leads to particularly poor OOD generalization. As an alternative, the node-level MoE model (Zhou et al., 2024a; Huang et al., 2025) employs a gating mechanism to assign each node embedding within a task instance to different experts, fostering node-specialized experts. However, this gating mechanism restricts expert vision to a narrow node subset, which weakens experts' cognition of task-level knowledge.

Unlike general multi-task learning, we observe that VRP tasks often involve combinations of multiple basic constraints. This motivates us to develop an innovative model architecture, called combination-of-experts with knowledge sharing (CoEKS). On the one hand, CoEKS utilizes constraint-specific experts to facilitate learning dedicated knowledge for every basic constraint. This enables flexible combinations of experts to manage diverse VRP tasks with constraint combinations and allows further plugging in of new experts to adapt even to unseen basic constraints. On the other hand, CoEKS transfers general knowledge across constraints to foster collaboration among experts and enhance OOD generalization.

The contributions of this paper can be summarized as follows: 1) We introduce a novel CoEKS model, building on the recognition of the prior structural characteristics of VRPs. This model is designed to enhance OOD generalization for VRPs with constraint combinations via two complementary components and adapts to unseen constraints by plugging in new experts. 2) We design a combination-of-experts (CoE) architecture to acquire specialized constraint-level knowledge, in which each expert specializes in a basic constraint, enabling the model to effectively solve diverse VRPs by flexibly combining corresponding experts. 3) We propose a multi-view knowledge sharing strategy to transfer general knowledge across constraints, utilizing mutual distillation and shared transformation layers to automatically learn coordination among experts. 4) We demonstrate that CoEKS can be deployed on both state-of-the-art (SOTA) and classic backbones to show its universality. Extensive experiments indicate that CoEKS outperforms SOTA cross-task neural methods for in-distribution (ID) and particularly OOD generalization.

## 2 RELATED WORKS

**Neural combinatorial optimization for VRPs.** Existing neural methods for VRPs can be primarily categorized into two distinct groups: 1) *Neural construction methods* leverage deep neural networks to generate feasible solutions in an end-to-end manner. Some works (Vinyals et al., 2015; Bello et al., 2017; Nazari et al., 2018) pioneer this direction to address the Traveling Salesman Problem (TSP) and VRP. Attention Model (Kool et al., 2019), which combines Transformer with reinforcement learning, is regarded as a milestone. Policy Optimization with Multiple Optima (POMO) (Kwon et al., 2020) leverages solution symmetry to improve policy learning and has become a typical backbone model for many subsequent extensions (Bi et al., 2024; Chen et al., 2025b; 2023a; Zhou et al., 2024b; Hou et al., 2023; Fang et al., 2024) due to its prominent performance and flexibility. Recently, RELD (Huang et al., 2025) has emerged as the SOTA backbone model by incorporating a feedforward neural network (FFN) and identity mapping (IDT) into the decoder. 2) *Neural improvement methods* employ neural networks to replace handcrafted rules, iteratively refining an initial solution to meet specified requirements (Wu et al., 2021; Ma et al., 2021; Luo et al., 2025; Ma et al., 2023). Although such methods often yield superior solutions, their computational overhead is considerable. Consequently, this paper focuses on neural construction methods.

Some subsequent works enhance the generalization of neural construction methods across problem sizes (Luo et al., 2023; Pan et al., 2025), node distributions (Bi et al., 2022; Liu et al., 2024a) and their interactions (Manchanda et al., 2022; Wang et al., 2024). Our work targets a more challenging and underexplored scenario: generalization across diverse VRP tasks.

**Cross-task generalization for VRPs.** Recent studies (Drakulic et al., 2024; Wang & Yu, 2023; Jiang et al., 2024b) investigate a unified model for cross-task CO problems, but focus only on ID

Figure 1: Illustrations of feasible solutions with various constraints.

generalization. In addition, Lin et al. (2024) develop fine-tuning adapters on a pre-trained TSP model for new tasks. These methods fall short of zero-shot OOD generalization across diverse VRPs.

Existing approaches to zero-shot OOD generalization for diverse VRPs with constraint combinations can be divided into two categories. One type, such as POMO-MTL (Liu et al., 2024b), RouteFinder (Berto et al., 2024), and CaDA (Li et al., 2025a), adopts a *task-shared dense model*, which overemphasizes coupled representations at the expense of task-specific ones, leading to negative transfer among tasks. The other type employs a *node-level MoE model* to route each node embedding to different experts, like MVMoE (Zhou et al., 2024a). ReLD-MoEL (Huang et al., 2025) integrates MVMoE with the ReLD backbone, achieving the SOTA performance on OOD generalization across VRP tasks. Nevertheless, these node-level MoE models suffer from narrow expert vision limited to a node subset, which deteriorates task-level generalization (Further comparisons with the MoE models are provided in Appendix E.1). In summary, existing models overlook the structural characteristic of VRPs, which motivates us to propose a novel architecture, CoEKS. Along another orthogonal lines of research, Liu et al. (2025) propose a pre-training paradigm for VRPs to improve generalization and Goh et al. (2025) study the multi-task multi-distribution VRPs using a mixture-of-depths architecture with clustering.

# 3 PRELIMINARIES

## 3.1 VEHICLE ROUTING PROBLEMS

Formally, VRP is modeled on a complete graph $G = (V, E)$, where $V = V_0 \cup V_c$ includes a depot node $V_0$ and customer nodes $V_c = \{v_1, \ldots, v_n\}$. Each node $v_i \in V$ is associated with 2D coordinate $x_i \in [0, 1]^2$. Edge set $E$ connects all node pairs, with each edge $(i, j) \in E$ associated with cost $c_{ij}$, measured by the Euclidean distance. Each customer node $v_i \in V_c$ has a non-negative demand $d_i \geq 0$, while the depot has zero demand. A VRP solution $\tau$ consists of a set of routes, each executed by a vehicle, that collectively visit all customers exactly once while satisfying task-specific constraints. The objective is to minimize total cost: $\min_{\tau \in \Phi} c(\tau)$, where $c(\tau) = \sum_{r \in \tau} \sum_{(i,j) \in r} c_{ij}$, $\Phi$ denotes the set of feasible solutions, and $r$ represents a route assigned to a vehicle.

VRP tasks are defined by applying different sets of practical constraints (see Figure 1) to reflect diverse real-world operational requirements. This paper focuses on six basic constraints from recent studies (Berto et al., 2024; Zhou et al., 2024a). **1) Capacity (C):** Each vehicle has a maximum capacity $Q$, i.e. the total demand of customers along any route must not exceed $Q$. **2) Open Route (O):** Vehicles are allowed to end their routes at the last customer instead of returning to the depot. **3) Backhaul (B):** Customers are divided into linehaul nodes that require goods from the depot (delivery demand $d_i$) and backhaul nodes that need goods to return to the depot (pickup demand $p_i$). Vehicles serve both on a single route, but all linehaul deliveries must precede backhaul pickups. **4) Duration Limit (L):** Each route is subject to a maximum duration or distance limit $L$, ensuring balanced workloads and operational feasibility. **5) Time Window (TW):** Each customer $n_i$ has a time window $[e_i, l_i]$ and service duration $s_i$. Service must start within $[e_i, l_i]$. Early arrivals must wait, and service is not allowed after $l_i$. Additionally, all vehicles must return to the depot before a global time limit $T_{\max}$. **6) Mixed Backhauls (MB):** Unlike backhaul, MB relaxes the strict linehaul-before-backhaul priority, allowing flexible sequences.

Each basic constraint can exist individually or in combination, resulting in a rich set of VRP tasks. Their interplay introduces significant task diversity, requiring flexible and generalizable methods. More details on the VRP configurations and data generation process are provided in Appendix A.

## 3.2 NEURAL CONSTRUCTION METHODS

Neural construction methods represent a cutting-edge approach to solving VRPs, using deep reinforcement learning to construct solutions in an autoregressive manner, eliminating the need for precomputed labels or handcrafted heuristics. Typically, such methods adopt a deep neural network $\theta$ with an encoder-decoder architecture to parameterize a stochastic policy. The encoder processes static VRP features (e.g., node coordinates and demands) to produce node embeddings. At each step, the decoder integrates these embeddings with dynamic context (e.g., remaining vehicle capacity, current route length) to output a probability distribution over unvisited nodes, from which the next node is sampled. This process repeats until all customers have been visited, forming a complete solution. The solution construction is modeled as a Markov Decision Process, where state consists of the instance and current partial solution, and the action comprises the set of selectable nodes. Given a graph $G$, the policy network $\theta$ specifies the probability of a solution $\tau$, expressed autoregressively as $p_\theta(\tau|G) = \prod_{t=1}^{T} p_\theta(a_t|s_t)$, where $a_t$ and $s_t$ are the action and state at step $t$, respectively. $T$ is the total number of decoding steps. The reward is defined as the negative cost of tour $\tau$, i.e., $r(\tau) = -c(\tau)$. The task loss $\mathcal{L}_p$ is defined as the expected total cost. The policy is optimized via REINFORCE with a shared baseline $b(G)$, defined as the average reward over multiple sampled trajectories per instance (Kwon et al., 2020). The policy gradient is estimated as:

$$\nabla_\theta \mathcal{L}_p(\theta|G) = \mathbb{E}_{p_\theta(\tau|G)} \left[ (r(\tau) - b(G)) \nabla_\theta \log p_\theta(\tau|G) \right]. \tag{1}$$

## 4 METHODOLOGY

This section presents the proposed combination-of-experts with knowledge sharing (CoEKS), which is tailored for cross-task generalization for VRPs with basic constraints and their combinations. The overall model structure is shown in Figure 2, where CoEKS is employed in the encoder (see Appendix B for details). CoEKS addresses this challenge through two complementary components: 1) a combination-of-experts (CoE) model that learns specialized knowledge and enables adaptive combinations of constraint-specific experts to handle diverse VRP tasks; and 2) a multi-view knowledge sharing strategy that enhances the model's learning of transferable general knowledge across different constraints, thereby improving cross-task generalization for VRPs.

### 4.1 COE MODEL

The CoE model extends the transformer-based architecture by introducing expert and combiner modules in the encoder. Each expert specializes in a basic constraint and adaptively aggregate their expertise through combiners, enabling efficient handling of VRPs with combinations of constraints.

**Constraint-specific expert.** In a standard transformer block, the FFN processes node embeddings to capture complex relationships. CoEKS replace the FFN into a pool of constraint-specific experts (i.e., FFNs), where each expert (for $j \in \mathcal{E} = \{C, O, B, L, TW\}$) specializes in a specific constraint: capacity ($E_C$), open route ($E_O$), backhaul ($E_B$), duration limit ($E_L$), and time window ($E_{TW}$). For a given VRP instance with constraint set $CS \subseteq \mathcal{E}$, only the corresponding experts are activated, with outputs defined as:

$$O_j^E(h) = \left\{ \begin{array}{ll} E_j(h), & \text{if } j \in CS, \\ 0, & \text{otherwise,} \end{array} \right. \tag{2}$$

where $E_j(h) = \text{FFN}_j(h) \in \mathbb{R}^d$ is the output of the $j$-th expert, and $h \in \mathbb{R}^d$ is the input embedding. Since the capacity $C$ is a fundamental constraint underlying all VRP tasks, a shared expert mechanism (Dai et al., 2024) is employed, where the expert $E_C$ is always activated as a shared expert corresponding to CVRP. This design ensures universal problem-solving capability while facilitating expert specialization.

**Combiner.** To adaptively the combine outputs of activated experts, we introduce a combiner for each expert $E_j$, parameterized by $W_j \in \mathbb{R}^{1 \times d}$. The raw values of activated combiners are $s_j(h) = W_j \cdot h$, $j \in CS$, and then weights corresponding to the experts are calculated by softmax normalization:

$$S_j(h) = \frac{\exp(s_j(h))}{\sum_{k \in CS} \exp(s_k(h))}, \quad j \in CS, \tag{3}$$

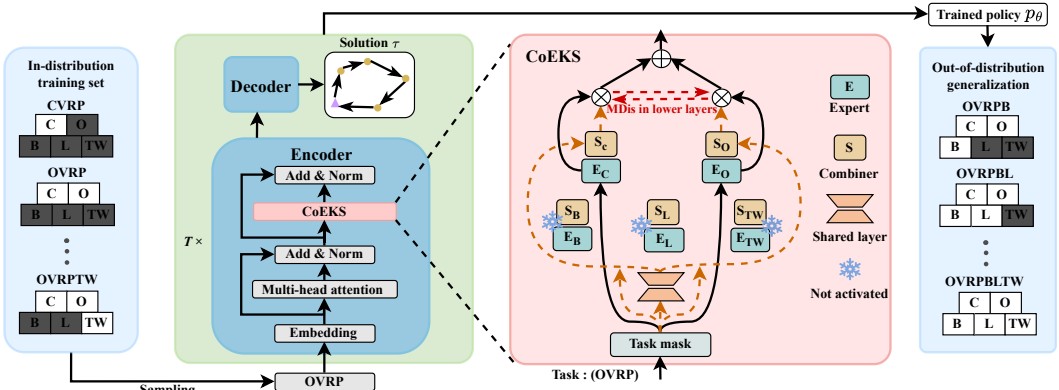

Figure 2: Workflow of the cross-task VRP method with CoEKS: Sampling an OVRP instance from the training set (gray parts indicate inactive constraints), the encoder generates node embeddings, and the decoder constructs a feasible solution. CoEKS output is determined by the activated experts and combiners, with mutual distillation (MDis) among the activated experts. The trained policy then generalizes to OOD tasks. Add represents residual connections and Norm denotes normalization.

with $S_k(h) = 0$ for inactive experts ($k \notin CS$). Therefore, the final output of CoE can be obtained by a weighted combination of active experts:

$$O(h) = \sum_{j \in CS} E_j(h) \cdot S_j(h). \tag{4}$$

## 4.2 MULTI-VIEW KNOWLEDGE SHARING STRATEGY

The CoE model effectively specializes experts in distinct constraints, serving as a foundation for handling diverse VRP tasks. To complement this, a multi-view knowledge sharing strategy is proposed to enhance the model's learning of transferable knowledge across different constraints, thereby improving OOD generalization. This strategy operates in two views: expert-view and combiner-view.

**Expert-view knowledge sharing.** To broaden the expert vision and strengthen their comprehensive understanding of VRPs, we introduce mutual distillation (MDis), where active experts exchange knowledge to capture shared patterns across constraints. Unlike traditional knowledge distillation, which transfers knowledge from a teacher to a student model, MDis encourages peer-to-peer learning among experts (Xie et al., 2024). An auxiliary loss $\mathcal{L}_{md}$ is incorporated to facilitate this process. The overall loss function is defined as follows, where $\mathcal{L}_p$ denotes the primary loss of the task, and $\alpha$ controls the distillation strength.

$$\mathcal{L} = \mathcal{L}_p + \alpha \cdot \mathcal{L}_{md}, \tag{5}$$

$\mathcal{L}_{md}$ is calculated as:

$$\mathcal{L}_{md} = \begin{cases} 0, & K = 1, \\ \text{MSE}(E_1(h), E_2(h)), & K = 2, \\ \frac{1}{K} \sum_{i=1}^{K} \text{MSE}(E_i(h), E_{\text{avg}}(h)), & K > 2, \end{cases} \tag{6}$$

where $K$ is the number of active experts, $E_i(h) \in \mathbb{R}^d$ is the output of the $i$-th expert, $E_{\text{avg}}(h) = \frac{1}{K} \sum_{i=1}^{K} E_i(h)$ is a virtual expert averaging active expert outputs, and MSE is the mean squared error. For $K = 1$ (i.e., CVRP with only $E_C$), $\mathcal{L}_{md} = 0$. The virtual expert simplifies computation for $K > 2$ by reducing the complexity from $O(K^2)$ (pairwise comparisons) to $O(K)$, while still guiding experts toward a consensus by minimizing the variance of their outputs.

To realize the trade-off between specialized and general knowledge, MDis is employed in the lower encoder layer (e.g., the first layer). This design choice is inspired by the property that lower layers in neural networks tend to capture general features, while higher layers focus on task-specific knowledge (Long et al., 2017). By localizing knowledge sharing to these early representations, our

method promotes the exchange of broadly useful information among experts without causing a homogenization of their expertise. The similarity of expert representations in the lower encoder layers is empirically validated via the t-SNE analysis in Appendix E.2.

**Combiner-view knowledge sharing.** To further enhance generalization to unseen constraint combinations, we introduce a combiner-view knowledge sharing mechanism. Specifically, a shared transformation layer $f_s$ is applied to the input embedding $h$ before it reaches the combiners introduced in Section 4.1. Therefore, $f_s$ can inject cross-task knowledge for all combiners, enabling them to make informed weighting decisions across diverse VRP tasks. Given the importance of nonlinearity in modeling complex functions, we introduce nonlinearity to improve the representation of $f_s$. For simplicity, $f_s$ is implemented as a low-rank multilayer perceptron (MLP) with a residual connection:

$$f_s(h) = W_2 \cdot \text{ReLU}(W_1 \cdot h) + h, \tag{7}$$

where $W_1 \in \mathbb{R}^{d \times r}$ and $W_2 \in \mathbb{R}^{r \times d}$ are weight matrices forming a bottleneck structure with $r \ll d$, enhancing parameter efficiency while preserving expressiveness. The final output of CoEKS is:

$$O_{\text{CoEKS}}(h) = \sum_{j \in CS} E_j(h) \cdot S_j(f_s(h)). \tag{8}$$

### 4.3 INFERENCE FOR CONSTRAINT COMBINATIONS AND ADAPTATION TO NEW CONSTRAINTS

During inference, CoEKS addresses diverse VRPs by activating experts corresponding to specific constraint combinations, as illustrated in Figure 2. It achieves zero-shot OOD generalization for unseen combinations by flexibly combining constraint-specific experts. When encountering unseen basic constraints, new experts are plugged into the trained model and fine-tuned in isolation, with all existing parameters frozen to prevent catastrophic forgetting. This design maintains acquired knowledge and enables continuous rapid adaptation to new constraints, facilitating scalable deployment.

## 5 EXPERIMENTS

In this section, extensive experiments are conducted on 48 VRP tasks. All experiments are carried out on an NVIDIA RTX 3090 GPU and an AMD Ryzen 5 3600. Our code and data are publicly available at `https://github.com/yuzikang0/CoEKS`.

We aim to answer the following research questions: **Q1.** Does CoEKS achieve superior ID and OOD generalization for tasks with unseen constraint combinations? **Q2.** Can CoEKS show superior scalability in OOD tasks with new constraints? **Q3.** Is universal CoEKS consistently effective across different backbones? **Q4.** How effective is the knowledge sharing strategy in CoEKS?

**Baselines.** 1) *Traditional methods.* Two heuristic solvers are employed in this study: the state-of-the-art PyVRP (Wouda et al., 2024) and Google OR-Tools (Furnon & Perron, 2023). Both methods use a single CPU core to solve each instance. For node sizes $n = 50$ and $n = 100$, the time limits are 10 and 20 seconds. 2) *Neural methods.* Recent representative cross-task VRP methods are considered, including POMO-MTL (Liu et al., 2024b), RF-TE (Berto et al., 2024), MVMoE (Zhou et al., 2024a), CaDA (Li et al., 2025a), and ReLD-MoEL (Huang et al., 2025). RF-TE and ReLD-MoEL are the strongest variants reported in RouteFinder (Berto et al., 2024) and ReLD (Huang et al., 2025). CoEKS is implemented on the SOTA ReLD backbone (see Appendix B for more details).

**Training.** Our settings mostly follow RouteFinder (Berto et al., 2024). Each model is trained for 300 epochs, with each epoch containing 100K VRP instances. The Adam optimizer is used with a learning rate of $3 \times 10^{-4}$ and batch sizes are set to 256 and 128 for $n = 50$ and $n = 100$, respectively. The learning rate is multiplied by 0.1 at epochs 270 and 295. Our training task set is similar to MVMoE, including CVRP, OVRP, VRPB, VRPL, VRPTW, OVRPTW, and OVRPL (see Appendix C.1 for further discussion). CoEKS adopts the mixed batch training and reward regularization scheme from RouteFinder, with the distillation strength $\alpha$ set to 0.01. MDis is employed in the first encoder layer. For all neural methods, the rest of the settings follow their original papers.

**Inference & Metrics.** For all neural methods, a greedy rollout with $\times 8$ instance augmentation (Zhou et al., 2024a) is employed. The test set is obtained through random sampling, with 1000 instances per VRP task to reduce the impact of randomness. We show the average results of the tests, including the objective value (total cost), the gap to the best traditional solver, and the total test time.

Table 1: Performance on 1K test instances of ID VRP tasks.

| | Method | $n=50$ Obj. | Gap | Time | $n=100$ Obj. | Gap | Time | | Method | $n=50$ Obj. | Gap | Time | $n=100$ Obj. | Gap | Time |
|---|---|---|---|---|---|---|---|---|---|---|---|---|---|---|---|
| **CVRP** | HGS-PyVRP# | 10.372 | * | 10.4m | 15.628 | * | 20.8m | **VRPTW** | HGS-PyVRP# | 16.031 | * | 10.4m | 25.423 | * | 20.8m |
| | OR-Tools# | 10.572 | 1.907% | 10.4m | 16.280 | 4.178% | 20.8m | | OR-Tools# | 16.089 | 0.347% | 10.4m | 25.814 | 1.506% | 20.8m |
| | POMO-MTL | 10.502 | 1.257% | 1s | 15.875 | 1.617% | 7s | | POMO-MTL | 16.428 | 2.471% | 1s | 26.487 | 4.173% | 7s |
| | MVMoE | 10.482 | 1.059% | 2s | 15.841 | 1.399% | 9s | | MVMoE | 16.439 | 2.550% | 2s | 26.472 | 4.113% | 9s |
| | RF-TE | 10.497 | 1.213% | 1s | 15.829 | 1.327% | 6s | | RF-TE | 16.390 | 2.237% | 1s | 26.283 | 3.363% | 7s |
| | CaDA | 10.491 | 1.148% | 3s | 15.822 | 1.277% | 11s | | CaDA | **16.297** | **1.651%** | 2s | **26.119** | **2.721%** | 12s |
| | ReLD-MoEL | 10.467 | 0.920% | 2s | 15.797 | 1.116% | 9s | | ReLD-MoEL | 16.414 | 2.386% | 2s | 26.388 | 3.782% | 9s |
| | CoEKS | **10.464** | **0.891%** | 2s | **15.787** | **1.057%** | 9s | | CoEKS | 16.361 | 2.050% | 2s | 26.300 | 3.433% | 9s |
| **OVRP** | HGS-PyVRP# | 6.507 | * | 10.4m | 9.725 | * | 20.8m | **VRPL** | HGS-PyVRP# | 10.587 | * | 10.4m | 15.766 | * | 20.8m |
| | OR-Tools# | 6.553 | 0.686% | 10.4m | 9.995 | 2.732% | 20.8m | | OR-Tools# | 10.570 | 2.343% | 10.4m | 16.466 | 5.302% | 20.8m |
| | POMO-MTL | 6.706 | 3.025% | 1s | 10.173 | 4.592% | 6s | | POMO-MTL | 10.756 | 1.550% | 1s | 16.090 | 2.059% | 7s |
| | MVMoE | 6.685 | 2.697% | 2s | 10.138 | 4.226% | 8s | | MVMoE | 10.736 | 1.362% | 2s | 16.053 | 1.822% | 9s |
| | RF-TE | 6.678 | 2.595% | 1s | 10.097 | 3.813% | 6s | | RF-TE | 10.742 | 1.434% | 1s | 16.017 | 1.606% | 6s |
| | CaDA | 6.683 | 2.668% | 2s | 10.105 | 3.882% | 11s | | CaDA | 10.729 | 1.317% | 2s | 16.014 | 1.584% | 11s |
| | ReLD-MoEL | 6.661 | 2.343% | 2s | 10.073 | 3.559% | 9s | | ReLD-MoEL | 10.713 | 1.153% | 2s | 15.998 | 1.482% | 9s |
| | CoEKS | **6.648** | **2.138%** | 2s | **10.046** | **3.290%** | 8s | | CoEKS | **10.712** | **1.152%** | 2s | **15.997** | **1.476%** | 9s |
| **VRPB** | HGS-PyVRP# | 9.687 | * | 10.4m | 14.377 | * | 20.8m | **OVRPTW** | HGS-PyVRP# | 10.510 | * | 10.4m | 16.926 | * | 20.8m |
| | OR-Tools# | 9.802 | 1.159% | 10.4m | 14.933 | 3.853% | 20.8m | | OR-Tools# | 10.519 | 0.078% | 10.4m | 17.027 | 0.583% | 20.8m |
| | POMO-MTL | 9.995 | 3.177% | 1s | 14.989 | 4.279% | 7s | | POMO-MTL | 10.691 | 1.700% | 2s | 17.500 | 3.367% | 7s |
| | MVMoE | 9.966 | 2.867% | 2s | 14.952 | 4.025% | 9s | | MVMoE | 10.696 | 1.747% | 2s | 17.485 | 3.278% | 10s |
| | RF-TE | 9.984 | 3.050% | 1s | 14.926 | 3.838% | 6s | | RF-TE | 10.675 | 1.542% | 1s | 17.363 | 2.555% | 7s |
| | CaDA | 9.965 | 2.860% | 2s | 14.906 | 3.699% | 11s | | CaDA | **10.626** | **1.084%** | 3s | **17.267** | **1.990%** | 13s |
| | ReLD-MoEL | 9.936 | 2.557% | 2s | 14.877 | 3.496% | 9s | | ReLD-MoEL | 10.682 | 1.613% | 2s | 17.429 | 2.945% | 10s |
| | CoEKS | **9.930** | **2.497%** | 2s | **14.854** | **3.338%** | 9s | | CoEKS | 10.659 | 1.393% | 2s | 17.376 | 2.633% | 10s |
| **OVRPL** | HGS-PyVRP# | 6.507 | * | 10.4m | 9.724 | * | 20.8m | **(ID Avg.)** | HGS-PyVRP# | 10.029 | * | 10.4m | 15.367 | * | 20.8m |
| | OR-Tools# | 6.552 | 0.668% | 10.4m | 10.001 | 2.791% | 20.8m | | OR-Tools# | 10.094 | 1.574% | 10.4m | 15.788 | 2.992% | 20.8m |
| | POMO-MTL | 6.709 | 3.070% | 1s | 10.177 | 4.625% | 6s | | POMO-MTL | 10.255 | 2.321% | 1s | 15.899 | 3.530% | 7s |
| | MVMoE | 6.687 | 2.737% | 2s | 10.140 | 4.244% | 6s | | MVMoE | 10.242 | 2.146% | 2s | 15.869 | 3.301% | 9s |
| | RF-TE | 6.678 | 2.606% | 1s | 10.096 | 3.803% | 6s | | RF-TE | 10.235 | 2.097% | 1s | 15.802 | 2.901% | 6s |
| | CaDA | 6.684 | 2.685% | 2s | 10.106 | 3.900% | 11s | | CaDA | 10.211 | 1.916% | 2s | 15.763 | 2.722% | 11s |
| | ReLD-MoEL | 6.661 | 2.341% | 2s | 10.075 | 3.583% | 9s | | ReLD-MoEL | 10.219 | 1.902% | 2s | 15.805 | 2.852% | 9s |
| | CoEKS | **6.648** | **2.135%** | 2s | **10.047** | **3.298%** | 9s | | CoEKS | **10.203** | **1.751%** | 2s | **15.773** | **2.646%** | 9s |

(ID Avg.): Average performance across ID VRP tasks. **bold**: Best results among learning-based methods. #: Results are adopted from Berto et al. (2024) for the convenience of comparison. *: Best traditional method, taken as the baseline for gap calculation.

**(Q1) Generalization for ID and OOD VRPs.** Table 1 presents the results on ID tasks. Across different problem scales, CoEKS outperforms all neural methods on ID average gap (ID Avg.) and achieves the smallest gaps in 10 out of 14 cases. To evaluate their zero-shot OOD generalization performance, all methods are examined on 9 VRP tasks with unseen constraint combinations. As shown in Table 2, CoEKS consistently achieves the best performance, demonstrating its ability to effectively handle unseen constraint combinations by adaptively combining experts. For $n=50$ and $n=100$, CoEKS outperforms all other neural methods in the overall OOD average gap (OOD Avg.), with relative improvements of at least 18.3% and 13%, respectively. Moreover, compared with the traditional solvers, CoEKS achieves competitive results with substantially lower solving time, offering notable efficiency for practical applications. We further conduct supplementary experiments on the OOD large-scale real-world instances of the CVRPLIB benchmark dataset ($n > 500$), where CoEKS consistently delivers superior generalization results (see Appendix E.4).

**(Q2) Scalability for adaptation to unseen constraints by plugging in new experts.** An unseen constraint MB is considered in VRP tasks, following the challenging setting in RouteFinder. Co-EKS plugs a new MB-specific expert into the trained model and only fine-tunes this expert. Lin et al. (2024) introduce task-specific adapter layers (AL) for fine-tuning. RouteFinder proposes efficient adaptation layers (EAL) (Berto et al., 2024), which extend the weight matrix with zero-padding to support new constraints. We compare: 1) AL-based methods, including RF-TE-AL and ReLD-MoEL-AL; 2) EAL-based methods, including RF-TE-EAL and ReLD-MoEL-EAL; 3) two variants of our method. CoEKS$^{+}$, where the new expert and combiner are randomly initialized; and CoEKS$^{c+}$, which reuses shared modules ($E_c$ and $S_c$) for initialization to accelerate learning, inspired by Jiang et al. (2024a). For simplicity, our variants both use EAL to adapt to new constraint attributes. Aligning with RouteFinder, fine-tuning is conducted for 10 epochs with 10K instances per epoch on tasks including VRPMB and VRPMBTW (see Appendix C.2 for more discussions).

As shown in Table 3, our methods consistently outperforms all baselines, where CoEKS$^{c+}$ exploits the knowledge of shared modules in CoE and performs best. This demonstrates that CoEKS has superior scalability in adapting to a new constraint by flexibly plugging in a new expert and combiner. This advantage is further amplified in OOD tasks with more constraints, where the extended model capability allows CoEKS to better handle complex generalization challenges (see Appendix E.3 for further validation on scalability to new Multi-Depot (MD) constraint).

Table 2: Performance on 1K test instances of OOD VRP tasks.

| | Method | $n=50$ | | | $n=100$ | | | | Method | $n=50$ | | | $n=100$ | | |
|---|---|---|---|---|---|---|---|---|---|---|---|---|---|---|---|
| | | Obj. | Gap | Time | Obj. | Gap | Time | | | Obj. | Gap | Time | Obj. | Gap | Time |
| OVRPB | HGS-PyVRP# | 6.898 | * | 10.4m | 10.335 | * | 20.8m | VRPBL | HGS-PyVRP# | 10.186 | * | 10.4m | 14.779 | * | 20.8m |
| | OR-Tools# | 6.928 | 0.412% | 10.4m | 10.577 | 2.315% | 20.8m | | OR-Tools# | 10.331 | 1.390% | 10.4m | 15.426 | 4.338% | 20.8m |
| | POMO-MTL | 7.447 | 7.886% | 1s | 12.091 | 16.926% | 7s | | POMO-MTL | 10.743 | 5.284% | 1s | 15.875 | 7.246% | 7s |
| | MVMoE | 7.371 | 6.797% | 2s | 11.720 | 13.305% | 9s | | MVMoE | 10.709 | 4.936% | 2s | 15.792 | 6.699% | 9s |
| | RF-TE | 7.378 | 6.900% | 1s | 11.840 | 14.520% | 7s | | RF-TE | 10.777 | 5.685% | 1s | 17.011 | 15.149% | 6s |
| | CaDA | 7.701 | 11.549% | 2s | 11.796 | 14.074% | 12s | | CaDA | 10.794 | 5.705% | 2s | 15.883 | 7.399% | 12s |
| | ReLD-MoEL | 7.335 | 6.272% | 2s | 11.446 | 10.691% | 10s | | ReLD-MoEL | **10.621** | **4.130%** | 2s | 15.639 | 5.694% | 9s |
| | CoEKS | **7.241** | **4.913%** | 2s | **11.251** | **8.811%** | 10s | | CoEKS | 10.650 | 4.387% | 2s | **15.636** | **5.691%** | 9s |
| VRPBTW | HGS-PyVRP# | 18.292 | * | 10.4m | 29.467 | * | 20.8m | VRPLTW | HGS-PyVRP# | 16.356 | * | 10.4m | 25.757 | * | 20.8m |
| | OR-Tools# | 18.366 | 0.383% | 10.4m | 29.945 | 1.597% | 20.8m | | OR-Tools# | 16.441 | 0.499% | 10.4m | 26.259 | 1.899% | 20.8m |
| | POMO-MTL | 19.105 | 4.409% | 1s | 31.419 | 6.592% | 8s | | POMO-MTL | 16.864 | 3.057% | 1s | 27.041 | 4.929% | 7s |
| | MVMoE | 18.976 | 3.717% | 2s | 31.441 | 6.675% | 10s | | MVMoE | 16.868 | 3.088% | 2s | 26.996 | 4.765% | 10s |
| | RF-TE | 19.029 | 4.011% | 1s | 31.383 | 6.479% | 7s | | RF-TE | 16.865 | 3.069% | 1s | 27.055 | 4.983% | 7s |
| | CaDA | 19.118 | 4.491% | 3s | 31.568 | 7.117% | 13s | | CaDA | **16.780** | **2.514%** | 2s | **26.853** | **4.194%** | 13s |
| | ReLD-MoEL | 18.994 | 3.821% | 2s | 31.218 | 5.920% | 10s | | ReLD-MoEL | 16.951 | 3.599% | 2s | 27.222 | 5.663% | 10s |
| | CoEKS | **18.882** | **3.210%** | 2s | **31.168** | **5.746%** | 10s | | CoEKS | 16.847 | 2.949% | 2s | 26.891 | 4.355% | 10s |
| OVRPBL | HGS-PyVRP# | 6.899 | * | 10.4m | 10.335 | * | 20.8m | OVRPBTW | HGS-PyVRP# | 11.669 | * | 10.4m | 19.156 | * | 20.8m |
| | OR-Tools# | 6.927 | 0.386% | 10.4m | 10.582 | 2.363% | 20.8m | | OR-Tools# | 11.682 | 0.109% | 10.4m | 19.303 | 0.757% | 20.8m |
| | POMO-MTL | 7.451 | 7.937% | 1s | 12.135 | 17.341% | 7s | | POMO-MTL | 12.094 | 3.596% | 1s | 20.255 | 5.690% | 8s |
| | MVMoE | 7.440 | 7.778% | 2s | 11.823 | 14.289% | 10s | | MVMoE | 12.027 | 3.038% | 2s | 20.270 | 5.759% | 11s |
| | RF-TE | 7.638 | 10.632% | 1s | 11.876 | 14.864% | 6s | | RF-TE | 12.088 | 3.548% | 1s | 20.314 | 5.986% | 8s |
| | CaDA | 7.699 | 11.505% | 2s | 11.791 | 14.024% | 12s | | CaDA | 12.143 | 4.005% | 3s | 20.401 | 6.430% | 13s |
| | ReLD-MoEL | 7.344 | 6.380% | 2s | 11.427 | 10.506% | 10s | | ReLD-MoEL | 12.039 | 3.125% | 2s | 20.159 | 5.192% | 11s |
| | CoEKS | **7.230** | **4.747%** | 2s | **11.245** | **8.755%** | 9s | | CoEKS | **11.972** | **2.566%** | 2s | **20.114** | **4.957%** | 10s |
| OVRPLTW | HGS-PyVRP# | 10.51 | * | 10.4m | 16.926 | * | 20.8m | VRPBLTW | HGS-PyVRP# | 18.361 | * | 10.4m | 29.026 | * | 20.8m |
| | OR-Tools# | 10.497 | 0.114% | 10.4m | 17.023 | 0.728% | 20.8m | | OR-Tools# | 18.422 | 0.332% | 10.4m | 29.830 | 2.770% | 20.8m |
| | POMO-MTL | 10.695 | 1.735% | 1s | 17.508 | 3.415% | 7s | | POMO-MTL | 19.482 | 4.746% | 1s | 31.940 | 7.081% | 7s |
| | MVMoE | 10.725 | 2.015% | 2s | 17.489 | 3.300% | 10s | | MVMoE | 19.361 | 4.119% | 2s | 31.996 | 7.286% | 10s |
| | RF-TE | 10.706 | 1.837% | 1s | 17.500 | 3.363% | 7s | | RF-TE | 19.410 | 4.379% | 1s | 31.936 | 7.089% | 7s |
| | CaDA | **10.627** | **1.097%** | 2s | **17.273** | **2.027%** | 13s | | CaDA | 19.472 | 4.713% | 3s | 32.447 | 8.832% | 14s |
| | ReLD-MoEL | 10.702 | 1.791% | 2s | 17.462 | 3.144% | 10s | | ReLD-MoEL | 19.563 | 5.212% | 2s | 32.222 | 8.056% | 11s |
| | CoEKS | 10.681 | 1.603% | 2s | 17.402 | 2.785% | 10s | | CoEKS | **19.283** | **3.714%** | 2s | **31.738** | **6.423%** | 10s |
| OVRPBLTW | HGS-PyVRP# | 11.668 | * | 10.4m | 19.156 | * | 20.8m | OOD Avg. | HGS-PyVRP# | 12.315 | * | 10.4m | 19.437 | * | 20.8m |
| | OR-Tools# | 11.681 | 0.106% | 10.4m | 19.305 | 0.767% | 20.8m | | OR-Tools# | 12.364 | 0.415% | 10.4m | 19.806 | 1.948% | 20.8m |
| | POMO-MTL | 12.101 | 3.659% | 1s | 20.287 | 5.854% | 8s | | POMO-MTL | 12.887 | 4.701% | 1s | 20.950 | 8.341% | 8s |
| | MVMoE | 12.100 | 3.655% | 2s | 20.306 | 5.949% | 10s | | MVMoE | 12.842 | 4.349% | 2s | 20.870 | 7.559% | 10s |
| | RF-TE | 12.063 | 3.342% | 1s | 20.291 | 5.887% | 7s | | RF-TE | 12.884 | 4.823% | 1s | 21.023 | 8.702% | 7s |
| | CaDA | 12.129 | 3.889% | 3s | 20.356 | 6.206% | 13s | | CaDA | 12.940 | 5.496% | 3s | 20.930 | 7.812% | 13s |
| | ReLD-MoEL | 12.082 | 3.490% | 2s | 20.266 | 5.747% | 11s | | ReLD-MoEL | 12.848 | 4.202% | 2s | 20.784 | 6.735% | 11s |
| | CoEKS | **11.999** | **2.797%** | 2s | **20.158** | **5.186%** | 10s | | CoEKS | **12.754** | **3.432%** | 2s | **20.623** | **5.857%** | 10s |

(OOD Avg.): Average performance across OOD VRP tasks. **bold**: Best results among learning-based methods. #: Results are adopted from Berto et al. (2024) for the convenience of comparison. *: Best traditional method, taken as the baseline for gap calculation.

**(Q3) Universality across backbone models.** we implement CoEKS on both the classic POMO (Kwon et al., 2020) and the SOTA ReLD (Huang et al., 2025) backbones. The POMO-based methods include POMO-MTL (Liu et al., 2024b), RF-TE (Berto et al., 2024), MVMoE (Zhou et al., 2024a), and our POMO-CoEKS. The ReLD-based methods include ReLD-MTL, ReLD-RF, ReLD-MoEL (Huang et al., 2025), and our ReLD-CoEKS (i.e., the original CoEKS implementation). MVMoE and ReLD-MoEL retain the MoE modules in their original decoders. Experiments are conducted on $n=50$, and all new methods are trained using the original settings of their corresponding baselines. As illustrated in Figures 3 (a-b), CoEKS consistently delivers the best average gap on both ID and OOD tasks under both backbone models. These results highlight the universality and consistent superiority of CoEKS.

**(Q4) Effectiveness of knowledge sharing strategy.** This section investigates the impact of the proposed multi-view knowledge sharing strategy. Ablation experiments are conducted on 16 VRP tasks with $n=50$, under consistent training settings. The SOTA ReLD-MoEL serves as the baseline for comparison. Specifically, three ablated variants are examined: without (w/o) MDis, w/o shared transformation layer, and w/o both. Figures 3 (c-d) show the average gap for all variants on ID and OOD instances. It demonstrates that the multi-view knowledge sharing strategy significantly enhances model performance, particularly in OOD generalization. The results verify that multi-view knowledge sharing contributes to the learning of transferable knowledge across constraints.

**Position of MDis.** The effect of mutual distillation (MDis) among experts at different encoder layers is studied. Starting from a CoEKS variant w/o MDis, the MDis mechanism is gradually introduced from lower to higher encoder layers, until enabled throughout. As shown in Figures 4 (a-b), incorporating MDis is generally beneficial for ID generalization unless applied to all layers. This suggests that moderate expert collaboration may promote their comprehensive understanding of the training task. For OOD generalization, improvements are observed only when MDis is applied to the

Table 3: Fine-tuning on VRPMB and VRPMBTW at $n$=50.

| Method | VRPMB | | OVRPMB | | VRPMBL | | VRPMBTW | | OVRPMBL | | OVRPMBTW | | VRPMBLTW | | OVRPMBLTW | |
|---|---|---|---|---|---|---|---|---|---|---|---|---|---|---|---|---|
| | Cost | Gap | Cost | Gap | Cost | Gap | Cost | Gap | Cost | Gap | Cost | Gap | Cost | Gap | Cost | Gap |
| HGS-PyVRP | 9.09 | * | 6.11 | * | 16.31 | * | 9.49 | * | 6.11 | * | 10.47 | * | 16.01 | * | 10.47 | * |
| RF-TE-AL | 11.31 | 24.69% | 9.03 | 47.92% | 22.32 | 137.22% | 19.20 | 20.32% | 13.75 | 125.04% | 14.22 | 36.37% | 19.95 | 22.81% | 14.48 | 38.90% |
| RF-TE-EAL | 9.25 | 1.76% | 6.38 | 4.42% | 9.73 | 2.54% | 16.36 | 2.16% | 6.39 | 4.48% | 10.71 | 2.24% | 16.80 | 2.97% | 10.88 | 3.81% |
| ReLD-MoEL-AL | 10.65 | 17.26% | 8.45 | 38.47% | 11.37 | 19.91% | 18.18 | 13.65% | 8.61 | 41.06% | 12.88 | 23.25% | 18.84 | 15.65% | 13.04 | 24.78% |
| ReLD-MoEL-EAL | 9.32 | 2.59% | 6.43 | 5.09% | 9.71 | 2.35% | 16.38 | 2.30% | 6.43 | 5.17% | 10.67 | 1.88% | 16.93 | 3.80% | 10.70 | 2.14% |
| CoEKS$^+$ | 9.24 | 1.61% | 6.37 | 4.19% | 9.71 | 2.23% | 16.33 | 1.92% | 6.35 | 3.84% | **10.64** | **1.57%** | 16.80 | 2.97% | 10.66 | 1.75% |
| CoEKS$^{+c}$ | **9.23** | **1.56%** | **6.33** | **3.52%** | **9.70** | **2.12%** | **16.33** | **1.97%** | **6.33** | **3.48%** | 10.65 | 1.62% | **16.79** | **2.92%** | **10.66** | **1.74%** |

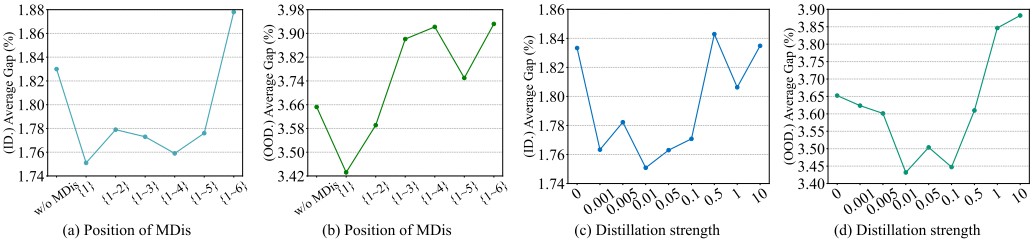

(a) Verification of universality     (b) Verification of universality     (c) Ablation study on CoEKS     (d) Ablation study on CoEKS

Figure 3: *Left two panels:* universality tests both **(a)** ID and **(b)** OOD tasks. *Right two panels:* ablation Study both **(c)** ID and **(d)** OOD tasks.

(a) Position of MDis     (b) Position of MDis     (c) Distillation strength     (d) Distillation strength

Figure 4: *Left two panels:* effect of MDis position on **(a)** ID and **(b)** OOD performance. The horizontal axis is the encoder layer number. *Right two panels:* effect of MDis strength on **(c)** ID and **(d)** OOD performance. The x-axis is the weight $\alpha$.

lower layers. The findings imply that deeper layers encode more task-specific patterns. Applying MDis on these layers may lead to homogenization of knowledge specialized in constraints, thus preventing model generalization to new VRPs with unseen constraint combinations. These results validate our choice of applying MDis only at the first encoder layer, which achieves a good balance between expert interaction and specialization.

**Distillation Strength.** We further assess the impact of the distillation coefficient $\alpha$ by measuring average gaps on both ID and OOD tasks. Figures 4 (c-d) show that a properly chosen $\alpha$ enhances overall performance. However, when $\alpha$ increases beyond a certain threshold, the experts tend to produce overly similar outputs, resulting in no performance gains. This supports our use of $\alpha = 0.01$ as a simple and effective setting.

## 6 CONCLUSION

This paper presents CoEKS, a novel model that leverages the structural characteristic of VRPs to address cross-task challenges. CoEKS integrates two complementary components: a combination-of-experts architecture that adaptively combines constraint-specific experts for diverse VRPs, and a multi-view knowledge sharing strategy that automatically learns transferable knowledge to enhance cross-task generalization. In addition, new experts can be seamlessly plugged into the trained model to handle unseen constraints. Extensive evaluations on 24 VRP tasks demonstrate that CoEKS achieves SOTA performance on ID tasks and yields even greater gains on OOD scenarios, including unseen constraint combinations and new constraints. Furthermore, CoEKS exhibits consistent superiority across backbone models, highlighting its universality.

A current limitation is that handling more constraints inevitably increases the number of parameters. However, this trade-off is natural, since more complex problems with more constraints demand stronger model capability. A promising future direction is to explore more efficient expert-sharing mechanisms enabling a single expert to serve multiple similar constraints, or more efficient parameterization strategies to scale model capability.

## ACKNOWLEDGMENTS

This work is supported by the National Natural Science Foundation of China (62472461), and the Guangdong Basic and Applied Basic Research Foundation (2025A1515010129).

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

# Combination-of-Experts with Knowledge Sharing for Cross-Task Vehicle Routing Problems (Appendix)

## A  DETAILS OF VRPS

This section provides details on the 6 basic constraints described in Section 3.1. By combining these constraints with the base CVRP task, a total of 48 VRP tasks are constructed. CoEKS is evaluated on 16 of these tasks as introduced in MVMoE (Zhou et al., 2024a). To further assess the scalability of the model to new constraints, scalability experiments are conducted on the remaining 32 VRP tasks with Mixed Backhauls or Multi-Depots or both (see Table 4), following RouteFinder (Berto et al., 2024). The data generation process for VRP tasks is detailed below.

**Node Coordinates.** The single depot and all customer nodes are uniformly sampled within the unit square $[0, 1]^2$.

**Capacity (C).** Following RouteFinder and MVMoE, the vehicle capacity $C$ is set to 40 for $n = 50$ and 50 for $n = 100$. For customer $i$, the linehaul demand $d_i$ is sampled from the integer set $\{1, 2, \ldots, 9\}$.

**Backhaul (B).** 20% of customers are randomly selected to sample their backhaul demand from the integer set $\{1, 2, \ldots, 9\}$, while the rest are set to 0. For the selected customers, the linehaul demand is set to 0. As a result, each customer has only one type of demand.

For non-backhaul instances, all backhaul demands are set to 0. Before passing into the model, all linehaul and backhaul demands are normalized by vehicle capacity $C$ to $[0, 1]$.

**Duration Limit (L).** This constraint imposes a maximum route length $L$ per vehicle. Following RouteFinder, $L$ is sampled from $U(2 \max(c_{0i}), 3.0)$, where $c_{0i}$ is the distance from the depot to customer $i$.

**Time Window (TW).** Following RouteFinder, the time window and service time for customer $i$ are generated through a multi-step process to ensure feasibility and diversity:

1. Service time: Sample service time $s_i \sim U[0.15, 0.18]$.
2. Window length: Sample time window length $\Delta t_i \sim U[0.18, 0.2]$.
3. Upper bound: Calculate the upper bound for start time as $u_i = \frac{T_{\max} - s_i - \Delta t_i}{c_{0i}} - 1$, where $T_{\max}$ is the maximum allowed duration for a route.
4. Start time: Set the start time $e_i = (1 + (u_i - 1) \cdot r_i) \cdot c_{0i}$, where $r_i \sim U(0, 1)$.
5. End time: Compute the end time $l_i = e_i + \Delta t_i$.

For the depot node, the time window is fixed to $[0, T_{\max}]$ and the service time is set to 0. In addition, the vehicle speed is 1.0.

**Open Route (O).** The O constraint alters the route structure, allowing vehicles to finish at any customer node instead of returning to the depot. It is implemented by setting a binary indicator $o = 1$, without additional data. When combined with other constraints, feasibility checks are adjusted dynamically:

- With L, the route length is computed without the return-to-depot distance.
- With TW, arrival time calculations omit the depot return segment.
- With C, B, or MB, the constraint logic remains unchanged, while the depot return requirement is removed.

**Mixed Backhauls (MB)** The demand configuration follows the same setup as the backhaul constraint. For instances involving MB, a binary flag $\mu$ is set to 1 and 0 otherwise. This flag is used to distinguish between instances with and without the MB constraint.

Table 4: 48 VRP tasks with 7 constraints. 32 VRP tasks with Mixed Backhauls or Multi-Depots or both are used to evaluate model's scalability in adapting to unseen constraints.

| VRP task | Capacity (C) | Open Route (O) | Backhaul (B) | Mixed Backhauls (MB) | Duration Limit (L) | Time Window (TW) | Multi-Depot (MD) |
|---|---|---|---|---|---|---|---|
| CVRP | ✓ | | | | | | |
| OVRP | ✓ | ✓ | | | | | |
| VRPB | ✓ | | ✓ | | | | |
| VRPL | ✓ | | | | ✓ | | |
| VRPTW | ✓ | | | | | ✓ | |
| OVRPTW | ✓ | ✓ | | | | ✓ | |
| OVRPL | ✓ | ✓ | | | ✓ | | |
| OVRPB | ✓ | ✓ | ✓ | | | | |
| VRPBL | ✓ | | ✓ | | ✓ | | |
| VRPBTW | ✓ | | ✓ | | | ✓ | |
| VRPLTW | ✓ | | | | ✓ | ✓ | |
| OVRPBL | ✓ | ✓ | ✓ | | ✓ | | |
| OVRPBTW | ✓ | ✓ | ✓ | | | ✓ | |
| OVRPLTW | ✓ | ✓ | | | ✓ | ✓ | |
| VRPBLTW | ✓ | | ✓ | | ✓ | ✓ | |
| OVRPBLTW | ✓ | ✓ | ✓ | | ✓ | ✓ | |
| VRPMB | ✓ | | ✓ | ✓ | | | |
| OVRPMB | ✓ | ✓ | ✓ | ✓ | | | |
| VRPMBL | ✓ | | ✓ | ✓ | ✓ | | |
| VRPMBTW | ✓ | | ✓ | ✓ | | ✓ | |
| OVRPMBL | ✓ | ✓ | ✓ | ✓ | ✓ | | |
| OVRPMBTW | ✓ | ✓ | ✓ | ✓ | | ✓ | |
| VRPMBLTW | ✓ | | ✓ | ✓ | ✓ | ✓ | |
| OVRPMBLTW | ✓ | ✓ | ✓ | ✓ | ✓ | ✓ | |
| MDCVRP | ✓ | | | | | | ✓ |
| MDOVRP | ✓ | ✓ | | | | | ✓ |
| MDVRPB | ✓ | | ✓ | | | | ✓ |
| MDVRPL | ✓ | | | | ✓ | | ✓ |
| MDVRPTW | ✓ | | | | | ✓ | ✓ |
| MDOVRPTW | ✓ | ✓ | | | | ✓ | ✓ |
| MDOVRPL | ✓ | ✓ | | | ✓ | | ✓ |
| MDOVRPB | ✓ | ✓ | ✓ | | | | ✓ |
| MDVRPBL | ✓ | | ✓ | | ✓ | | ✓ |
| MDVRPBTW | ✓ | | ✓ | | | ✓ | ✓ |
| MDVRPLTW | ✓ | | | | ✓ | ✓ | ✓ |
| MDOVRPBL | ✓ | ✓ | ✓ | | ✓ | | ✓ |
| MDOVRPBTW | ✓ | ✓ | ✓ | | | ✓ | ✓ |
| MDOVRPLTW | ✓ | ✓ | | | ✓ | ✓ | ✓ |
| MDVRPBLTW | ✓ | | ✓ | | ✓ | ✓ | ✓ |
| MDOVRPBLTW | ✓ | ✓ | ✓ | | ✓ | ✓ | ✓ |
| MDVRPMB | ✓ | | ✓ | ✓ | | | ✓ |
| MDOVRPMB | ✓ | ✓ | ✓ | ✓ | | | ✓ |
| MDVRPMBL | ✓ | | ✓ | ✓ | ✓ | | ✓ |
| MDVRPMBTW | ✓ | | ✓ | ✓ | | ✓ | ✓ |
| MDOVRPMBL | ✓ | ✓ | ✓ | ✓ | ✓ | | ✓ |
| MDOVRPMBTW | ✓ | ✓ | ✓ | ✓ | | ✓ | ✓ |
| MDVRPMBLTW | ✓ | | ✓ | ✓ | ✓ | ✓ | ✓ |
| MDOVRPMBLTW | ✓ | ✓ | ✓ | ✓ | ✓ | ✓ | ✓ |

**Multi-Depots (MD)** The single-depot setting is extended to a multi-depot configuration. Vehicles may start from any depot but must return to the depot they depart from. Appendix E.3 reports results on 24 variants with the MD constraint. Following RouteFinder, the number of depots is fixed to three.

# B  DETAILED ARCHITECTURE OF CoEKS

## B.1  ENCODER

The encoder transforms static node features into embeddings for various VRP tasks. For the $i$-th ($i \in \{1, \ldots, n\}$) customer node, the static feature is defined as $F_i = \{x_i, y_i, d_i, p_i, e_i, l_i, s_i\}$, where $x_i, y_i$ represent coordinates, $d_i, p_i$ denote linehaul and backhaul requirements, $e_i, l_i$ specify the time window, and $s_i$ indicates service time. The depot node $F_0 = \{x_0, y_0, \mu\}$, where $\mu$ is a binary flag indicating the presence of mixed backhaul. These features are projected into an initial embedding $h^0 \in \mathbb{R}^{(n+1) \times d}$ through linear layers:

$$h^0 = \text{Concat}(W_{s_1} F_0, W_{s_2} F_1, W_{s_2} F_2, \ldots, W_{s_2} F_n). \qquad (9)$$

$W_{s_1} \in \mathbb{R}^{3 \times d}$ and $W_{s_2} \in \mathbb{R}^{7 \times d}$ are learnable parameter matrices, where $d = 128$. $N$ ($N = 6$) encoder layers process $h^0$ to produce the final embeddings $h^N$. Each encoder layer comprises

two components: a multi-head attention (MHA) layer followed by a feedforward network (FFN) layer. Both are integrated with residual connections and instance normalization (IN) to stabilize training (Vaswani et al., 2017). Formally, for the $\ell$-th layer ($\ell \in [0, N-1]$):

$$\tilde{h}_i^{\ell} = \text{IN}(h_i^{\ell} + \text{MHA}(h_i^{\ell}, h_i^{\ell}, h_i^{\ell})), \tag{10}$$

$$h_i^{\ell+1} = \text{IN}(\tilde{h}_i^{\ell} + \text{FFN}(\tilde{h}_i^{\ell})). \tag{11}$$

**Multi-head attention (MHA).** The MHA mechanism employs $A$ ($A = 8$) attention heads to compute diverse node interactions in parallel. Their outputs are then aggregated into a unified representation. For an input embedding $h_i^{\ell} \in \mathbb{R}^d$, each head $a \in \{1, 2, \ldots, A\}$ computes query ($Q$), key ($K$), and value ($V$) vectors:

$$Q_i^{\ell,a} = W_Q^a h_i^{\ell}, \quad K_i^{\ell,a} = W_K^a h_i^{\ell}, \quad V_i^{\ell,a} = W_V^a h_i^{\ell}, \tag{12}$$

where $W_Q^a, W_K^a, W_V^a \in \mathbb{R}^{d_k \times d}$ are learnable parameter matrices, and $d_k = d/A = 16$. These projections are computed for all nodes $i \in \{0, 1, \ldots, n\}$, with node 0 as the depot. The compatibility between nodes $i$ and $j \in \{0, 1, \ldots, n\}$ is measured via scaled dot-product attention, followed by a softmax:

$$u_{ij}^{\ell,a} = \text{Softmax}\left(\frac{(Q_i^{\ell,a})^T K_j^{\ell,a}}{\sqrt{d}}\right). \tag{13}$$

A weighted sum over the value vectors is then computed for each head:

$$z_i^{\ell,a} = \sum_{j=0}^{n} u_{ij}^{\ell,a} V_j^{\ell,a}. \tag{14}$$

The outputs from heads are concatenated. Finally, a linear transformation is applied to obtain the output of the $i$-th node in the MHA layer:

$$\text{MHA}(h_i^{\ell}, h_i^{\ell}, h_i^{\ell}) = \text{Concat}(z_i^{\ell,1}, z_i^{\ell,2}, \ldots, z_i^{\ell,A}) W_O, \tag{15}$$

where $W_O \in \mathbb{R}^{d \times d}$ is a learnable parameter matrix.

**Feedforward network (FFN).** The FFN layer contains two linear layers with a ReLU activation:

$$\text{FFN}(\tilde{h}_i^{\ell}) = W_{F_1}^{\ell} \cdot \text{ReLU}(W_{F_2}^{\ell} \tilde{h}_i^{\ell}), \tag{16}$$

where $W_{F_1}^{\ell} \in \mathbb{R}^{d_f \times d}$ and $W_{F_2}^{\ell} \in \mathbb{R}^{d \times d_f}$ are learnable parameter matrices, and $d_f = 512$ denotes the hidden dimension. In our framework, each FFN layer in each encoder layer is replaced with a CoEKS layer, comprising $k$ FFNs $\{\text{FFN}_1, ..., \text{FFN}_k\}$. $k = 5$ aligns with the number of VRP constraints considered in our experiments. It can be flexibly extended to adapt new constraints. According to Eq. (8), Eq. (11) can be rewritten as:

$$h_i^{\ell+1} = \text{IN}(\tilde{h}_i^{\ell} + \sum_{j \in CS} \text{FFN}_j(\tilde{h}_i^{\ell}) \cdot S_j(f_s(\tilde{h}_i^{\ell}))). \tag{17}$$

where $CS$ is the set of constraints activated for the current instance, $S_j(\cdot)$ represents the $j$-th activated combiner function, and $f_s(\cdot)$ is the shared transformation layer.

## B.2 DECODER

The decoder constructs solutions by sequentially selecting nodes based on static embeddings $h^N$ (produced by the encoder) and dynamic features $\mathcal{D}_t = \{c_t, t_t, d_t, o_t, b_t\}$, where $c_t, t_t, d_t, o_t, b_t$ represent the remaining linehaul capacity, current time, current route length, binary open route indicator and remaining backhaul capacity, respectively. At decoding step $t$, the context embedding is computed as $h_t^c = W_c \cdot \text{Concat}(h_{t-1}^N, \mathcal{D}_t)$, where $W_c \in \mathbb{R}^{d \times (d+5)}$ is a learnable parameter matrix and $h_{t-1}^N$ denotes the node embedding visited at step $t-1$. The context embedding is then updated via the MHA layer and the identity mapping function (IDT) (Huang et al., 2025):

$$h_t^{c'} = \text{MHA}(h_t^c, h^N, h^N) + \text{IDT}(h_t^c). \tag{18}$$

In MHA, $h_t^c$ is used to compute queries, and $h^N$ is used to compute keys and values:

$$Q^{c,a} = W_Q^{c,a} h_c^t, \quad K^{c,a} = W_K^{c,a} h^N, \quad V^{c,a} = W_V^{c,a} h^N, \tag{19}$$

where $W_Q^{c,a} \in \mathbb{R}^{d_k \times (d+5)}$, $W_K^{c,a}, W_V^{c,a} \in \mathbb{R}^{d_k \times d}$ are learnable parameter matrices of the $a$-th attention head. Then, the output of the MHA layer is obtained by Eqs. (13)-(15). The IDT function explicitly injects context information into $h_t^{c'}$, complementing the attention-based update,

$$\mathrm{IDT}(h_t^c) = h_{t-1}^N + W^{\mathrm{IDT}} \mathcal{D}_t, \tag{20}$$

where $W^{\mathrm{IDT}} \in \mathbb{R}^{d \times 5}$ is a learnable parameter matrix. $h_t^{c'}$ is passed through an FFN layer (see Eq. 16) with a residual connection to generate the query $q_t^c$. The logits for all nodes are then computed as:

$$s_t^i = \begin{cases} C \cdot \tanh\left( \dfrac{(q_t^c)^T h_i^N}{\sqrt{d}} \right), & \text{if } i \in \mathcal{F}_t \\ -\infty, & \text{otherwise} \end{cases} \tag{21}$$

where $C = 10$ is a clipping hyperparameter that bounds the logits to promote exploration. $\mathcal{F}_t$ represents the set of feasible nodes at step $t$, defined by task-specific constraints. Finally, the node selection probability is computed using softmax over the logits.

## C  TRAINING AND FINE-TUNING DATASETS

### C.1  TRAINING DATASET

The training task set (see Section 5) follows the configuration in MVMoE (Zhou et al., 2024a), covering CVRP, OVRP, VRPB, VRPL, VRPTW, and OVRPTW tasks. In addition, we include the OVRPL task, which is motivated by two key reasons: *(1) More complex instance generation.* Instead of applying a fixed duration limit $L$ as in prior work (Liu et al., 2024b; Zhou et al., 2024a; Huang et al., 2025), we adopt a sampling-based strategy (Berto et al., 2024) to generate $L$, resulting in more diverse and challenging instances. *(2) Complex interaction between constraints O and L.* During training, the VRP tasks involving O and L constraints consistently degrade generalization performance (see validation curves in Figure 5). This instability arises from complex interaction between the two constraints: O enlarges the solution space by removing the depot-return requirement, while L restricts it with strict route-length bounds, resulting in convergence difficulties. To address this, the OVRPL task is included in the training set and all models are retrained to ensure a fair and consistent comparison.

### C.2  FINE-TUNING DATASET

This section explains the rationale for including the VRPMB and VRPMBTW tasks in the fine-tuning task set. Initially, all methods are fine-tuned solely on the VRPMB task, treating the remaining VRP tasks with MB as out-of-distribution (OOD) generalization targets. As shown in Figure 6, the generalization performance of VRP tasks involving both TW and MB degrades significantly. The results may arise from the spatio-temporal conflict between the flexible routing requirements of MB and the strict deadlines imposed by TW. To mitigate this, VRPMBTW is added to the fine-tuning task set. As a result, our method consistently improves performance across all tasks, whereas RF-TE and ReLD-MoEL exhibit convergence failures on several tasks. These findings underscore the challenges of adapting to unseen constraints and demonstrate the superior scalability of CoEKS.

## D  EFFECTS OF DIFFERENT TRAINING SETS

To further investigate the impact of CoEKS under different training datasets, we consider incorporating all possible combinations of basic constraints into the training set for VRP tasks, reflecting RouteFinder's philosophy of establishing a foundational model for VRPs (Berto et al., 2024). To ensure a fair comparison, all methods are trained under the same settings.

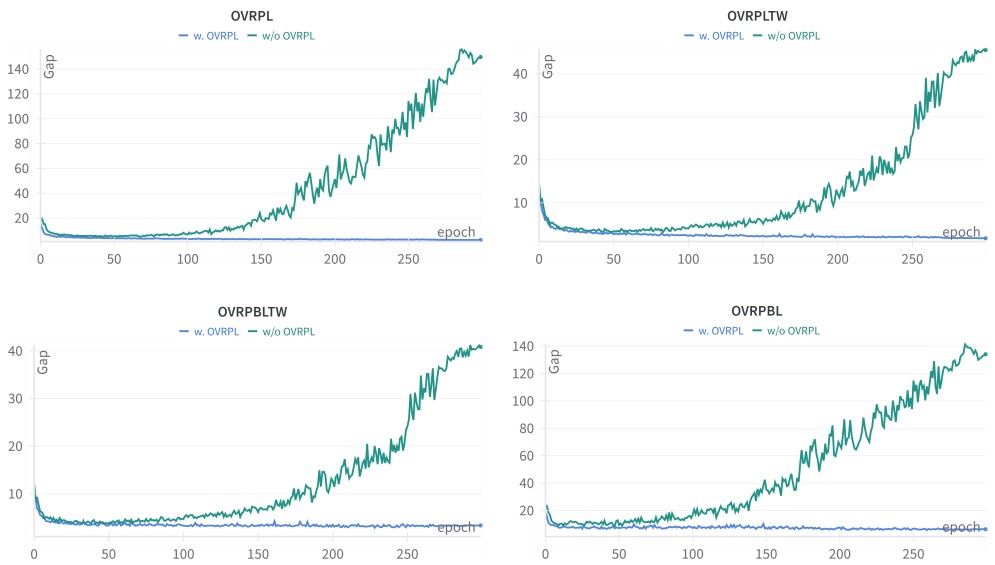

Figure 5: The validation curves of ReLD-MoEL trained without (w/o) OVRPL and with (w.) OVRPL on $n = 50$.

### D.1 TRAINING ON 16 VRP TASKS

The ID generalization results are reported in Table 5, where task types of the test set are identical to the training set. Among the neural methods, CoEKS delivers the best overall performance. CaDA shows strength on TW-constrained tasks but struggles to maintain competitive performance on the remaining variants. In contrast, CoEKS consistently achieves top ranks across all tasks, highlighting its superior ability to balance diverse VRP variants within a unified model. These results represent an in-distribution (ID) scenario, as all constraint combinations are included in the training set.

Beyond this ID setting, CoEKS significantly outperforms CaDA in the OOD scenario (see Table 2), which underscores its potential as a foundation model capable of generalizing to unseen tasks.

### D.2 FINE-TUNING ON ALL VRP TASKS WITH MB

To evaluate scalability to new constraints, we previously fine-tuned CoEKS on a small set of tasks with new constraints and evaluated generalization to all tasks. Building on this, we further consider fine-tuning on all VRP tasks with new constraints, aligning with RouteFinder. Following this setup, all methods are fine-tuned on all VRP tasks with MB. The results are presented in Table 6, where CoEKS consistently demonstrates superior performance across all tasks. Furthermore, the relative gap with comparative methods widens, suggesting that the new expert can further refine itself through interaction with diverse experts. This highlights CoEKS's exceptional scalability to adapt a new constraint.

## E ADDITIONAL EMPIRICAL RESULTS

### E.1 COMPARISON WITH MOE

**Difference from MoE.** 1) *Semantically grounded routing:* MoE architectures typically use learned gating to select top-k experts per node embedding, lacking semantic alignment or cross-task reuse. In contrast, CoEKS better leverages prior knowledge to combine experts, which is both interpretable and efficient. 2) *Broader expert vision:* Gating mechanism of MOE-based methods (Zhou et al., 2024a; Huang et al., 2025) restricts expert vision to a narrow node subset, which weakens experts' cognition of task-level knowledge. In contrast, CoEKS effectively learns constraint-level knowledge through dedicated experts and understand task-level knowledge via combination of experts. 3) *Stable utilization and load balancing:* Existing MOE-based methods (Zhou et al., 2024a; Huang et al.,

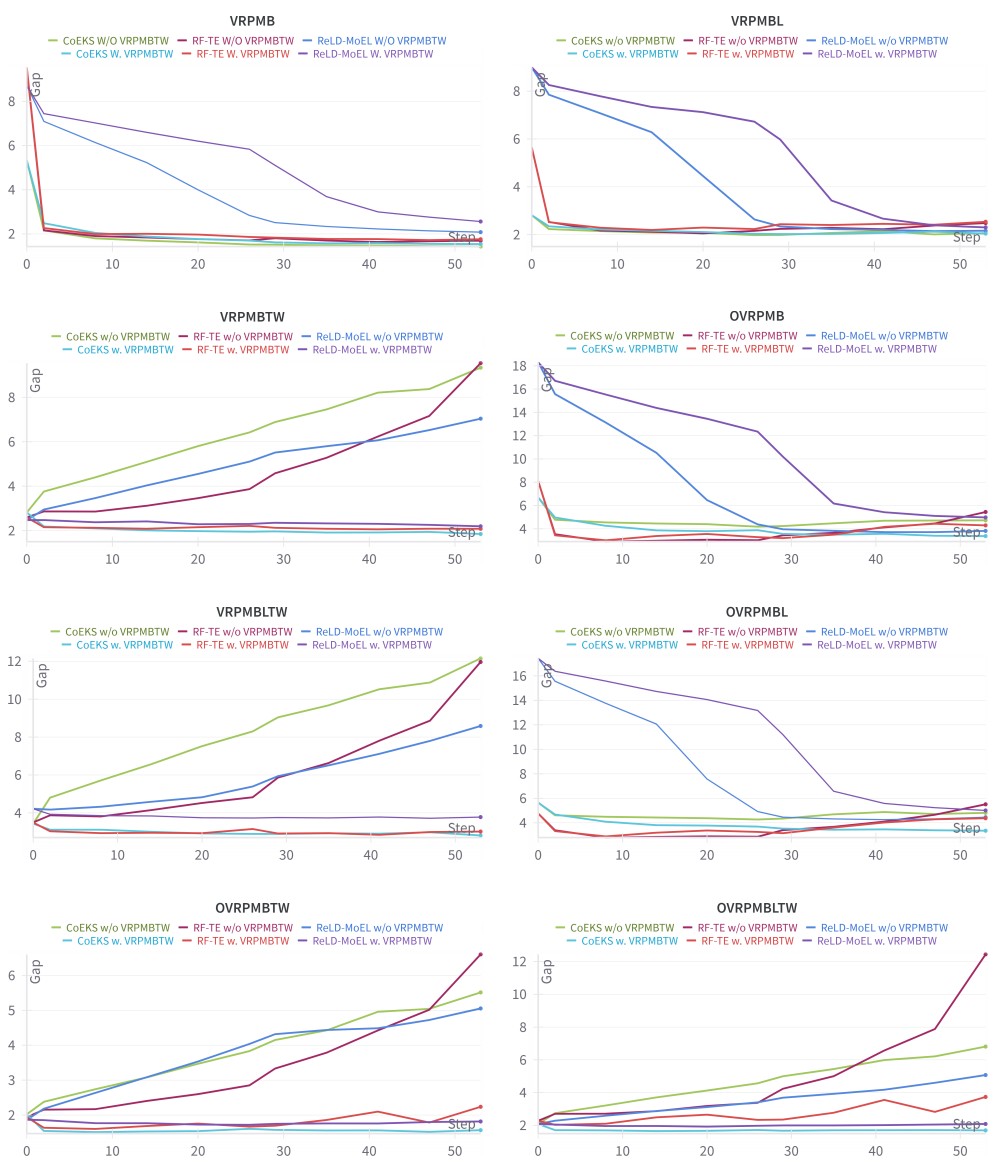

Figure 6: The validation curves of CoEKS, RF-TE, and ReLD-MoEL trained without (w/o) VRPMBTW and with (w.) VRPMBTW on $n = 50$.

2025) typically rely on gating mechanisms to route data to experts, which may lead to load balancing issues. In contrast, CoEKS explicitly activates experts based on task constraints, ensuring more stable and balanced expert utilization. 4) *Scalability via plugging in new experts:* New constraints can be handled by adding and fine-tuning a dedicated expert without modifying the rest of the trained model, as validated in Section 5 (Q2). This structural modularity offers practical advantages over MOE-based methods.

**Parameter efficiency analysis compared with MoE methods:** When compared with MoE-based methods (e.g., ReLD-MoEL (Huang et al., 2025)), CoEKS shares the same decoder architecture, but CoEKS's encoder has one additional expert. To validate CoEKS's parameter efficiency, we use five experts in ReLD-MoEL-E5, resulting in a total of 4.6 million parameters, which matches that of CoEKS. During training, CoEKS activates an average of two experts, aligning with ReLD-MoEL-E5 that activates the top-2 experts. During inference, the number of activated experts in ReLD-MoEL-E5 is dynamically adjusted based on task constraints to maintain consistent parameter usage with CoEKS. The results are presented in Tables 7 and 8. Given the same total and activated parameters,

Table 5: Performance on 1K test instances of 16 VRPs (the training set includes all 16 VRPs).

**Left panel**

| Task | Method | Obj. (n=50) | Gap (n=50) | Time (n=50) | Obj. (n=100) | Gap (n=100) | Time (n=100) |
|---|---|---|---|---|---|---|---|
| CVRP | HGS-PyVRP# | 10.372 | * | 10.4m | 15.628 | * | 20.8m |
| | OR-Tools# | 10.572 | 1.907% | 10.4m | 16.280 | 4.178% | 20.8m |
| | POMO-MTL | 10.520 | 1.429% | 1s | 15.910 | 1.844% | 7s |
| | MVMOE | 10.501 | 1.240% | 2s | 15.880 | 1.641% | 9s |
| | RF-TE | 10.509 | 1.330% | 1s | 15.861 | 1.533% | 7s |
| | CaDA | 10.505 | 1.281% | 3s | 15.856 | 1.489% | 11s |
| | ReLD-MoEL | 10.482 | 1.062% | 9s | 15.832 | 1.340% | 9s |
| | CoEKS | **10.477** | **1.017%** | 2s | **15.816** | **1.242%** | 9s |
| OVPR | HGS-PyVRP# | 6.507 | * | 10.4m | 9.725 | * | 20.8m |
| | OR-Tools# | 6.553 | 0.686% | 10.4m | 9.995 | 2.732% | 20.8m |
| | POMO-MTL | 6.716 | 3.185% | 1s | 10.193 | 4.786% | 6s |
| | MVMOE | 6.702 | 2.967% | 2s | 10.164 | 4.490% | 9s |
| | RF-TE | 6.687 | 2.731% | 1s | 10.119 | 4.031% | 6s |
| | CaDA | 6.684 | 2.679% | 2s | 10.116 | 3.987% | 12s |
| | ReLD-MoEL | 6.679 | 2.616% | 2s | 10.101 | 3.851% | 9s |
| | CoEKS | **6.667** | **2.424%** | 2s | **10.073** | **3.562%** | 8s |
| VRPB | HGS-PyVRP# | 9.687 | * | 10.4m | 14.377 | * | 20.8m |
| | OR-Tools# | 9.802 | 1.159% | 10.4m | 14.933 | 3.853% | 20.8m |
| | POMO-MTL | 10.032 | 3.556% | 1s | 15.054 | 4.725% | 6s |
| | MVMOE | 10.008 | 3.298% | 2s | 15.012 | 4.432% | 8s |
| | RF-TE | 9.986 | 3.083% | 1s | 14.934 | 3.891% | 6s |
| | CaDA | 9.978 | 2.987% | 2s | 14.932 | 3.873% | 11s |
| | ReLD-MoEL | 9.967 | 2.875% | 2s | 14.921 | 3.799% | 9s |
| | CoEKS | **9.948** | **2.674%** | 2s | **14.884** | **3.546%** | 9s |
| VRPBL | HGS-PyVRP# | 10.186 | * | 10.4m | 14.779 | * | 20.8m |
| | OR-Tools# | 10.331 | 1.390% | 10.4m | 15.426 | 4.338% | 20.8m |
| | POMO-MTL | 10.675 | 4.733% | 1s | 15.688 | 6.103% | 7s |
| | MVMOE | 10.632 | 4.309% | 2s | 15.621 | 5.643% | 9s |
| | RF-TE | 10.584 | 3.856% | 1s | 15.515 | 4.950% | 7s |
| | CaDA | 10.569 | 3.708% | 2s | 15.506 | 4.872% | 11s |
| | ReLD-MoEL | 10.567 | 3.675% | 2s | 15.498 | 4.828% | 9s |
| | CoEKS | **10.546** | **3.480%** | 2s | **15.466** | **4.621%** | 9s |
| VRPBTW | HGS-PyVRP# | 18.292 | * | 10.4m | 29.467 | * | 20.8m |
| | OR-Tools# | 18.366 | 0.383% | 10.4m | 29.945 | 1.597% | 20.8m |
| | POMO-MTL | 18.647 | 1.915% | 1s | 30.447 | 3.324% | 7s |
| | MVMOE | 18.637 | 1.863% | 2s | 30.439 | 3.292% | 10s |
| | RF-TE | 18.604 | 1.685% | 1s | 30.265 | 2.702% | 7s |
| | CaDA | **18.519** | **1.227%** | 2s | **30.080** | **2.064%** | 12s |
| | ReLD-MoEL | 18.611 | 1.725% | 2s | 30.349 | 2.986% | 10s |
| | CoEKS | 18.567 | 1.489% | 2s | 30.213 | 2.523% | 10s |
| OVRPB | HGS-PyVRP# | 6.898 | * | 10.4m | 10.335 | * | 20.8m |
| | OR-Tools# | 6.928 | 0.412% | 10.4m | 10.577 | 2.315% | 20.8m |
| | POMO-MTL | 7.106 | 2.989% | 1s | 10.852 | 4.973% | 7s |
| | MVMOE | 7.086 | 2.696% | 2s | 10.825 | 4.707% | 9s |
| | RF-TE | 7.075 | 2.538% | 1s | 10.769 | 4.179% | 6s |
| | CaDA | 7.062 | 2.341% | 2s | 10.745 | 3.941% | 12s |
| | ReLD-MoEL | 7.064 | 2.375% | 2s | 10.754 | 4.034% | 9s |
| | CoEKS | **7.048** | **2.139%** | 2s | **10.710** | **3.611%** | 9s |
| OVRPBLTW | HGS-PyVRP# | 11.668 | * | 10.4m | 19.156 | * | 20.8m |
| | OR-Tools# | 11.681 | 0.106% | 10.4m | 19.305 | 0.767% | 20.8m |
| | POMO-MTL | 11.823 | 1.304% | 1s | 19.635 | 2.482% | 7s |
| | MVMOE | 11.815 | 1.244% | 2s | 19.657 | 2.603% | 10s |
| | RF-TE | 11.804 | 1.145% | 1s | 19.552 | 2.049% | 7s |
| | CaDA | **11.767** | **0.832%** | 3s | **19.434** | **1.430%** | 13s |
| | ReLD-MoEL | 11.807 | 1.170% | 2s | 19.586 | 2.228% | 11s |
| | CoEKS | 11.789 | 1.024% | 2s | 19.541 | 1.990% | 10s |
| OVRPL | HGS-PyVRP# | 6.507 | * | 10.4m | 9.724 | * | 20.8m |
| | OR-Tools# | 6.552 | 0.668% | 10.4m | 10.001 | 2.791% | 20.8m |
| | POMO-MTL | 6.719 | 3.230% | 1s | 10.193 | 4.795% | 6s |
| | MVMOE | 6.701 | 2.949% | 2s | 10.169 | 4.544% | 9s |
| | RF-TE | 6.685 | 2.701% | 1s | 10.119 | 4.033% | 6s |
| | CaDA | 6.684 | 2.679% | 2s | 10.118 | 4.007% | 12s |
| | ReLD-MoEL | 6.677 | 2.588% | 2s | 10.103 | 3.875% | 9s |
| | CoEKS | **6.666** | **2.410%** | 2s | **10.073** | **3.568%** | 9s |

**Right panel**

| Task | Method | Obj. (n=50) | Gap (n=50) | Time (n=50) | Obj. (n=100) | Gap (n=100) | Time (n=100) |
|---|---|---|---|---|---|---|---|
| VRPTW | HGS-PyVRP# | 16.031 | * | 10.4m | 25.423 | * | 20.8m |
| | OR-Tools# | 16.089 | 0.347% | 10.4m | 25.814 | 1.506% | 20.8m |
| | POMO-MTL | 16.421 | 2.432% | 1s | 26.417 | 3.896% | 7s |
| | MVMOE | 16.397 | 2.287% | 2s | 26.389 | 3.780% | 9s |
| | RF-TE | 16.362 | 2.060% | 1s | 26.267 | 3.304% | 7s |
| | CaDA | **16.291** | **1.611%** | 2s | **26.078** | **2.560%** | 12s |
| | ReLD-MoEL | 16.381 | 2.171% | 2s | 26.320 | 3.515% | 9s |
| | CoEKS | 16.332 | 1.873% | 2s | 26.209 | 3.070% | 9s |
| VRPL | HGS-PyVRP# | 10.587 | * | 10.4m | 15.766 | * | 20.8m |
| | OR-Tools# | 10.570 | 2.343% | 10.4m | 16.466 | 5.302% | 20.8m |
| | POMO-MTL | 10.774 | 1.722% | 1s | 16.132 | 2.324% | 6s |
| | MVMOE | 10.749 | 1.491% | 2s | 16.088 | 2.047% | 9s |
| | RF-TE | 10.750 | 1.514% | 1s | 16.057 | 1.865% | 6s |
| | CaDA | 10.745 | 1.465% | 2s | 16.043 | 1.768% | 11s |
| | ReLD-MoEL | 10.728 | 1.303% | 2s | 16.032 | 1.695% | 9s |
| | CoEKS | **10.724** | **1.266%** | 2s | **16.023** | **1.644%** | 9s |
| OVRPTW | HGS-PyVRP# | 10.510 | * | 10.4m | 16.926 | * | 20.8m |
| | OR-Tools# | 10.519 | 0.078% | 10.4m | 17.027 | 0.583% | 20.8m |
| | POMO-MTL | 10.673 | 1.526% | 1s | 17.418 | 2.880% | 7s |
| | MVMOE | 10.671 | 1.511% | 2s | 17.429 | 2.946% | 10s |
| | RF-TE | 10.654 | 1.350% | 1s | 17.333 | 2.377% | 7s |
| | CaDA | **10.622** | **1.041%** | 2s | **17.230** | **1.772%** | 12s |
| | ReLD-MoEL | 10.658 | 1.391% | 2s | 17.368 | 2.593% | 10s |
| | CoEKS | 10.638 | 1.192% | 2s | 17.313 | 2.263% | 10s |
| VRPBLTW | HGS-PyVRP# | 18.361 | * | 10.4m | 29.026 | * | 20.8m |
| | OR-Tools# | 18.422 | 0.332% | 10.4m | 29.830 | 2.770% | 20.8m |
| | POMO-MTL | 19.001 | 2.186% | 1s | 30.934 | 3.740% | 7s |
| | MVMOE | 18.981 | 2.083% | 2s | 30.905 | 3.648% | 10s |
| | RF-TE | 18.942 | 1.885% | 1s | 30.719 | 3.026% | 7s |
| | CaDA | **18.858** | **1.432%** | 2s | **30.531** | **2.393%** | 13s |
| | ReLD-MoEL | 18.959 | 1.966% | 2s | 30.800 | 3.299% | 10s |
| | CoEKS | 18.913 | 1.728% | 2s | 30.680 | 2.896% | 10s |
| VRPLTW | HGS-PyVRP# | 16.356 | * | 10.4m | 25.757 | * | 20.8m |
| | OR-Tools# | 16.441 | 0.499% | 10.4m | 26.259 | 1.899% | 20.8m |
| | POMO-MTL | 16.833 | 2.886% | 1s | 26.895 | 4.379% | 7s |
| | MVMOE | 16.804 | 2.712% | 2s | 26.858 | 4.234% | 9s |
| | RF-TE | 16.751 | 2.389% | 1s | 26.717 | 3.690% | 7s |
| | CaDA | **16.682** | **1.964%** | 2s | **26.525** | **2.945%** | 13s |
| | ReLD-MoEL | 16.767 | 2.496% | 2s | 26.768 | 3.894% | 10s |
| | CoEKS | 16.737 | 2.298% | 2s | 26.673 | 3.517% | 10s |
| OVRPBL | HGS-PyVRP# | 6.899 | * | 10.4m | 10.335 | * | 20.8m |
| | OR-Tools# | 6.927 | 0.386% | 10.4m | 10.582 | 2.363% | 20.8m |
| | POMO-MTL | 7.111 | 3.046% | 1s | 10.863 | 5.081% | 7s |
| | MVMOE | 7.096 | 2.818% | 2s | 10.832 | 4.778% | 9s |
| | RF-TE | 7.079 | 2.580% | 1s | 10.772 | 4.207% | 7s |
| | CaDA | 7.064 | 2.362% | 2s | 10.748 | 3.962% | 11s |
| | ReLD-MoEL | 7.063 | 2.354% | 2s | 10.750 | 3.990% | 9s |
| | CoEKS | **7.050** | **2.155%** | 2s | **10.711** | **3.616%** | 9s |
| OVRPBTW | HGS-PyVRP# | 11.669 | * | 10.4m | 19.156 | * | 20.8m |
| | OR-Tools# | 11.682 | 0.109% | 10.4m | 19.303 | 0.757% | 20.8m |
| | POMO-MTL | 11.821 | 1.293% | 1s | 19.631 | 2.464% | 7s |
| | MVMOE | 11.814 | 1.231% | 2s | 19.654 | 2.587% | 10s |
| | RF-TE | 11.804 | 1.140% | 1s | 19.551 | 2.045% | 8s |
| | CaDA | **11.766** | **0.828%** | 3s | **19.435** | **1.432%** | 13s |
| | ReLD-MoEL | 11.808 | 1.177% | 2s | 19.586 | 2.230% | 11s |
| | CoEKS | 11.791 | 1.033% | 2s | 19.539 | 1.980% | 10s |
| OVRPLTW | HGS-PyVRP# | 10.510 | * | 10.4m | 16.926 | * | 20.8m |
| | OR-Tools# | 10.497 | 0.114% | 10.4m | 17.023 | 0.728% | 20.8m |
| | POMO-MTL | 10.672 | 1.520% | 1s | 17.421 | 2.896% | 7s |
| | MVMOE | 10.673 | 1.531% | 2s | 17.432 | 2.967% | 10s |
| | RF-TE | 10.655 | 1.361% | 1s | 17.331 | 2.370% | 7s |
| | CaDA | **10.622** | **1.047%** | 2s | **17.230** | **1.768%** | 12s |
| | ReLD-MoEL | 10.661 | 1.411% | 2s | 17.368 | 2.589% | 10s |
| | CoEKS | 10.639 | 1.204% | 2s | 17.319 | 2.293% | 10s |

**bold**: Best results among learning-based methods.
underline: Second-best results among learning-based methods.
#: Results are adopted from Berto et al. (2024) for the convenience of comparison.

CoEKS consistently outperforms ReLD-MoEL-E5 across all tasks, in both ID and OOD scenarios. This underscores that CoEKS achieves superior parameter efficiency through its model architecture.

## E.2 T-SNE VISUALIZATION ANALYSIS

To gain insights into how the experts in CoEKS learn and specialize, we visualize their embedding tokens using t-distributed Stochastic Neighbor Embedding (t-SNE). We performed t-SNE analysis

Table 6: Fine-tuning performance on all VRPs with MB.

| Method | VRPMB | | OVRPMB | | VRPMBL | | VRPMBTW | | OVRPMBL | | OVRPMBTW | | VRPMBLTW | | OVRPMBLTW | |
|---|---|---|---|---|---|---|---|---|---|---|---|---|---|---|---|---|
| | Cost | Gap | Cost | Gap | Cost | Gap | Cost | Gap | Cost | Gap | Cost | Gap | Cost | Gap | Cost | Gap |
| HGS-PyVRP | 9.09 | * | 6.11 | * | 16.31 | * | 9.49 | * | 6.11 | * | 10.47 | * | 16.01 | * | 10.47 | * |
| RF-TE-AL | 11.74 | 29.66% | 9.44 | 54.62% | 11.27 | 18.94% | 18.46 | 15.59% | 8.58 | 40.46% | 13.27 | 27.14% | 18.88 | 16.06% | 13.29 | 27.33% |
| RF-TE-EAL | 9.36 | 2.98% | 6.26 | 2.47% | 9.74 | 2.67% | 16.40 | 2.38% | 6.26 | 2.42% | 10.66 | 1.73% | 16.79 | 2.94% | 10.66 | 1.76% |
| ReLD-MoEL-AL | 10.63 | 17.07% | 8.13 | 33.15% | 11.09 | 16.99% | 18.22 | 13.88% | 8.21 | 34.49% | 12.61 | 20.60% | 18.64 | 14.39% | 12.64 | 20.93% |
| ReLD-MoEL-EAL | 9.34 | 2.73% | 6.28 | 2.66% | 9.71 | 2.28% | 16.39 | 2.34% | 6.28 | 2.73% | 10.66 | 1.75% | 16.78 | 2.86% | 10.66 | 1.80% |
| CoEKS$^+$ | 9.25 | 1.73% | 6.21 | 1.68% | 9.68 | 1.97% | 16.35 | 2.07% | 6.22 | 1.75% | 10.63 | 1.44% | 16.74 | 2.65% | 10.64 | 1.54% |
| CoEKS$^{c+}$ | **9.24** | **1.66%** | **6.20** | **1.50%** | **9.66** | **1.81%** | **16.35** | **2.07%** | **6.21** | **1.57%** | **10.63** | **1.43%** | **16.73** | **2.56%** | **10.63** | **1.50%** |

Table 7: Parameter efficiency comparison. (In-distribution tasks ($n = 50$))

| Method\Gap↓ | CVRP | OVRP | VRPB | VRPL | VRPTW | OVRPTW | OVRPL | (ID Avg.) |
|---|---|---|---|---|---|---|---|---|
| ReLD-MoEL-E5 | 1.086% | 2.324% | 2.509% | 1.180% | 2.308% | 1.628% | 2.465% | 2.069% |
| CoEKS | **0.891%** | **2.138%** | **2.497%** | **1.152%** | **2.050%** | **1.393%** | **2.135%** | **1.751%** |

Table 8: Parameter efficiency comparison. (Out-of-distribution tasks ($n = 50$))

| Method\Gap↓ | OVRPB | VRPBL | VRPBTW | VRPLTW | OVRPBL | OVRPBTW | OVRPLTW | VRPBLTW | OVRPBLTW | (OOD Avg.) |
|---|---|---|---|---|---|---|---|---|---|---|
| ReLD-MoEL-E5 | 6.753% | **4.323%** | 3.609% | 4.491% | 6.738% | 2.673% | 1.780% | 6.262% | 3.026% | 4.406% |
| CoEKS | **4.913%** | 4.387% | **3.210%** | **2.949%** | **4.747%** | **2.566%** | **1.603%** | **3.714%** | **2.797%** | **3.432%** |

on the expert embeddings for the full-constraint task OVRPBLTW. For each expert at each encoder layer, we sampled 5,000 embedding samples and projected them into a 2D space for visualization.

As illustrated in Figure 7, the t-SNE plots reveal a significant overlap in the embeddings of all experts, except for the capacity expert ($E_C$), within the lower layers (layer 1). This indicates that the lower layers capture shared, transferable representations, aligning with our design where knowledge sharing is applied at the lower layers. The capacity expert ($E_C$), which is active across all VRP tasks, learns more universal representations, resulting in its distinct and stable representation early in the model. As we move to deeper layers, the clusters become clearly separated, showing that experts gradually specialize and align with their assigned constraints.

We also visualize the expert embeddings for the variant without the knowledge sharing strategy in Figure 8. In this case, experts' representations begin to separate even in the first layer, indicating early specialization and a lack of transferable knowledge. This confirms that our knowledge sharing strategy is necessary for building meaningful cross-task representations.

### E.3 SCALABILITY TO CONSTRAINT MULTI-DEPOTS

To further verify the scalability of CoEKS to new constraints, we introduced the Multi-Depot (MD) tasks, which is an extension of the single-depot tasks. Following the configuration of RouteFinder, we set the number of depots to 3. In total, we added 24 new VRP tasks, which include:

- 16 tasks that incorporate only the new MD constraint.
- 8 tasks that combine the new MD with the new MB constraint.

The experimental results are presented in Tables 9, 10, Table 11 and 12, where CoEKS continues to achieve the best performance under both few-shot fine-tuning and zero-shot generalization. These findings strongly support our claim that CoEKS is robustly scalable to diverse and previously unseen constraints.

### E.4 CVRPLIB BENCHMARK

On the CVRPLIB benchmark dataset, we conduct additional experiments to evaluate the generalization ability of the model on real-world instances. Several representative cross-task methods are compared as a supplementary validation of OOD generalization performance. Each model is trained on uniformly distributed instances with $n = 100$. Additionally, the original POMO model (Kwon et al., 2020) trained on a single task is also reported. The evaluation primarily focuses on large-scale datasets ($n > 500$) in the classic Set-X (Uchoa et al., 2017), following the setup in MVMoE (Zhou

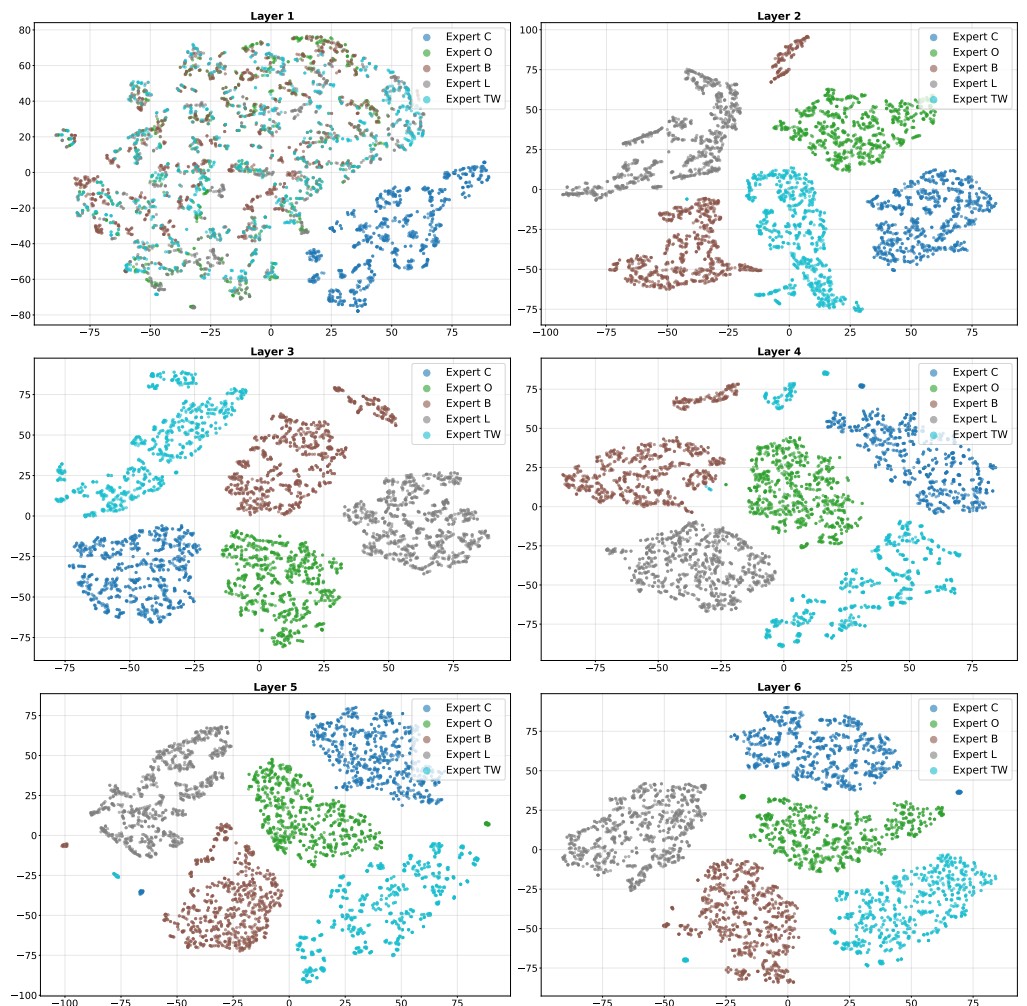

Figure 7: t-SNE visualization of 5 experts' latent representations across encoder layers.

Table 9: Zero-shot generalization performance (the gap to the best traditional solver) on 24 VRPs with MD. These methods do not add new experts and are only allowed to activate experts corresponding to previously known constraints.

| Method\Gap↓ | MDCVRP | MDOVRP | MDOVRPB | MDOVRPBL | MDOVRPBLTW | MDOVRPBTW | MDOVRPL | MDOVRPLTW |
|---|---|---|---|---|---|---|---|---|
| RF-EAL | **37.514%** | 29.887% | 37.903% | 40.128% | 38.314% | 45.111% | 29.934% | 34.305% |
| ReLD-MoEL | 39.359% | 20.266% | 33.504% | 32.543% | 25.134% | 28.718% | 20.108% | 25.210% |
| CoEKS | 38.174% | **15.678%** | **22.690%** | **21.883%** | **21.920%** | **22.901%** | **15.616%** | **20.949%** |
| Method\Gap↓ | MDOVRPTW | MDVRPB | MDVRPBL | MDVRPBLTW | MDVRPBTW | MDVRPL | MDVRPLTW | MDVRPTW |
| RF-EAL | 40.836% | 45.689% | 59.955% | 42.723% | 46.764% | 43.181% | 39.398% | 42.167% |
| ReLD-MoEL | 28.634% | 48.687% | 47.975% | 30.311% | 39.094% | 39.211% | 30.267% | 36.833% |
| CoEKS | **21.730%** | **38.612%** | **38.188%** | **27.884%** | **31.094%** | **36.288%** | **26.986%** | **28.731%** |

Table 10: Zero-shot generalization performance (the gap to the best traditional solver) on 8 VRPs with MB with MD. These methods do not add new experts and are only allowed to activate experts corresponding to previously known constraints.

| Method\Gap↓ | MDOVRPMB | MDOVRPMBL | MDOVRPMBLTW | MDOVRPMBTW | MDVRPMB | MDVRPMBL | MDVRPMBLTW | MDVRPMBTW |
|---|---|---|---|---|---|---|---|---|
| RF-EAL | 44.410% | 44.523% | 38.577% | 38.608% | 58.138% | 57.110% | 41.310% | 41.821% |
| ReLD-MoEL | 47.741% | 45.581% | 26.322% | 29.812% | 64.400% | 60.381% | 31.776% | 38.646% |
| CoEKS | **35.186%** | **33.701%** | **24.938%** | **25.365%** | **53.428%** | **50.735%** | **30.900%** | **32.874%** |

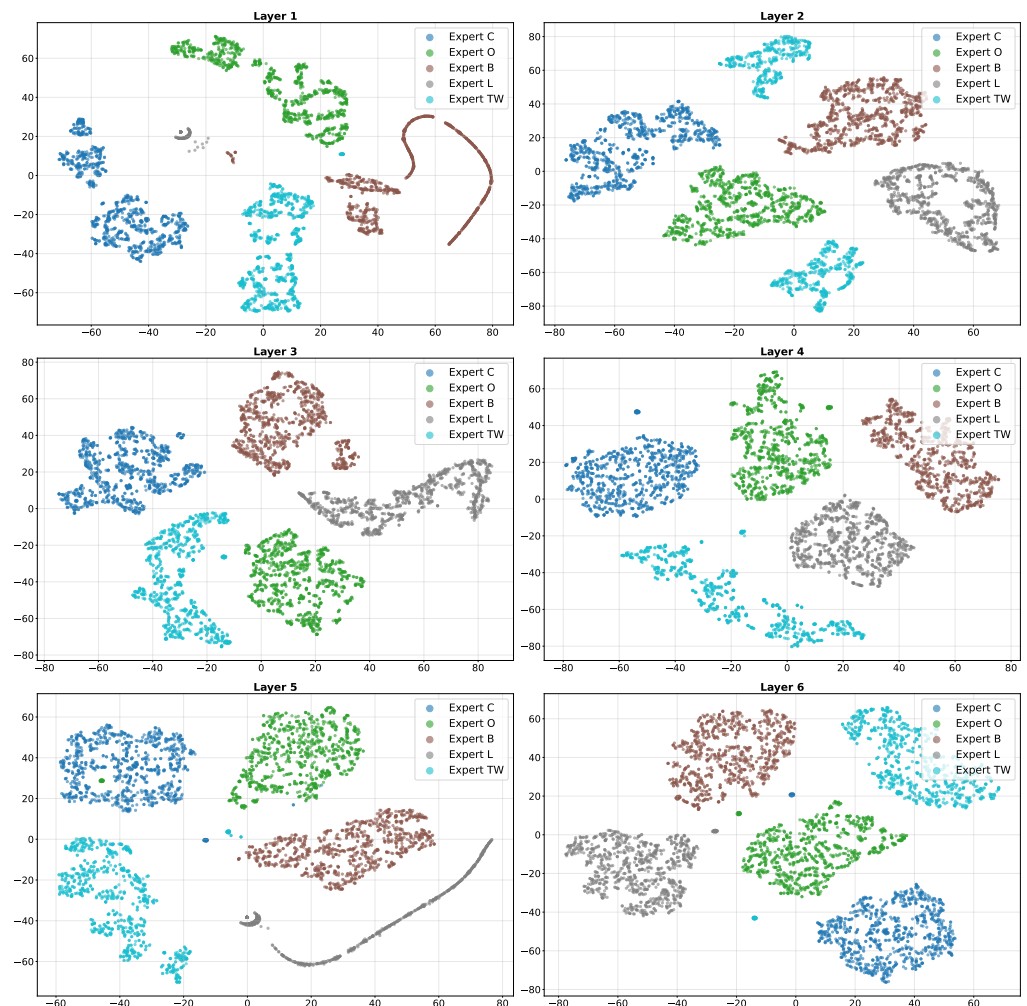

Figure 8: t-SNE visualization of 5 experts' latent representations across encoder layers (without knowledge sharing strategy)

.

Table 11: Fine-tuning performance (the gap to the best traditional solver) on 24 VRPs with MD.

| Method\Gap↓ | MDCVRP | MDOVRP | MDOVRPB | MDOVRPBL | MDOVRPBLTW | MDOVRPBTW | MDOVRPL | MDOVRPLTW |
|---|---|---|---|---|---|---|---|---|
| RF-EAL | 15.251% | 14.250% | 18.996% | 19.479% | 19.841% | 19.315% | 15.421% | 17.977% |
| ReLD-MoEL | 15.235% | 13.525% | 17.125% | 16.933% | 11.405% | 11.278% | 13.333% | 10.758% |
| CoEKS | **8.928%** | **6.491%** | **7.599%** | **7.588%** | **5.324%** | **5.318%** | **6.543%** | **5.038%** |

| Method\Gap↓ | MDOVRPTW | MDVRPB | MDVRPBL | MDVRPBLTW | MDVRPBTW | MDVRPL | MDVRPLTW | MDVRPTW |
|---|---|---|---|---|---|---|---|---|
| RF-EAL | 17.300% | 21.943% | 23.149% | 25.549% | 24.098% | 15.537% | 21.952% | 20.244% |
| ReLD-MoEL | 10.622% | 20.408% | 20.572% | 15.326% | 15.340% | 15.315% | 14.502% | 13.966% |
| CoEKS | **4.978%** | **12.093%** | **12.487%** | **9.532%** | **9.324%** | **9.109%** | **8.970%** | **8.642%** |

Table 12: Fine-tuning performance (the gap to the best traditional solver) on 8 VRPs with MB with MD.

| Method\Gap↓ | MDOVRPMB | MDOVRPMBL | MDOVRPMBLTW | MDOVRPMBTW | MDVRPMB | MDVRPMBL | MDVRPMBLTW | MDVRPMBTW |
|---|---|---|---|---|---|---|---|---|
| RF-EAL | 16.536% | 18.378% | 18.301% | 17.592% | 33.316% | 21.222% | 21.698% | 19.965% |
| ReLD-MoEL | 17.790% | 17.543% | 11.430% | 11.206% | 23.673% | 22.101% | 15.081% | 14.417% |
| CoEKS | **6.959%** | **7.061%** | **5.250%** | **5.130%** | **12.128%** | **12.032%** | **9.122%** | **8.804%** |

et al., 2024a) and RouteFinder (Berto et al., 2024). CoEKS achieves the best OOD generalization, surpassing the state-of-the-art ReLD-MoEL (Huang et al., 2025). The performance gains be-

come more pronounced as the problem size increases. Notably, the single-task training method (i.e., POMO) demonstrates limited generalization ability on diverse real-world benchmarks, potentially due to overfitting the uniform training distribution. Conversely, cross-task training substantially enhances model generalization.

Table 13: Results on large-scale CVRPLIB instances. # Results are adopted from MVMoE(Zhou et al., 2024a), with the model trained on a single task.

| Set-X | POMO# | | RF-TE | | POMO-MTL | | MVMoE | | ReLD-MoEL | | CoEKS | |
|---|---|---|---|---|---|---|---|---|---|---|---|---|
| Instance | Obj. | Gap | Obj. | Gap | Obj. | Gap | Obj. | Gap | Obj. | Gap | Obj. | Gap |
| X-n502-k39 | 75617 | 9.232% | 72098 | **4.149%** | 84021 | 21.372% | 81611 | 17.891% | 73073 | 5.557% | 73947 | 6.820% |
| X-n513-k21 | 30518 | 26.102% | 30330 | 25.325% | 29022 | 19.921% | 27368 | 13.086% | 27063 | **11.826%** | 27124 | 12.078% |
| X-n524-k153 | 201877 | 30.586% | 168473 | **8.978%** | 173838 | 12.449% | 174427 | 12.830% | 174430 | 12.832% | 174325 | 12.764% |
| X-n536-k96 | 106073 | 11.837% | 102320 | **7.880%** | 106851 | 12.657% | 105167 | 10.882% | 102548 | 8.121% | 103497 | 9.121% |
| X-n548-k50 | 103093 | 18.908% | 102078 | 17.737% | 102217 | 17.897% | 107767 | 24.299% | 99121 | **14.326%** | 103115 | 18.933% |
| X-n561-k42 | 49370 | 15.575% | 49632 | 16.188% | 48553 | 13.662% | 47759 | 11.803% | 47022 | 10.078% | 46838 | **9.647%** |
| X-n573-k30 | 83545 | 64.871% | 55296 | 9.123% | 60870 | 20.123% | 66531 | 31.295% | 57249 | 12.977% | 54699 | **7.945%** |
| X-n586-k159 | 229887 | 20.792% | 208397 | 9.501% | 211421 | 11.089% | 214247 | 12.574% | 206793 | 8.658% | 205612 | **8.037%** |
| X-n599-k92 | 150572 | 38.839% | 117226 | 8.091% | 122028 | 12.519% | 126915 | 17.025% | 116463 | **7.388%** | 116547 | 7.465% |
| X-n613-k62 | 68451 | 14.976% | 68066 | 14.329% | 82141 | 37.971% | 67944 | 14.124% | 67272 | 12.996% | 66050 | **10.943%** |
| X-n627-k43 | 84434 | 35.825% | 69046 | 11.071% | 70923 | 14.090% | 70572 | 13.526% | 68141 | 9.615% | 67571 | **8.698%** |
| X-n641-k35 | 75573 | 18.672% | 73071 | 14.740% | 72378 | 13.652% | 70445 | 10.616% | 69360 | 8.913% | 68650 | **7.798%** |
| X-n655-k131 | 127211 | 19.134% | 112355 | **5.221%** | 123144 | 15.325% | 126352 | 18.329% | 120650 | 12.989% | 113905 | 6.673% |
| X-n670-k130 | 208079 | 42.197% | 167786 | 14.661% | 167131 | 14.214% | 168834 | 15.377% | 169163 | 15.602% | 167007 | **14.129%** |
| X-n685-k75 | 79482 | 16.534% | 77681 | 13.893% | 99452 | 45.813% | 78080 | 14.478% | 78090 | 14.493% | 76402 | **12.018%** |
| X-n701-k44 | 97843 | 19.433% | 92541 | 12.961% | 90283 | 10.205% | 89840 | 9.664% | 87883 | 7.275% | 87862 | **7.249%** |
| X-n716-k35 | 51381 | 18.463% | 50333 | 16.047% | 49420 | 13.942% | 50218 | 15.782% | 47981 | 10.624% | 47793 | **10.191%** |
| X-n733-k159 | 159098 | 16.823% | 162059 | 18.997% | 184714 | 35.633% | 153087 | 12.409% | 153884 | 12.995% | 150508 | **10.516%** |
| X-n749-k98 | 87786 | 13.611% | 85623 | 10.812% | 88493 | 14.526% | 86961 | 12.543% | 86380 | 11.791% | 84974 | **9.972%** |
| X-n766-k71 | 135464 | 18.395% | 132819 | 16.083% | 127674 | 11.587% | 129107 | 12.839% | 126139 | 10.245% | 125801 | **9.950%** |
| X-n783-k48 | 90289 | 24.733% | 86445 | 19.422% | 84220 | 16.348% | 82163 | 13.507% | 80269 | 10.890% | 79444 | **9.751%** |
| X-n801-k40 | 124278 | 69.536% | 92149 | 25.696% | 96438 | 31.546% | 88091 | 20.161% | 85315 | **16.374%** | 86477 | 17.959% |
| X-n819-k171 | 193451 | 22.344% | 187863 | 18.810% | 188537 | 19.236% | 187714 | 18.715% | 175282 | 10.853% | 173464 | **9.703%** |
| X-n837-k142 | 237884 | 22.787% | 209629 | 8.203% | 218437 | 12.749% | 223912 | 15.575% | 210889 | 8.853% | 208673 | **7.709%** |
| X-n856-k95 | 152528 | 71.447% | 99082 | 11.372% | 157894 | 77.479% | 175074 | 96.790% | 100320 | 12.763% | 98740 | **10.987%** |
| X-n876-k59 | 119764 | 20.609% | 109566 | 10.339% | 110488 | 11.268% | 115516 | 16.331% | 106631 | **7.384%** | 106684 | 7.437% |
| X-n895-k37 | 70245 | 30.421% | 67995 | 26.244% | 67527 | 25.375% | 64649 | 20.032% | 62172 | 15.433% | 61740 | **14.631%** |
| X-n916-k207 | 399372 | 21.324% | 354011 | 7.544% | 382125 | 16.084% | 372237 | 13.080% | 355853 | 8.103% | 352206 | **6.995%** |
| X-n936-k151 | 237625 | 79.049% | 164931 | 24.275% | 193030 | 45.447% | 160648 | 21.047% | 160460 | 20.906% | 158551 | **19.467%** |
| X-n957-k87 | 130850 | 53.104% | 110516 | 29.311% | 108401 | 26.837% | 127388 | 49.053% | 101629 | **18.913%** | 103700 | 21.336% |
| X-n979-k58 | 147687 | 24.132% | 133825 | 12.481% | 134759 | 13.266% | 132546 | 11.406% | 129738 | 9.046% | 129074 | **8.487%** |
| X-n1001-k43 | 100399 | 38.759% | 92837 | 28.308% | 89098 | 23.140% | 86107 | 19.006% | 81081 | 12.060% | 80458 | **11.199%** |
| Avg. Gap | 29.66% | | 14.931% | | 21.482% | | 19.252% | | 11.590% | | **10.832%** | |

# F  THE USE OF LARGE LANGUAGE MODELS (LLMs)

In this research, we employed Large Language Models (LLMs) as a general-purpose tool to assist with writing polish. These LLMs were utilized to enhance textual clarity without contributing to research conception or methodological development.

