# OpenReview forum: "Combination-of-Experts with Knowledge Sharing for Cross-Task Vehicle Routing Problems"
_ICLR.cc/2026/Conference — ICLR 2026 Poster_

### Official Review · Reviewer_qhjy · 2025-10-30

**Soundness:** 2
**Presentation:** 3
**Contribution:** 2
**Rating:** 2
**Confidence:** 4

**Summary:**

This paper proposes CoEKS, a novel neural routing solver for cross-task VRPs. The core idea is to leverage the structural property that VRP variants are defined by combinations of basic constraints. The model consists of two main components: 1) a Combination-of-Experts (CoE) architecture, where individual constraint-specific FFNs are activated via a prior assignment based on known constraints, and 2) a multi-view knowledge sharing strategy to enhance the learning of transferable knowledge. The authors claim this method achieves state-of-the-art performance on ID tasks and demonstrates superior generalization on OOD tasks, including unseen constraint combinations and rapid adaptation to new constraints by plugging in new experts.

**Strengths:**

1. The paper is well-written and easy to follow.

2. The experimental design is rigorous and comprehensive, the inclusion of strong SOTA baselines provides a high-quality empirical evaluation.

3. The implementation on both POMO and ReLD backbone networks demonstrates the universality of the CoEKS architecture.

**Weaknesses:**

My main concerns stem from the model's core assumption, CoEKS must rely on **prior assignment of constraint-specific experts**, which limits its technical novelty and the general applicability of its claims.

1. The contribution is relatively incremental. The model essentially replaces a learned gating mechanism in standard MoE-based multi-task solvers (ReLD-MoEL[1], MVMoE[2]) with a handcrafted external gating mechanism. While the idea of constraint-specific experts is effective, the implementation offloads the most complex gating selection logic to the user (or what we might call the real expert with domain knowledge).

2. For the scalability for adaptation to unseen constraints, Appendix C.2 mentions the spatio-temporal conflict between MB and TW. Although the model's overall performance is promising, the process requires the inclusion of a TW  constraint to mitigate performance degradation.  I also notice that CoEKS is less effective than ReLD[1] on some tasks when only the MB constraint is applied.

3. The claim of robust scalability to new constraints is supported by a weak experiment. Mixed Backhauls (MB) is merely a relaxation of the Backhaul constraint, which is already part of the training set. Both constraints deal with the same problem type (linehaul and backhaul demands). This is an easy test case and does not sufficiently support the claim that the method can adapt to any new constraint.

4. The authors acknowledge that handling more constraints increases parameters. This one-to-one mapping of constraints-to-experts seems inefficient.

**Questions:**

1. It would greatly enhance the paper's contribution to evaluate the zero-shot generalization capabilities on unseen constraints(e.g., Mixed Backhauls and Multi-Depot) without applying additional expert networks and fine-tuning operations.

2. For the scalability for adaptation to unseen constraints, the authors conducted fine-tuning experiments on unseen constraints with 50-node VRP variants. I am curious about how the performance of CoEKS would compare with existing methods if the problem size increases to 100?

References:

[1] Huang, Z., Zhou, J., Cao, Z., & XU, Y. Rethinking Light Decoder-based Solvers for Vehicle Routing Problems. ICLR, 2025.

[2] Zhou, J., Cao, Z., Wu, Y., Song, W., Ma, Y., Zhang, J., & Xu, C. MVMoE: Multi-Task Vehicle Routing Solver with Mixture-of-Experts. ICML, 2024.

---

> ### Author Response · Authors · 2025-11-21
> **Response to Reviewer qhjy [1/6]**
>
> We sincerely appreciate the reviewer's recognition of our paper as well-written and easy to follow, and for acknowledging the rigorous and comprehensive experimental design and the universality of the CoEKS architecture. Below, we address each concern point by point. All revisions and additions made to the manuscript have been marked in blue to aid in your review.
>
> ## To the weakness of prior assignment of constraint-specific experts: "My main concerns ... applicability of its claims."
>
> We sincerely appreciate your careful evaluation of our design premise. We would like to clarify that CoEKS is purposefully designed for the practically important yet challenging setting of cross-task VRPs, where tasks differ by their active constraint sets. Within this scope, leveraging explicit constraint information is not a limitation but a principled and technically novel architectural choice that enables strong generalization.
>
> **1. Core Advantage Within Cross-task VRPs: A Strong Inductive Bias Aligned with the Problem Structure**
>
> CoEKS is not intended as a universal architecture for all problems. It is deliberately crafted for the practical **cross-task VRPs**, where tasks are defined by **different combinations of well-established modular constraints**. In this setting:
>
> * **Intrinsic VRP structure:** VRPs are compositional by nature, built from a set of standard constraints such as capacity and time windows. Exploiting this structural decomposition is not only common in combinatorial optimization (CO) but also known to be **highly effective**.
> * **CoEKS’s architectural insight:** CoEKS leverages this domain structure through constraint-specific experts, allowing the model to selectively combine the relevant modules for each task. This provides a **clean and principled mechanism** for capturing constraint compositionality—precisely the aspect that existing neural VRP solvers struggle to utilize.
>
> Thus, the constraint-aware design is a necessary and well-founded inductive bias for cross-task VRPs, not a drawback.
>
> **2. General Applicability to Broader CO Problems**
>
> While CoEKS is intentionally tailored for cross-task VRPs, its modular architecture naturally extends to other CO problems that share a similar structure—namely, those defined by distinct and combinable constraints. For instance, Job Shop Scheduling (JSP) involves multiple interpretable constraints (e.g., priority, machine conflicts), and CoEKS’s mechanism of dynamically activating relevant experts can be readily applied.

---

> ### Author Response · Authors · 2025-11-21
> **Response to Reviewer qhjy [2/6]**
>
> ## To Weakness 1:
>
> Thank you very much for the reviewer’s insightful comments. The core of this concern is the perception that our method replaces a learned gating mechanism with a handcrafted one, leading to only incremental innovation. We would like to clarify that CoEKS is **not a handcrafted gating of MoE experts**, but an **architectural redesign enabled by the intrinsic compositional structure of VRP constraints**, which fundamentally changes how task knowledge and constraint knowledge are represented, shared, and recombined. This structural prior—absent in generic MoE-based multi-task approaches—directly addresses the key bottleneck that limits OOD generalization in existing models.
>
> **1. Key Limitation of MoE and MoE with handcrafted gating**
>
> * Previous node-level MoE models (ReLD-MoEL [1], MVMoE [2]) treat expert routing as a generic, data-driven "black box" problem. This often leads to premature specialization at the node level, where experts fail to develop a coherent, task-level understanding of individual constraints. Consequently, they struggle to compose knowledge for unseen task combinations (OOD generalization).
> * MoE models (ReLD-MoEL, MVMoE) with handcrafted gating do not resolve this issue, because there is no explicit prior knowledge to guide which node embedding should be handed to which expert. Such a manual assignment would be arbitrary and is likely to perform worse than a learned gate that can find some data-driven pattern.
>
> **2. Methodological Innovations of CoEKS Beyond MoE with Handcrafted Gating**
>
> * **Problem-specific architectural design:** Leveraging the **unique prior structure of multi-task VRPs unlike general multi-task problems**—where each VRP task is defined by a set of constraints (e.g., capacity, time window)—CoEKS introduces a Combination-of-Experts (CoE) architecture in which each expert corresponds to a specific constraint. Unlike general multi-task problems, which lack this constraint-compositional structure, existing MoE designs operate at the node level (ReLD-MoEL, MVMoE) or at the task level (by routing tasks to experts). The proposed CoE, therefore, cannot be achieved through a standard MoE with handcrafted gating. It is a unique architectural design tailored to the combinatorial nature of VRP constraints.
> * **Complementary components integrating specialized and general knowledge:** Constraint-specialized experts serve as modular knowledge units aligned with the prior structure, while multi-view knowledge sharing captures cross-constraint general knowledge through mutual distillation and shared transformation layers. These two components work synergistically to enhance both specialization and generalization.
> * **Plug-in Scalability**: CoEKS enables efficient scaling to new constraints by simply adding and fine-tuning new experts without modifying the existing model (as demonstrated in Tables 3 and 7). This plug-and-play capability makes the approach well-suited for continual learning and deployment in dynamic environments.
>
> **3. Clarification on Gating Logic**
>
> The reviewer’s concern that complex gating logic is shifted to the user reflects a misunderstanding. The gating mechanism in CoEKS is **not determined by the user** for each instance. It is a **direct mapping from the set of active constraints**, which is an explicit and standard part of VRP problem input. CoEKS does not ask the user to decide gating; it simply exposes the problem’s known compositional structure to the model, in a principled and deterministic manner.
>
> In summary, CoEKS is not a handcrafted adaptation of MoE but an **architectural rethinking driven by domain-specific structural priors**. By explicitly modeling constraints as modular experts and equipping the system with a complementary knowledge sharing mechanism, CoEKS addresses the root cause of poor OOD generalization in existing neural methods. CoEKS enables scalable, compositional generalization that neither standard MoE nor handcrafted MoE is capable of achieving.

---

> ### Author Response · Authors · 2025-11-21
> **Response to Reviewer qhjy [3/6]**
>
> ## To Weakness 2:
> Thank you for the reviewer’s insightful observation regarding the scalability experiment. We would like to clarify that the observed performance on "VRPMB-only" fine-tuning is primarily an artifact of model convergence instability, rather than a structural weakness of CoEKS.
>
> **1. The VRPMB-only fine-tuning setting leads to unstable convergence, making comparisons unreliable**
>
> When fine-tuning only on VRPMB, we observed that models failed to converge stably on tasks involving both MB and TW constraints (e.g., VRPMBTW, OVRPMBTW), as shown in Figure 6. This makes the MB evaluation intrinsically unreliable—not only for CoEKS, but for all baselines including RouteFinder [3] and ReLD-MoEL, which also fail to converge in this setting.
> Among the few tasks that did converge in this unstable regime, CoEKS performed overall on par with ReLD-MoEL, with small differences in both directions:
> * CoEKS performed better on VRPMB and VRPMBL.
> * ReLD-MoEL showed an advantage on OVRPMB and OVRPMBL.
>
> However, because this setting lacks stable convergence, these fluctuations are not a scientifically meaningful measure of model capability.
>
> **2. Convergence Issue Caused by Task Difficulty**
>
> The core challenge lies in the difficulty of learning to generalize from a single, isolated task (VRPMB) to tasks involving complex, unseen constraint combinations. This type of distribution shift is inherently difficult in NP-hard combinatorial optimization problems, where constraint interactions introduce non-trivial solution space changes. VRPMB alone does not expose the model to sufficient structural diversity to prepare it for more complex, multi-constraint tasks.
>
> **3. Stability and Superiority Upon Convergence**
>
> When we introduced the minimal structural diversity by including the Time Window (TW) constraint—fine-tuning on VRPMBTW—the key result is that CoEKS achieved stable convergence on all target tasks. However, the existing methods still exhibit convergence failures or poor generalization capabilities on certain tasks.
>
> Furthermore, our robust learning ability is explicitly confirmed when fine-tuned on all MB-related tasks, where CoEKS significantly surpasses ReLD-MoEL on all MB-related tasks (see Table 7). This also powerfully confirms that CoEKS can effectively learn MB-related patterns.
>
> In conclusion, the fluctuated performance observed in the MB-only setting is a result of lack of convergence, not a model weakness. When evaluated under a convergent and meaningful setting (MB+TW or all MB tasks), CoEKS consistently surpasses ReLD-MoEL and RF-TE on all MB-related tasks. This validates the scalability and reliability of the plug-in expert design for adapting to unseen constraints.

---

> ### Author Response · Authors · 2025-11-21
> **Response to Reviewer qhjy [4/6]**
>
> ## To Weakness 3:
> We appreciate the reviewer's observation, but we believe the MB experiment was not a simple test case and did validate our scalability to a certain extent.
>
> **1. Usefulness of the MB experiment**
>
> Although the MB constraint is somewhat similar in form to the B constraint, it presents a new, non-trivial challenge to the model. The MB constraint allows mixing pickup and delivery services, which fundamentally alters the demand structure and sequencing requirements, differentiating it from the purely B constraint. Consequently, the ability to adapt to the MB constraint validates the method's scalability to a certain extent.
>
> **2. New Complementary Experiment: Scalability to Multi-Depot (MD)**
>
> Nevertheless, we fully agree that supporting scalability with an even more structurally distinct constraint would further strengthen our claim. Therefore, in response to the reviewer's valuable suggestion, we introduced the Multi-Depot (MD) constraint for an additional, stricter experiment.
>
> To thoroughly assess scalability, we added 24 new VRP tasks at the 50-node scale:
> * 16 tasks containing only the new MD constraint, and
> * 8 tasks combining MD with MB.
>
> The experimental results are provided below. Across all newly included tasks, CoEKS continues to achieve the **best performance**, demonstrating strong scalability to unseen constraints.
>
> We have included these additional results in the appendix F.3 of the revised manuscript. This new evidence decisively supports our claim that CoEKS is robustly scalable to new, structurally diverse constraints.
>
> **Fine-tuning performance on VRPs with MD:**
>
>  Method \ Gap↓ | MDCVRP | MDOVRP | MDOVRPB | MDOVRPBL | MDOVRPBLTW | MDOVRPBTW | MDOVRPL | MDOVRPLTW
> ---|---|---|---|---|---|---|---|---
>  ReLD-MoE | 15.235% | 13.525% | 17.125% | 16.933% | 11.405% | 11.278% | 13.333% | 10.758%
>  CoEKS | **8.928%** | **6.491%** | **7.599%** | **7.588%** | **5.324%** | **5.318%** | **6.543%** | **5.038%**
>
>  ---
>
>
>  Method \ Gap↓  | MDOVRPTW | MDVRPB | MDVRPBL | MDVRPBLTW | MDVRPBTW | MDVRPL | MDVRPLTW | MDVRPTW
> ---|---|---|---|---|---|---|---|---
>  ReLD-MoE | 10.622% | 20.408% | 20.572% | 15.326% | 15.340% | 15.315% | 14.502% | 13.966%
>  CoEKS | **4.978%** | **12.093%** | **12.487%** | **9.532%** | **9.324%** | **9.109%** | **8.970%** | **8.642%**
>
> ---
>
> **Fine-tuning performance on VRPs with MD and MB:**
> Method \ Gap↓ | MDOVRPMB | MDOVRPMBL | MDOVRPMBLTW | MDOVRPMBTW | MDVRPMB | MDVRPMBL | MDVRPMBLTW | MDVRPMBTW
> ---|---|---|---|---|---|---|---|---
>  ReLD-MoE | 17.790% | 17.543% | 11.430% | 11.206% | 23.673% | 22.101% | 15.081% | 14.417%
>   CoEKS | **6.959%** | **7.061%** | **5.250%** | **5.130%**| **12.128%** | **12.032%** | **9.122%** | **8.804%**

---

> ### Author Response · Authors · 2025-11-21
> **Response to Reviewer qhjy [5/6]**
>
> ## To Weakness 4:
> We appreciate the reviewer’s comment. We agree that adding more constraints naturally introduces more experts, but we respectfully argue that this is a **necessary and beneficial trade-off**, and CoEKS provides high parameter efficiency and substantial scalability advantages that go beyond its parameter count. Below we present a structured response.
>
> **1. Justified Trade-Off Between Parameter Count and Performance**
>
> As originally discussed in our submission, the increase in expert count is necessary to capture complex constraint knowledge. Indeed, further experiments demonstrate that the performance gains significantly outweigh the parameter cost:
> * **For Simpler Tasks (2 Constraints):** When the task complexity is lower, both CoEKS and ReLD-MoEL (the SOTA MoE model) activate only 2 experts, while CoEKS still achieves a 7% average relative improvement.
> * **For More Complex Tasks (3 Constraints):** When the task complexity increases, CoEKS automatically adapts by activating **one additional expert** (adding only **0.79 million** parameters). This minor increment in active parameter count yields a disproportionately larger performance boost, resulting in a substantial **13% average relative improvement** over ReLD-MoEL.
>
> These results show that the parameter overhead of the dynamically activated expert is **highly cost-effective**. By utilizing more experts only when needed for complex tasks, CoEKS ensures that the marginal increase in computational cost is overwhelmingly justified by the **significant jump in solution quality**, confirming the necessity of our flexible expert system for tackling increasing task complexity.
>
> **2. Linear expert growth enables exponential scalability in task coverage**
>
> A key advantage of CoEKS is that the number of experts grows linearly with the number of basic constraints, but the number of constraint combinations (tasks) grows exponentially.
> For example:
> * With **4 basic constraints** → **4 experts** → can cover up to **$2^{(4−1)} = 8$** composite tasks.
> * With **5 basic constraints** → **5 experts** → can cover up to **$2^{(5−1)} = 16$** tasks.
>
> Thus, adding just one more expert enables the model to scale to twice as many possible constraint combinations. This demonstrates that the **linear parameter growth yields exponential coverage** of the task space, making it a highly efficient scaling strategy.
>
> **3. High Runtime Efficiency Through Sparse Activation**
>
> CoEKS does not activate all experts simultaneously. Instead, it uses sparse constraint-driven activation, meaning the model only engages experts associated with the active constraints of a given task.
>
> In the ID task set, the CoEKS model activates an average of 2 out of 5 experts, utilizing just **40% of the expert parameters per sample**. Compared to ReLD-MoEL-E5 (Appendix F.1), which aligns with CoEKS by adding one expert and also activates two experts on average, CoEKS achieves a **15.4%** relative performance improvement. Even on OOD datasets with more complex constraints, CoEKS activates only **74% of the expert parameters** (3.7 experts on average), yielding a **22.1%** performance gain over ReLD-MoEL-E5.
>
> **4. Superior Parameter Efficiency Against Dense Equivalents**
>
> We directly verified CoEKS's parameter efficiency advantage by constructing a baseline model with strictly matched parameters. Specifically, we replace the CoE architecture in CoEKS with a dense multi-layer perceptron (MLP). The hidden layer dimension of this dense MLP is set to 2560, making its total parameter count (4.6 million) equal to that of CoEKS. The results are shown in the following table. In all tasks, the performance of the dense baseline is significantly inferior to that of CoEKS. This demonstrates that CoEKS achieves **superior performance** with the **same or fewer parameters**, highlighting its parameter efficiency advantage over dense models.
>
> **In-Distribution ($n=50$):**
>
> | Method \ Gap↓        | CVRP    | OVRP    | VRPB    | VRPL    | VRPTW   | OVRPTW  | OVRPL   | (ID Avg.)  |
> |--|-|-|-|-|-|-|-|-|
>  Dense baseline | 0.917% | 2.395% | 2.634% | 1.151% | 2.408% | 1.608% | 2.398% | 1.9303%
>  CoEKS                  | **0.8914%** | **2.1376%** | **2.4968%** | **1.1523%** | **2.0500%** | **1.3928%** | **2.1352%** | **1.7509%**
>
> **Out-of-Distribution ($n=50$):**
>
> | Method \ Gap↓        | OVRPB   | VRPBL   | VRPBTW  | VRPLTW  | OVRPBL  | OVRPBTW | OVRPLTW | VRPBLTW | OVRPBLTW | (OOD Avg.)  |
> |-|-|-|-|-|-|-|-|-|-|-|
> Dense baseline | 5.868% | 5.248% | 6.840% | 4.133% | 4.377% | 3.420% | 1.896% | 7.369% | 4.477% | 4.8474%
> CoEKS                  | **4.9127%** | **4.3871%** | **4.7473%**  | **3.2098%** | **2.9489%** | **2.5664%** | **1.6026%** | **3.7140%** | **2.7969%**  | **3.4317%**
>
>
> These points collectively show that the constraint-to-expert mapping is not inefficient, but rather a highly effective and principled design for cross-task VRP generalization.

---

> ### Author Response · Authors · 2025-11-21
> **Response to Reviewer qhjy [6/6]**
>
> ## To Question 1:
>
> Thank you for the insightful question. We fully agree that zero-shot generalization to entirely new constraints, such as Mixed Backhauls (MB) or Multi-Depot (MD), is both important and highly desirable. However, this setting is fundamentally different from distribution shifts in problem size or node patterns. Those shifts still preserve the underlying constraint structure, whereas **introducing a new constraint changes the feasible region, alters the problem semantics, and requires reasoning over rules the model has never encountered**. Consequently, true zero-shot generalization to unseen constraints remains extremely challenging and is beyond the capability of existing neural VRP solvers.
>
> Despite these difficulties, since the reviewer expressed interest, we conducted an additional experiment comparing CoEKS with the state-of-the-art ReLD-MoEL under this strict zero-shot setting—no new expert is added, and no fine-tuning is performed. The model can only activate experts for previously known constraints. As shown in the table below, the overall performance drops for both methods due to the harsh setting, but **CoEKS still achieves a noticeably smaller gap than ReLD-MoEL**, indicating that our constraint-specialized architecture confers stronger inherent robustness even with zero adaptation.
>
> Method \ Gap↓($n=50$)  | MDVRPMB | MDOVRPMB | MDVRPMBL | MDVRPMBTW | MDOVRPMBL | MDOVRPMBTW | MDVRPMBLTW | MDOVRPMBLTW
> ---|---|---|---|---|---|---|---|---
>  ReLD-MoEL | 483.256% | 318.240% | 106.360% | 251.104% | 116.280% | 172.352% | 94.440% | 91.368%
>  CoEKS | 356.216% | 234.045% | 71.848% | 188.496% | 78.554% | 130.795% | 81.088% | 89.796%
>
> That said, the gaps for all methods in this scenario remain large, reaffirming that current approaches are still far from addressing truly novel constraint types in a zero-shot manner. This suggests that the practical significance of such comparisons is limited at the current stage. Nevertheless, we agree that this is a **highly important and impactful research direction** for the community, and we view advancing zero-shot reasoning over structurally new constraints as a central and exciting avenue for future work.
>
> ## To Question 2:
> Thank you for the insightful question. To further assess scalability of CoEKS on unseen constraints, we additionally conducted fine-tuning experiments on **100-node** VRP tasks involving the unseen MB and MD constraints. As shown below, we consistently maintained our best performance.
>
> These findings indicate that the scalability insight observed at $n=50$ generalizes to larger problem sizes. CoEKS remains effective and robust when adapting to new constraints across different VRP scales, further confirming its strength in cross-task generalization.
>
> Method \ Gap↓  |  MDOVRPMB | MDOVRPMBL | MDOVRPMBLTW | MDOVRPMBTW | MDVRPMB | MDVRPMBL | MDVRPMBLTW | MDVRPMBTW
> ---|---|---|---|---|---|---|---|-
>  ReLD-MoE | 37.904% | 37.253% | 15.355% | 15.418% | 35.031% | 34.981% | 18.735% | 17.932%
>  CoEKS | **8.791%** | **8.859%** | **7.181%** | **7.058%** | **13.581%** | **13.625%** | **10.761%** | **10.553%**
>
> [1] Rethinking light decoder-based solvers for vehicle routing problems. ICLR, 2025.
> [2] Mvmoe: multi-task vehicle routing solver with mixture-of-experts. ICML, 2024.
> [3] Routefinder: Towards foundation models for vehicle routing problems. TMLR, 2025.

---

> > ### Comment · Reviewer_qhjy · 2025-11-26
> >
> > I thank the authors for their detailed response and the additional Multi-Depot (MD) experiments. However, several concerns regarding the core methodology and experimental results remain.
> >
> > 1. Echoing Reviewer dA3D, I question the method's scalability when applied to real-world scenarios. One-to-one constraint-to-expert mapping may be ill-suited for real-world scenarios where constraints are often open-ended and may not be clearly separable (i.e., ill-defined).  Furthermore, the accumulation of numerous expert modules for complex problems poses potential risks to memory capacity. In addition, the reliance on auxiliary constraints (e.g., adding TW to stabilize MB) to ensure fine-tuning convergence indicates potential sensitivity in the optimization process.
> >
> > 2. I acknowledge that zero-shot generalization is challenging for current multi-task solvers. However, evaluating zero-shot generalization is of significant positive value for practical model application, and it has already become a standard benchmark for applicability (e.g., in RF, CaDA). The excessively high gaps reported (>100%) diverge significantly from recent literature, suggesting the architecture struggles to leverage general knowledge without explicit expert training.
> >
> > 3. I am concerned about the fairness of the zero-shot comparison. You report a >400% gap for the ReLD-MoEL baseline on 50-node instances, whereas standard pre-trained models I have tried typically yield gaps <50% even on harder 100-node instances at the same 8 cases. This drastic discrepancy suggests that the baseline may have been evaluated under disadvantageous settings. If you retrained the baseline to match a specific setting of your method and this resulted in such a drastic performance drop, the comparison is unfair and lacks rigorous consideration.
> >
> > Although the proposed method shows promising results in specific ID scenarios, considering the reliance on deterministic routing and the questionable fairness of the baseline comparisons, I believe the limitations currently outweigh the contributions. I will maintain my current score and hope the author provides further response.

---

> ### Author Response · Authors · 2025-11-27
>
> We sincerely thank the reviewer for the rigorous scrutiny, particularly regarding the fairness of the zero-shot comparison. Below, we address your specific concerns about real-world scalability, optimization sensitivity, and the fairness of our baseline comparisons.
> ## To Concern 1:
> **1. Scalability and Applicability to Real-World Scenarios**
> We appreciate your concerns regarding the practical deployment of our method. We address your three specific points below:
> * **Regarding the concern that "One-to-one constraint-to-expert mapping may be ill-suited for real-world scenarios where constraints are often open-ended... (i.e., ill-defined)":**
>     * **Scalability as a Design Advantage: The Need for Expandable Capacity in Complex VRPs.** Real-world VRPs increase in structural complexity as more heterogeneous constraints are introduced. In such settings, allowing the model to scale its capacity is a rational and worth trade-off as discussed in our previous response to Reviewer zfcw's Weakness 2. CoEKS's plug-in expert architecture provides this flexibility, whereas existing fixed-capacity methods such as ReLD-MoEL or CaDA cannot easily expand their representational power when encountering substantially more complex or diverse constraints. This fixed-capacity design of existing methods should be regarded as a limitation rather than an advantage.
>     * **A Resource-Constrained Fallback Option: Shared Expert Configuration.** We also acknowledge the concern that extremely many constraint or strict memory budgets may make per-constraint experts impractical. CoEKS is not strictly tied to unlimited expansion; when resources are limited, a single expert can be assigned to handle multiple constraints or hard-to-decompose constraints. To verify this, we conducted an experiment on new MD and MB tasks by introducing only one expert ($E_{DB}$)  shared across all these new constraints. As shown in the table below, CoEKS (FT-$E_{DB}$) still significantly outperforms the ReLD-MoEL baseline, demonstrating that the architecture remains effective even under fixed-capacity settings like existing methods.
>
> Method \ Gap↓($n=50$)| MDOVRPMB | MDOVRPMBL | MDOVRPMBLTW | MDOVRPMBTW | MDVRPMB | MDVRPMBL | MDVRPMBLTW | MDVRPMBTW
> ---|---|---|---|---|---|---|---|---
>  ReLD-MoEL | 17.790% | 17.543% | 11.430% | 11.206% | 23.673% | 22.101% | 15.081% | 14.417%
>  CoEKS(FT-$E_{DB}$) | 8.813% | 8.857% | 5.784% | 5.654% | 14.659% | 14.700% | 9.840% | 9.580%
>  CoEKS | **6.959%** | **7.061%** | **5.250%** | **5.130%**| **12.128%** | **12.032%** | **9.122%** | **8.804%**
>
> Moreover, we note that sharing one expert across multiple or hard-to-decompose constraints inevitably introduces some performance trade-off, as shown in the table above. Future work may explore more efficient expert-sharing mechanisms to mitigate this (as discussed in Conclusion in our paper), though such developments are beyond the scope of the present study.
>
>
>
> **2. Regarding the concern that "the accumulation of numerous expert modules... poses potential risks to memory capacity":**
> We provide a quantitative analysis to demonstrate that this risk is negligible in practice:
>   * **Storage Memory:** Each expert adds only ~0.79M parameters, as mentioned in our previous response to weakness 4. Surveys on Rich Vehicle Routing Problems [1], categorize real-world operational requirements into a finite taxonomy of attributes, typically numbering fewer than 20 distinct constraints. Even if we adopt this comprehensive upper bound of 20 distinct basic constraints, the total parameters for all experts would be roughly 15.7 million. In terms of GPU memory (FP32), this amounts to roughly 64 MB. This fits easily within the memory of even entry-level consumer GPUs (which typically have 8GB+), let alone industrial hardware.
>   * **Training Memory (Imperceptible):**  In practical training, GPU memory is dominated by storing activation maps (which scales with batch size), not by model parameters. For example, training CoEKS on $N=50$ instances with a batch size of 256 consumes approximately 6300 MB of GPU memory. The ~3 MB memory increase from adding a new expert (0.79M params) is virtually imperceptible in this context.
>
>
> **3. Regarding the concern that "reliance on auxiliary constraints... indicates potential sensitivity in the optimization process":**
> You correctly noted that we used TW tasks to ensure convergence. However, as mentioned in our previous response to Weakness 2, this reflects the inherent difficulty of the task distribution shift (universal instability), not a specific weakness of CoEKS. As shown in our paper, all methods (including ReLD-MoEL and RF-TE) failed to converge in the "VRPMB-only" setting. However, once the training stabilized, CoEKS consistently outperformed the baselines, demonstrating robustness in viable training regimes.
>
> [1] Rich vehicle routing problem: Survey. ACM Computing Surveys, 2014.

---

> ### Author Response · Authors · 2025-11-27
>
> ## To Concern 2 and 3:
>
> We are deeply grateful for your skepticism regarding the gap reported for the ReLD-MoEL baseline on MD tasks. We agree that such a discrepancy implies a misalignment in the comparison setting.
>
> In our original submission, we primarily compared CoEKS against ReLD-MoEL as it is the current SOTA for cross-task OOD generalization (as discussed in the Related Works section). Consequently, we initially overlooked the zero-shot results in the RouteFinder paper, which led us to miss the inconsistency. To address your concern about the divergence from literature, we have now included RouteFinder in our comparison to verify the correctness of our rectified setting.
>
> **Explanation of the Discrepancy & Evaluation Correction:** Based on your feedback, we conducted a thorough review and found that the implementation of the MD constraint in our evaluation script was inconsistent with the standard protocols used in existing literature. This discrepancy led to incorrect and excessively high cost values for the baseline model (and our initial results). We have now rectified the evaluation protocol to align strictly with recent work.
>
> **Corrected Zero-Shot Results — CoEKS Consistently Outperforms:** We first present the corrected zero-shot performance below. Under this fair setting, the gaps for the baselines (ReLD-MoEL and RF) remains within a reasonable range, aligning with your experience. Crucially, CoEKS consistently outperforms both the SOTA ReLD-MoEL and RF in this zero-shot setting.
>
> **Commitment to Reproducibility (Code & Checkpoints):** To further ensure the transparency and credibility of these results, we have made our full codebase and the trained model checkpoints publicly available in our anonymous repository https://anonymous.4open.science/r/CoEKS-B0D9/
>
> **Zero-shot generalization on VRPs with MD:**
> Task | mdcvrp | mdovrp | mdovrpb | mdovrpbl | mdovrpbltw | mdovrpbtw | mdovrpl | mdovrpltw
> -|-|-|-|-|-|-|-|-
> RF-EAL | **37.514%** | 29.887% | 37.903% | 40.128% | 38.314% | 45.111% | 29.934% | 34.305%
> ReLD-MoEL | 39.359% | 20.266% | 33.504% | 32.543% | 25.134% | 28.718% | 20.108% | 25.210%
> CoEKS | 38.174% | **15.678%** | **22.690%** | **21.883%** | **21.920%** | **22.901%** | **15.616%** | **20.949%**
>
> Task | mdovrptw | mdvrpb | mdvrpbl | mdvrpbltw | mdvrpbtw | mdvrpl | mdvrpltw | mdvrptw
>  -|-|-|-|-|-|-|-|-
>  RF-EAL | 40.836% | 45.689% | 59.955% | 42.723% | 46.764% | 43.181% | 39.398% | 42.167%
> ReLD-MoEL | 28.634% | 48.687% | 47.975% | 30.311% | 39.094% | 39.211% | 30.267% | 36.833%
> CoEKS | **21.730%** | **38.612%** | **38.188%** | **27.884%** | **31.094%** | **36.288%** | **26.986%** | **28.731%**
>
> **Zero-shot generalization on VRPs with MD and MB:**
> Task  | MDOVRPMB | MDOVRPMBL | MDOVRPMBLTW | MDOVRPMBTW | MDVRPMB | MDVRPMBL | MDVRPMBLTW | MDVRPMBTW
> -|-|-|-|-|-|-|-|-
> RF-EAL | 44.410% | 44.523% | 38.577% | 38.608% | 58.138% | 57.110% | 41.310% | 41.821%
> ReLD-MoEL | 47.741% | 45.581% | 26.322% | 29.812% | 64.400% | 60.381% | 31.776% | 38.646%
> CoEKS | **35.186%** | **33.701%** | **24.938%** | **25.365%** | **53.428%** | **50.735%** | **30.900%** | **32.874%**
>
> **Updated Fine-Tuning Results — CoEKS Consistently Outperforms:** Since the correction of the MD constraint evaluation affects the calculation of costs, we have naturally updated the fine-tuning results as well. As shown below, CoEKS maintains its superiority after fine-tuning.
>
> **Fine-tuning on VRPs with MD:**
> Task | mdcvrp | mdovrp | mdovrpb | mdovrpbl | mdovrpbltw | mdovrpbtw | mdovrpl | mdovrpltw
> -|-|-|-|-|-|-|-|-
>  RF-EAL | 15.251% | 14.250% | 18.996% | 19.479% | 19.841% | 19.315% | 15.421% | 17.977%
>  ReLD-MoEL | 15.235% | 13.525% | 17.125% | 16.933% | 11.405% | 11.278% | 13.333% | 10.758%
>  CoEKS | **8.928%** | **6.491%** | **7.599%** | **7.588%** | **5.324%** | **5.318%** | **6.543%** | **5.038%**
>
>  Task | mdovrptw | mdvrpb | mdvrpbl | mdvrpbltw | mdvrpbtw | mdvrpl | mdvrpltw | mdvrptw
>  -|-|-|-|-|-|-|-|-
>  RF-EAL | 17.300% | 21.943% | 23.149% | 25.549% | 24.098% | 15.537% | 21.952% | 20.244%
>  ReLD-MoEL | 10.622% | 20.408% | 20.572% | 15.326% | 15.340% | 15.315% | 14.502% | 13.966%
>   CoEKS | **4.978%** | **12.093%** | **12.487%** | **9.532%** | **9.324%** | **9.109%** | **8.970%** | **8.642%**
>
> **Fine-tuning on VRPs with MD and MB:**
> Task | MDOVRPMB | MDOVRPMBL | MDOVRPMBLTW | MDOVRPMBTW | MDVRPMB | MDVRPMBL | MDVRPMBLTW | MDVRPMBTW
> -|-|-|-|-|-|-|-|-
>  RF-EAL | 16.536% | 18.378% | 18.301% | 17.592% | 33.316% | 21.222% | 21.698% | 19.965%
>  ReLD-MoEL | 17.790% | 17.543% | 11.430% | 11.206% | 23.673% | 22.101% | 15.081% | 14.417%
>  CoEKS | **6.959%** | **7.061%** | **5.250%** | **5.130%**| **12.128%** | **12.032%** | **9.122%** | **8.804%**
>
>
> We hope these clarifications regarding model application and memory quantization alleviate your concerns about scalability. We once again thank you for helping us ensure the rigor of our experiments. The corrected results reaffirm the effectiveness of CoEKS under fair conditions.

---

### Official Review · Reviewer_dA3D · 2025-10-30

**Soundness:** 3
**Presentation:** 3
**Contribution:** 2
**Rating:** 6
**Confidence:** 4

**Summary:**

A neural "cross-task" vehicle routing problem (VRP) solver is presented. The main insight is that model components are introduced for each constraint type, e.g., time windows or backhauls. These are called experts, and the authors define a strategy for combining their outputs to deal with problems with combinations of multiple outputs. The approach is compared against other multi-task approaches for the VRP on a part of a standard benchmark of VRP tasks. It outperforms the current state-of-the-art approaches for this multi-task setting.

**Strengths:**

1. The idea is generally simple and well-founded, I think it basically makes sense even if it has scaling issues in the number of constraints supported.
2. The performance is good, but I do have a very important question regarding the benchmark (see below)
3. The paper is generally easy to understand.

**Weaknesses:**

1. The experimental section is written very short, and important details are pushed to the appendix multiple times. This is unfortunate, as the paper wastes space on page 6 providing what I can only describe as a marketing rehash of the introduction. That entire paragraph ought to go towards describing the results. Perhaps the authors can address this.
2. The results are only incrementally better than previous work; this paper is a small step forward, but certainly not a breakthrough.
3. I do not think the idea is one that will scale in general; the concept of adding a new expert for each type of constraint will eventually just overload the model with parameters that are likely unnecessary. The approach reminds me of the Poppy approach from Grinsztajn et al. (2023) in NeurIPS in which multiple decoders are used. The extra decoders in Poppy were shown in the Polynet paper (Hottung et al. (2025) in ICLR) to not be necessary, a vector input can be used instead. The same likely holds here, so I feel like this paper is not quite finished.

Some other comments that the authors should consider addressing:
1. The contributions in the introduction are vague, it would be better to be more precise about what is offered in this work.
2. The first paragraph of the related work has some grammar issues, let an LLM fix it.
3. The research questions offered in the experimental section are well done.
4. I would like to see more about the ablation and sensitivity analysis in the main text.

**Questions:**

1. RouteFinder has 48 problem variants, this paper only has 24. What happened to the other variants and why are they not used?

---

> ### Author Response · Authors · 2025-11-21
> **Response to Reviewer dA3D [1/3]**
>
> We sincerely appreciate the reviewer's recognition of the soundness and intuitiveness of our idea, the good performance, and the clarity of our paper. Below, we address each concern point by point. All revisions and additions made to the manuscript have been marked in blue to aid in your review.
> ## To Weakness 1:
> We appreciate the reviewer’s constructive suggestion and understand the concern regarding the brevity of the experimental section. In response, we have removed Section 4.4. In addition, we have expanded the experimental analysis in the main body of the paper, specifically focusing on the ablation and sensitivity analysis (e.g., the application of mutual distillation). This change allows us to provide a more comprehensive and focused discussion of the experimental results. We believe this adjustment improves the clarity and depth of the experimental section.
> ## To Weakness 2:
> Thank you for the comment. While we understand the reviewer’s impression, we would like to clarify that the contributions of this work extend substantially beyond incremental improvements. Our advances operate at two complementary levels—empirical impact and methodological contribution—and together form a meaningful step forward for cross-task neural combinatorial optimization.
>
> **Significant Improvement of Performance**
>
> In combinatorial optimization problems like VRPs, where prior methods have already achieved strong baselines through extensive optimization, subsequent improvements become increasingly difficult due to diminishing returns. Our contributions are validated through a large-scale and comprehensive experimental study across 5 constraints and their combinations, covering both ID and OOD settings. To provide context, we compare against recent state-of-the-art methods published in top conferences. For instance:
> * MVMoE [1] achieved relative improvements of +8.4% (ID) / +4.8% (OOD) over its baselines,
> * ReLD-MoEL [2] reported +21.6% (ID) / +15.1% (OOD),
> * CaDA [3] showed +13.1% overall,
> * RouteFinder [4] gained +9.8% overall.
>
> Under the same 50-node setting, CoEKS achieves **+7.9% (ID)** and **+18.3% (OOD)** over these methods. Notably, the +18.3% OOD gain is the **largest reported to date** among recent cross-task VRP methods, marking a substantial breakthrough in OOD generalization despite the inherent challenges of further advancing well-optimized CO problems. This demonstrates that our empirical improvement is significant rather than relatively small.
>
> **Methodological Contribution:**
>
> * **Problem-specific architectural design:**
> Our work begins with a core structural observation: many VRP variants can be expressed as modular compositions of constraints (e.g., capacity, time windows), an aspect that existing neural cross-task methods largely overlook. Leveraging this prior, we design a Combination-of-Experts (CoE) architecture where each constraint is assigned to a dedicated expert and recombined dynamically. This structural perspective is not limited to VRPs but extends to other combinatorial optimization problems with similar forms of constraint modularity, such as job shop scheduling (e.g., decomposing by machine precedence or operation constraints).
> * **Complementary components integrating specialized and general knowledge:** Constraint-specialized experts serve as modular knowledge units aligned with the prior structure, while multi-view knowledge sharing captures cross-constraint general knowledge through mutual distillation and shared transformation layers. These two components work synergistically to enhance both specialization and generalization.
> * **Plug-in Scalability**: CoEKS enables efficient scaling to new constraints by simply adding and fine-tuning new experts without modifying the existing model (as demonstrated in Tables 3 and 7). This plug-and-play capability makes the approach well-suited for continual learning and deployment in dynamic environments.
>
> Most importantly, the innovation of CoEKS lies primarily in its **contribution to a comprehensive methodology**. We identify and exploit the modular constraint structure of VRPs—an overlooked but fundamental property—and operationalize it through a principled combination of constraint-specialized experts and controlled knowledge sharing. This insight and methodology extend naturally to other combinatorial optimization domains that share compositional constraint structures, such as Job Shop Scheduling Problem involves a specific subset of active constraints (e.g., priority + machine conflicts), making CoEKS a **general and impactful design paradigm** rather than an incremental variant. We hope this clarifies that our contribution is conceptual and broadly applicable, and thus substantially more than a minor improvement in performance.

---

> ### Author Response · Authors · 2025-11-21
> **Response to Reviewer dA3D [2/3]**
>
> ## To Weakness 3:
> Thank you for your comments. We believe CoEKS offers efficient practical scalability and is fundamentally distinct from the Poppy and PolyNet approach.
>
> **1. Linear expert growth enables exponential scalability in task coverage**
>
> A key advantage of CoEKS is that the number of experts grows linearly with the number of basic constraints, but the number of constraint combinations (tasks) grows exponentially.
> For example:
> * With **4 basic constraints** → **4 experts** → can cover up to **$2^{(4−1)} = 8$** composite tasks.
> * With **5 basic constraints** → **5 experts** → can cover up to **$2^{(5−1)} = 16$** tasks.
>
> Thus, **adding just one more expert** enables the model to scale to **twice as many** possible constraint combinations. This demonstrates that **linear parameter growth yields exponential coverage**  of the task space, making it a highly efficient scaling strategy for cross-task VRPs.
>
> **2. Long-Term Strategy**
>
> The concern about unrestricted expert growth is a theoretical extreme. Future work can mitigate this by implementing strategies to fix the total number of experts (allowing each to handle a cluster of related constraints), which would ensure bounded complexity while retaining modularity
>
> **3. Difference from PolyNet and Poppy**
>
> Poppy and PolyNet respectively employ multi-decoders and vector inputs to focus on learning diverse solution strategies in single-task combinatorial optimization problems, thereby enhancing solution quality through diversity. In contrast, CoEKS's experts represent specialized knowledge for specific constraints and further enhance their cross-task generalization capabilities through a knowledge sharing mechanism.
>
> **4. Architectural Superiority Against CaDA that Uses a Design Concept Similar to PolyNet**
>
> A similar approach is used in CaDA, where constraint information is encoded through vector inputs rather than expert modules. However, CoEKS outperforms CaDA across all ID and OOD benchmarks (Tables 1–2). This indicates that CoEKS offers a more powerful and principled architectural design for cross-task VRP generalization than vector-based formulations.
>
> ## To "Some other comments that the authors should consider addressing":
> We thank the reviewer for the thoughtful and helpful feedback! We have made the following revisions and uploaded a new manuscript:
> 1. For the first point, we refined the contribution part in the introduction to more clearly explain what our work provides.
> 2. For the second point, we corrected the grammatical issues in the first paragraph of the related work section.
> 3. For the third point, we sincerely appreciate the reviewer’s positive feedback on the structure and clarity of the research questions. We aimed to present our experimental validation in a clear and logical manner, and we are delighted that it was well-received.
> 4. For the fourth point, we now present more ablation and sensitivity analyses in the main text (page 9).

---

> ### Author Response · Authors · 2025-11-21
> **Response to Reviewer dA3D [3/3]**
>
> ## To Question:
> We sincerely thank the reviewer for raising this point regarding the completeness of our experimental setup relative to RouteFinder. Our initial focus on 24 variants was sufficient to achieve our primary goal: assessing CoEKS's adaptability to unseen constraint combinations and its performance when a new constraint arises.
>
> To address the reviewer’s concern, we have followed the setup in RouteFinder and added the remaining 24 variants under the 50-node setting, including:
> * 16 variants containing only the new MD constraint, and
> * 8 variants combining MD with MB
>
> The experimental results are provided below. Across all newly included tasks, CoEKS continues to achieve the best performance, demonstrating strong adaptability to unseen constraints. We have included these additional results in the appendix F.3 of the revised manuscript.
>
> **Fine-tuning performance on VRPs with MD:**
>
>  Method \ Gap↓ | MDCVRP | MDOVRP | MDOVRPB | MDOVRPBL | MDOVRPBLTW | MDOVRPBTW | MDOVRPL | MDOVRPLTW
> ---|---|---|---|---|---|---|---|---
>  RF-EAL | 15.251% | 14.250% | 18.996% | 19.479% | 19.841% | 19.315% | 15.421% | 17.977%
> CoEKS | **8.928%** | **6.491%** | **7.599%** | **7.588%** | **5.324%** | **5.318%** | **6.543%** | **5.038%**
>
>  Method \ Gap↓ | MDOVRPTW | MDVRPB | MDVRPBL | MDVRPBLTW | MDVRPBTW | MDVRPL | MDVRPLTW | MDVRPTW
> ---|---|---|---|---|---|---|---|---
>  RF-EAL | 17.300% | 21.943% | 23.149% | 25.549% | 24.098% | 15.537% | 21.952% | 20.244%
>  CoEKS | **4.978%** | **12.093%** | **12.487%** | **9.532%** | **9.324%** | **9.109%** | **8.970%** | **8.642%**
>
> ---
>
> **Fine-tuning performance on VRPs with MD and MB:**
> Method \ Gap↓ | MDOVRPMB | MDOVRPMBL | MDOVRPMBLTW | MDOVRPMBTW | MDVRPMB | MDVRPMBL | MDVRPMBLTW | MDVRPMBTW
> ---|---|---|---|---|---|---|---|---
>  RF-EAL | 16.536% | 18.378% | 18.301% | 17.592% | 33.316% | 21.222% | 21.698% | 19.965%
> CoEKS | **6.959%** | **7.061%** | **5.250%** | **5.130%**| **12.128%** | **12.032%** | **9.122%** | **8.804%**
>
> [1] Mvmoe: multi-task vehicle routing solver with mixture-of-experts. ICML, 2024.
> [2] Rethinking light decoder-based solvers for vehicle routing problems. ICLR, 2025.
> [3] CaDA: Cross-Problem Routing Solver with Constraint-Aware Dual-Attention. ICML, 2025.
> [4] Routefinder: Towards foundation models for vehicle routing problems. TMLR, 2025.

---

> > ### Comment · Reviewer_dA3D · 2025-11-25
> >
> > I thank the authors for their thoughtful discussion.
> >
> > Regarding weakness 2, I want to emphasize that my criticism was only regarding the experimental results themselves and not the contribution of the paper, which I do not believe to be inremental. The authors are correct that other papers have similar gains, we might ask ourselves if any of this is really good enough when other approaches are reaching state of the art performance. Nonetheless, I do think the performance is sufficient for ICLR.
> >
> > Regarding weakness 3, I am not entirely sure I am on board with the authors here so I would appreciate another response from them. My issue is if you take a real-world problem, will you need so many "experts" that the model does not fit into memory anymore? Of course there is an exponential growth in the number of problems covered, but that I already knew and do not really care about. My concern is whether this approach actually scales to real problems with many different constraint types. How many experts are possible / what is the size of the model in terms of parameters for each extra expert? How much harder does the model become to train? This is a legitimate concern, and I do not think the arguments regarding CaDA really work here.

---

> > > ### Author Response · Authors · 2025-11-27
> > >
> > > Thank you for your acknowledgement and constructive discussion. We are encouraged that you view our contributions as meaningful and that you recognize the experimental performance sufficient for ICLR.
> > >
> > > Below we address your remaining concerns.
> > >
> > > ## Regarding Weakness 2:
> > >
> > > We fully agree with your broader point, "we might ask ourselves if any of this is really good enough when other approaches are reaching state of the art performance." Improvements driven purely by engineering tricks would indeed offer limited long-term value. Our goal, however, is not solely to push performance numbers. Rather, the observed gains primarily serve as evidence that the methodological insight behind our design is effective.
> > >
> > > The architectural idea we introduce is grounded in the structural characteristics of VRP variants. We believe such general and conceptual insights are valuable to the community, as they can guide the development of methods for other problems with similar structural properties, beyond the specific benchmarks evaluated here.
> > >
> > > In this sense, our contribution lies not only in numerical improvements but also in expanding the methodological toolkit with a principled design that has demonstrated practical impact.

---

> > > ### Author Response · Authors · 2025-11-27
> > >
> > > ## Regarding Weakness 3:
> > >
> > > We appreciate your follow-up regarding scalability. Below, we provide a quantitative analysis of expert size, memory usage, and training difficulty (Points 1-3) and demonstrate CoEKS's feasibility of scaling to real-world VRPs (Point 4).
> > >
> > > **1. What is the size of the model in terms of parameters for each extra expert?**
> > >
> > > Based on our architecture, each expert consists of two linear layers (with $d=128$ and $d_f=512$). With $N=6$ encoder layers, adding one constraint-specific expert adds approximately 0.79 million parameters.
> > >
> > > **2. How many experts are possible？**
> > >
> > > * **Real-world VRP Constraints:** While theoretically bounded only by hardware, a survey on Rich Vehicle Routing Problems [1] categorizes real-world operational requirements into a finite taxonomy (e.g., time windows, multi-depot, compatibility), typically numbering fewer than 20 distinct constraints (corresponding to fewer than 20 experts).
> > > * **Memory-based Feasibility:** Even if we adopt this comprehensive upper bound of 20 distinct experts, the total parameter count for the expert library would be roughly 15.7 million. In terms of memory, this occupies only ~64 MB (FP32), making the deployment of 20 experts entirely feasible.
> > >
> > > **3. How much harder does the model become to train？**
> > >
> > > * **Memory Consumption:** In practical training, GPU memory is dominated by storing activation maps (which also scales with batch size). For example, training CoEKS on $N=50$ instances with a batch size of 256 consumes approximately 6300 MB of GPU memory. The ~3 MB memory increase from adding a new expert (0.79M parameters) is virtually imperceptible in this context.
> > > * **Model Convergence:** Our "Plug-and-Play" strategy (see Section 4.3) freezes the backbone and optimizes only the new expert. Training such a small sub-module converges significantly faster than retraining a full dense model and avoids catastrophic forgetting.
> > >
> > > **4. Feasibility of Scaling to Real-world VRPs with Many Constraints**
> > >
> > > * **Scalability as a Design Advantage: The Need for Expandable Capacity in Complex VRPs.** Real-world VRPs increase in structural complexity as more heterogeneous constraints are introduced. In such settings, allowing the model to scale its capacity is a rational and worth trade-off as discussed in our previous response to Reviewer zfcw's Weakness 2. CoEKS's plug-in expert architecture provides this flexibility, whereas existing fixed-capacity methods such as ReLD-MoEL or CaDA cannot easily expand their representational power when encountering substantially more complex or diverse constraints. This fixed-capacity design of existing methods should be regarded as a limitation rather than an advantage.
> > > * **A Resource-Constrained Fallback Option: Shared Expert Configuration.** We also acknowledge the concern that extremely many constraint or strict memory budgets may make per-constraint experts impractical. CoEKS is not strictly tied to unlimited expansion; when resources are limited, a single expert can be assigned to handle multiple constraints or hard-to-decompose constraints. To verify this, we conducted an experiment on new MD and MB tasks by introducing only one expert ($E_{DB}$)  shared across all these new constraints. As shown in the table below, CoEKS (FT-$E_{DB}$) still significantly outperforms the ReLD-MoEL baseline, demonstrating that the architecture remains effective even under fixed-capacity settings like existing methods.
> > >
> > > Method \ Gap↓($n=50$)| MDOVRPMB | MDOVRPMBL | MDOVRPMBLTW | MDOVRPMBTW | MDVRPMB | MDVRPMBL | MDVRPMBLTW | MDVRPMBTW
> > > ---|---|---|---|---|---|---|---|---
> > >  ReLD-MoEL | 17.790% | 17.543% | 11.430% | 11.206% | 23.673% | 22.101% | 15.081% | 14.417%
> > >  CoEKS(FT-$E_{DB}$) | 8.813% | 8.857% | 5.784% | 5.654% | 14.659% | 14.700% | 9.840% | 9.580%
> > >  CoEKS | **6.959%** | **7.061%** | **5.250%** | **5.130%**| **12.128%** | **12.032%** | **9.122%** | **8.804%**
> > >
> > > Moreover, we note that sharing one expert across multiple or hard-to-decompose constraints inevitably introduces some performance trade-off, as shown in the table above. Future work may explore more efficient expert-sharing mechanisms to mitigate this (as discussed in Conclusion in our paper), though such developments are beyond the scope of the present study.
> > >
> > > **5. Clarification of Discussion of CaDA**
> > >
> > > Our earlier discussion of CaDA was only meant to respond to your comment on vector-based approaches like PolyNet, highlighting that their effectiveness is more limited in multi-task VRP. It was not intended as an argument related to your scalability concern, which we address separately above.
> > >
> > > [1] Rich vehicle routing problem: Survey. ACM Computing Surveys, 2014.

---

### Official Review · Reviewer_zFcW · 2025-10-31

**Soundness:** 3
**Presentation:** 4
**Contribution:** 3
**Rating:** 6
**Confidence:** 4

**Summary:**

This paper proposes CoEKS, a new method to encode problems in multi-task learning for VRPs. Instead of employing MoE in encoder side, the authors propose to assign an expert to each constraint. For problems with multiple constraints, all the respective experts of each constraint will be activated at the same time, and a sharing mechanism enhances performance. The model can also be adapted for new problems by adding a new expert for each constraint. CoEKS demonstrates SOTA results among neural methods.

**Strengths:**

1. The paper is well-written, easy to follow and with good motivation

2. CoEKS is novel to my knowledge and tackles the problem in an interesting manner

3. Good performance

**Weaknesses:**

1. My major concern is the following: given that we have multiple experts, why not simply train on all of them at the same time and always activate them? In other words, why bother with specialization when one can have multiple MLPs at the same time (or a larger MLP with the same number of parameters with the sum of experts)? I believe the bump in performance might be simply due to having more parameters

2. The method will need to activate all the parameters for problems, including all of the constraints, which is however acknowledged by the authors

3. Overall, the work is a bit incremental, in the sense that it just adds a specific mechanism for the encoder layers with a relatively small improvement in performance

4. Code is not provided

**Questions:**

1. What if instead of experts we just had all MLPs work at the same time, or a larger MLP with the same number of parameters?

2. I am not too clear about the relevance of OOD generalization to “unseen” tasks. As the tasks in Table 2 are merely a combination of constraints, in practice, why not simply train a method that incorporates them already?

3. Why do you finetune only the added experts? Wouldn’t the performance improve if you finetune all of the models as well?

4. It would be interesting to see what the learned representations of experts are in the latent space. Do they align with the specified constraints, or do experts learn generic representations? It would be interesting to see more about this in say a t-SNE analysis similar to POMO-MTL or RF

---

> ### Author Response · Authors · 2025-11-21
> **Response to Reviewer zFcW [1/5]**
>
> We sincerely appreciate the reviewer's recognition of our paper as well-written, easy to follow, and novel with good motivation and performance. Below, we address each concern point by point. All revisions and additions made to the manuscript have been marked in blue to aid in your review.
>
> ## To Weakness 1 and Question 1:
> We appreciate the reviewer’s thoughtful comment. We address your concern from two perspectives: the efficiency of sparse activation and the impact of CoEKS's architecture on performance.
>
> **1. The Efficiency of Sparse Activation**
>
> “always activating all experts” is not a reasonable or effective alternative. Simply activating all experts is equivalent to replacing the expert module with a dense, fully activated model, which contradicts the core rationale behind sparse expert architectures. Prior work on sparse models [1-2] has shown that:
> * **Sparse activation enables parameter efficiency:** only a small subset of experts is evaluated per instance, allowing the model to scale capacity without proportional increases in compute.
> * **Specialization emerges naturally:** experts trained on different subsets (or different structural priors, as in CoEKS) learn complementary behaviors, which is essential for generalization across different tasks.
> * **Fully dense expert usage destroys specialization:** This prevents the model from leveraging constraint-level Knowledge, one of the core challenges in cross-task VRPs. To verify this motivation and test your hypothesis, we replaced the combination-of-experts (CoE) architecture in CoEKS with a large Dense MLP. The Dense MLP's hidden dimension was set to 2560, resulting in a total parameter count of 4.6 million, which was matched to CoEKS. The results are shown in the tables below. These results clearly show that CoEKS achieves superior performance with the same (or fewer) parameters than the large dense MLP variant:
> * **ID tasks:** CoEKS outperforms the large MLP by **+9.2%** on average.
> * **OOD tasks:** The gap becomes even larger, with CoEKS achieving a **+29.3%** relative improvement.
>
> **In-Distribution ($n=50$):**
>
> | Method \ Gap↓        | CVRP    | OVRP    | VRPB    | VRPL    | VRPTW   | OVRPTW  | OVRPL   | (ID Avg.)  |
> |--|-|-|-|-|-|-|-|-|
>  large Dense MLP | 0.917% | 2.395% | 2.634% | 1.151% | 2.408% | 1.608% | 2.398% | 1.9303%
>  CoEKS                  | **0.8914%** | **2.1376%** | **2.4968%** | **1.1523%** | **2.0500%** | **1.3928%** | **2.1352%** | **1.7509%**
>
> **Out-of-Distribution ($n=50$):**
>
> | Method \ Gap↓        | OVRPB   | VRPBL   | VRPBTW  | VRPLTW  | OVRPBL  | OVRPBTW | OVRPLTW | VRPBLTW | OVRPBLTW | (OOD Avg.)  |
> |-|-|-|-|-|-|-|-|-|-|-|
> large Dense MLP | 5.868% | 5.248% | 6.840% | 4.133% | 4.377% | 3.420% | 1.896% | 7.369% | 4.477% | 4.8474%
> CoEKS                  | **4.9127%** | **4.3871%** | **4.7473%**  | **3.2098%** | **2.9489%** | **2.5664%** | **1.6026%** | **3.7140%** | **2.7969%**  | **3.4317%**
>
> **2. CoEKS’s Performance Is Driven by Its Architecture Design, Not by Simply Increasing Parameters**
>
> While it may seem that adding more experts (and thus more parameters) could lead to better performance, we believe that the performance gains of CoEKS are not simply due to having more parameters, but are instead driven by the two complementary components of CoEKS.
>
> To verify this, in Appendix F.1, we also compared CoEKS with sparse model ReLD-MoEL-E5 (adapted from the state-of-the-art MoE model [3]), whose parameters are matched to CoEKS by adding extra experts. CoEKS still demonstrated superior performance, achieving relative improvements of **15.4% (ID)** and **22.1% (OOD)**.
>
> These results further confirm that **CoEKS’s performance does not stem from model size**, but from its two complementary components: a **sparse CoE architecture** that adaptively combines constraint-specific experts for diverse VRPs, and a **multi-view knowledge sharing strategy** that automatically learns transferable knowledge to enhance cross-task generalization.

---

> ### Author Response · Authors · 2025-11-21
> **Response to Reviewer zFcW [2/5]**
>
> ## To Weakness 2:
> We appreciate the reviewer's observation. We must clarify that our method employs sparse activation for parameters, achieving a favorable trade-off between parameter count and performance.
>
> **1. Conditional Expert Activation.**
>
> CoEKS activates only the expert parameters associated with the constraints present in a given VRP instance. Therefore, tasks involving all constraint types naturally require activating all corresponding experts, whereas tasks containing only a subset of constraints activate only the relevant subset of parameters.
>
> **2. Justified Trade-Off Between Parameter Count and Performance.**
>
> As originally discussed in our submission, the increase in expert count is necessary to capture complex constraint knowledge. Indeed, further experiments demonstrate that the performance gains significantly outweigh the parameter cost:
> * **For Simpler Tasks (2 Constraints):** When the task complexity is lower, both CoEKS and ReLD-MoEL (the SOTA MoE model) activate only 2 experts, while CoEKS still achieves a 7% average relative improvement.
> * **For More Complex Tasks (3 Constraints):** When the task complexity increases, CoEKS automatically adapts by activating **one additional expert** (adding only **0.79 million** parameters). This minor increment in active parameter count yields a disproportionately larger performance boost, resulting in a substantial **13% average relative improvement** over ReLD-MoEL.
>
> These results show that the parameter overhead of the dynamically activated expert is **highly cost-effective**. By utilizing more experts only when needed for complex tasks, CoEKS ensures that the marginal increase in computational cost is overwhelmingly justified by the **significant jump in solution quality**, confirming the necessity of our flexible expert system for tackling increasing task complexity.

---

> ### Author Response · Authors · 2025-11-21
> **Response to Reviewer zFcW [3/5]**
>
> ## To Weakness 3:
> Thank you for this thoughtful comment. We would like to clarify that CoEKS is not an incremental encoder tweak, but a principled framework grounded in two key contributions: a methodological design and substantial improvements demonstrated through comprehensive experiments.
>
> **Methodological Contribution:**
> * **Problem-specific architectural design:**
> Our work begins with a core structural observation: many VRP variants can be expressed as modular compositions of constraints (e.g., capacity, time windows), an aspect that existing neural cross-task methods largely overlook. Leveraging this prior, we design a Combination-of-Experts (CoE) architecture where each constraint is assigned to a dedicated expert and recombined dynamically. This structural perspective is not limited to VRPs but extends to other combinatorial optimization problems with similar forms of constraint modularity, such as job shop scheduling (e.g., decomposing by machine precedence or operation constraints).
> * **Complementary components integrating specialized and general knowledge:** Constraint-specialized experts serve as modular knowledge units aligned with the prior structure, while multi-view knowledge sharing captures cross-constraint general knowledge through mutual distillation and shared transformation layers. These two components work synergistically to enhance both specialization and generalization.
> * **Plug-in Scalability**: CoEKS enables efficient scaling to new constraints by simply adding and fine-tuning new experts without modifying the existing model (as demonstrated in Tables 3 and 7). This plug-and-play capability makes the approach well-suited for continual learning and deployment in dynamic environments.
>
> These innovations make CoEKS a principled and practical solution for solving cross-task VRPs, with significant implications for advancing the field.
>
> **Significant Improvement of Performance:**
> In CO problems like VRPs, where prior methods have already achieved strong baselines through extensive optimization, subsequent improvements become increasingly difficult due to diminishing returns. Our contributions are validated through a comprehensive experimental study across 5 constraints and their combinations, covering both ID and OOD settings. To provide context, we compare against recent state-of-the-art methods published in top conferences. For instance:
> * MVMoE [5] achieved relative improvements of +8.4% (ID) / +4.8% (OOD) over its baselines,
> * ReLD-MoEL reported +21.6% (ID) / +15.1% (OOD),
> * CaDA [6] showed +13.1% overall,
> * RouteFinder gained +9.8% overall.
>
> Under the same 50-node setting, CoEKS achieves **+7.9% (ID)** and **+18.3% (OOD)** over these methods. Notably, the +18.3% OOD gain represents the **largest reported to date** among recent cross-task VRP methods, marking a substantial breakthrough in OOD generalization despite the inherent challenges of further advancing well-optimized CO problems. This demonstrates that our empirical improvement is significant rather than relatively small.
>
> Most importantly, the novelty of our work lies in its **methodological contribution**, not in any single implementation detail. The core idea is to identify and exploit the modular constraint structure of VRPs—an overlooked but fundamental property—and operationalize it through a principled combination of constraint-specialized experts and controlled knowledge sharing. The specific encoder-level realization presented in the paper is simply one instantiation of this broader design paradigm. Indeed, many recent top-tier works [3–6] similarly contribute new insights through implementations that may appear lightweight but are grounded in strong conceptual advances. In this sense, CoEKS should not be viewed as incremental; its strength comes from the **architectural principle**, not from the surface-level complexity of the modification.

---

> ### Author Response · Authors · 2025-11-21
> **Response to Reviewer zFcW [4/5]**
>
> ## To Weakness 4:
> Thank you for the comment. We have made the full CoEKS code publicly available at [https://anonymous.4open.science/r/CoEKS-B0D9/](https://anonymous.4open.science/r/CoEKS-B0D9/), and the access link is already included in the experiment section of the manuscript.
>
> ## To Question 2:
> Thank you for the question. We appreciate the opportunity to clarify the scope of OOD generalization and its practical relevance.
>
> **1. Clarification of OOD Scope**
>
> The term OOD generalization in our paper covers three distinct scenarios. Two of these are our primary focus (as stated in lines 51-53) and relate directly to task-level generalization, which addresses your question about "unseen" tasks:
>
> 1. Unseen Constraint Combinations: Generalizing to tasks that involve novel combinations of constraints already present in the training set (e.g., training on {VRPL, VRPBTW}, but testing on {VRPBL, VRPLTW, VRPBLTW}).
> 2. Tasks Involving New Constraints: Generalizing to tasks that incorporate basic constraints that were not part of the training set.
>
> Both categories consist of tasks not observed during training, which is why we describe them as **“unseen.”**
>
> The third OOD scenario involves generalizing to unseen distributions (e.g., benchmark datasets like CVRPLIB with larger-scale instances or different node distributions), as discussed in Appendix D. The core purpose is to test the model's generalization capability beyond its training distribution.
>
> **2. Rationale for OOD Evaluation**
>
> The experimental setup intentionally excludes some combinations to test OOD capabilities, because in real-world logistics systems:
> * It is impractical to anticipate and enumerate **all possible combinations** beforehand.
> * Some constraints may appear only in rare or **future scenarios**.
>
> Our goal is to build a model that can be **efficiently and effectively generalized** to new VRP variants without retraining.
>
> **3. Additional Results with All Tasks Included in Training**
>
> In Appendix E.1 of the original submission, we also explored in-distribution performance under a different training setup. Following the same configuration as RouteFinder, we trained the model on tasks involving all constraint combinations. The results show that **CoEKS achieves the best overall performance**, demonstrating that its superiority is consistent across different training regimes.

---

> ### Author Response · Authors · 2025-11-21
> **Response to Reviewer zFcW [5/5]**
>
> ## To Question 3:
> Thank you for the question. We intentionally restrict fine-tuning to only the newly added experts for two primary, interconnected reasons: **Robustness** and **Efficiency**.
>
> **1. Preventing Catastrophic Forgetting (Robustness)**
>
> Fine-tuning all parameters of the model would inevitably overwrite the knowledge learned from previous tasks, leading to catastrophic forgetting. As shown in the table below, we empirically verified this: when the entire model undergoes fine-tuning (Full FT), performance on previously mastered tasks drops significantly. Conversely, our approach of fine-tuning only the added expert (Added Expert FT) successfully preserves this prior knowledge and maintains high performance across the complete set of tasks.
>
>  | Method \ Gap↓             | CVRP   | OVRP   | VRPB    | VRPL   | VRPTW  | OVRPTW | OVRPL  |
> |-----------------|--------|--------|---------|--------|--------|--------|--------|
> | Full FT         | 1.413% | 3.010% | 17.207% | 1.630% | 2.735% | 2.207% | 2.920% |
> | added Expert FT | **0.891%** | **2.138%** | **2.497%** | **1.152%** | **2.050%** | **1.393%** | **2.135%**|
>
>
> | Method \ Gap↓     | OVRPB   | VRPBL   | OVRPBL  | VRPBTW | VRPLTW | OVRPBTW | OVRPLTW | VRPBLTW | OVRPBLTW |
> |-----------------|---------|---------|---------|--------|--------|---------|---------|---------|----------|
> | Full FT         | 11.606% | 15.359% | 11.614% | 3.684% | 3.221% | 3.391%  | 2.104%  | 3.948%  | 3.338%   |
> | added Expert FT | **4.913%**  | **4.387%**  | **4.747%**  | **3.210%** | **2.949%** | **2.566%**  | **1.603%** | **3.714%**  | **2.797%**   |
>
> **2. Cost-Effective Scalability (Efficiency):**
>
> Our strategy allows the model to quickly and effectively adapt to new constraints at a substantially lower computational cost. By only updating a small fraction of the total parameters, we drastically reduce the required computation time and resources. This is crucial for promoting a scalable, "plug-and-play"deployment paradigm, especially in dynamic, real-world environments where new constraints are introduced frequently.
> ## To Question 4:
> Thank you for this excellent and insightful suggestion! We fully agree that a t-SNE analysis is crucial for visually demonstrating the nature of our learned expert representations. We have included this analysis in the Appendix F.2, and the results strongly align with our architectural design principles.
>
> **1. t-SNE Analysis Setup**
>
> We conducted t-SNE on the **expert output embeddings** for the full-constraint task OVRPBLTW. Specifically, for each expert at each encoder layer, we collected 1,000 embedding samples and projected them into a 2-D latent space.
>
> **2. Analysis of CoEKS Model**
>
> The t-SNE plots clearly illustrate the intended two-stage learning process, validating our complementary design of learning general knowledge in shallow layers and specialized knowledge in deep layers:
> * Shallow Layers (layer 1): With the exception of the Capacity Expert $E_C$, the embeddings of the other experts show high overlap. This is a strong indicator that shallow layers successfully capture shared, transferable representations, consistent with our design where knowledge sharing is applied only at lower layers. Furthermore, since the capacity expert ($E_C$) remains active across all VRP tasks, the representations it learns tend to be more universal and fundamental. Consequently, Expert C acquires a distinct and stable representation early on, differing from those of other experts.
> * Deeper Layers (Layer 2-6): As we move to deeper layers, the clusters become clearly separated, showing that experts gradually specialize and align with their assigned constraints.
>
> We also visualized the model without the knowledge sharing mechanism. In this case, experts become separated even in the first layer, indicating premature specialization and a lack of transferable knowledge. This confirms that our knowledge sharing mechanism is necessary for building meaningful cross-task representations.
>
> Compared with prior works such as POMO-MTL and RouteFinder, which show task-level clustering, our t-SNE results demonstrate constraint-level disentanglement. Our ability to explicitly decouple and differentiate learning for individual constraints is the key to achieving superior cross-task generalization.
>
> [1] Outrageously Large Neural Networks: The Sparsely-Gated Mixture-of-Experts Layer. arXiv, 2017.
> [2] DeepSeekMoE: Towards Ultimate Expert Specialization in Mixture-of-Experts Language Models. ACL, 2024.
> [3] Rethinking light decoder-based solvers for vehicle routing problems. ICLR, 2025.
> [4] Routefinder: Towards foundation models for vehicle routing problems. TMLR, 2025.
> [5] Mvmoe: multi-task vehicle routing solver with mixture-of-experts. ICML, 2024.
> [6] CaDA: Cross-Problem Routing Solver with Constraint-Aware Dual-Attention. ICML, 2025.

---

> > ### Comment · Reviewer_zFcW · 2025-11-26
> >
> > I thank the authors for their comprehensive rebuttal. I reviewed other rebuttals as well, and I did not find any major outstanding concerns. As my concerns have been fully addressed, I would be happy to recommend CoEKS for acceptance.

---

> > > ### Author Response · Authors · 2025-11-27
> > >
> > > We sincerely thank the reviewer for the positive feedback and the recommendation for acceptance. We are glad to learn that our rebuttal has successfully addressed your concerns. We appreciate the time and effort you have dedicated to reviewing our paper.

---

### Official Review · Reviewer_nRLf · 2025-11-01

**Soundness:** 3
**Presentation:** 2
**Contribution:** 2
**Rating:** 4
**Confidence:** 3

**Summary:**

This paper addresses the problem of vehicle routing problem in a multi task setting.
This work modifies the Mixture of Experts (MoE) model into a Combination of Experts (CoE) framework, where experts are constraint-specific and are combined later. Knowledge sharing with MSE loss is applied between the experts. A combiner is introduced to regularize the model. Experiments are conducted to show the validitty of the method.

**Strengths:**

1. The idea of combining experts is intuitive and well-aligned with the CVRP multi-task setting.
2. The use of separate heads for each constraint is straightforward and easy to understand.

**Weaknesses:**

1. **Writing needs improvement:**
   The description of the combiner is presented in two separate sections (lines 203–211 and lines 263–275), with no clear connection between them. It is unclear whether these sections refer to the same concept or different ones. If they are the same, the repetition needs to be justified. If they are different, the components should be clearly defined.

2. **MD loss for \( K = 2 \) vs \( K > 2 \):**
   The rationale behind the different MD loss for \( K = 2 \) and \( K > 2 \) is not provided. There is no explanation for this distinction.

3. **Distillation loss (MD) adoption from (Xie et al., 2024): [1]**
   The MD loss is adopted from (Xie et al., 2024), but it is unclear whether this approach has been extensively studied in other works besides Xie et al. (2024). Regardless, the reasoning behind using distillation when both heads are learning simultaneously is not clear. It appears to be primarily a regularization technique, restricting the output heads to remain similar. Even with this interpretation, it is not clear why this regularization is combined with the Mixture of Experts model (or the Combination of Experts model, in this case), which involves a weighted sum of outputs. The main motivation behind MoE (or CoE) is to capture distinct modes across different network heads and combine them. This regularization approach would work against that motivation, as it encourages all experts to output similar embeddings.

4. **Comparison with SHIELD:**
   A relevant work, **SHIELD**[2], presented at ICML, shares a similar motivation of generalizing across different tasks and distributions, but incorporates tasks as part of the input features. This work is neither compared nor discussed in the paper.

## Overall:
The idea of combining different heads for different tasks is reasonable. However, the framework appears somewhat incremental, with the primary modification being the adaptation of the Mixture of Experts model to fit experts to different subtasks, along with the addition of regularization techniques.

## References:
1. Zhitian Xie, Yinger Zhang, Chenyi Zhuang, Qitao Shi, Zhining Liu, Jinjie Gu, and Guannan Zhang. Mode: A mixture-of-experts model with mutual distillation among the experts. In Proceedings of the AAAI Conference on Artificial Intelligence, volume 38, pp. 16067–16075, 2024
1. Goh, Yong Liang, et al. "SHIELD: Multi-task Multi-distribution Vehicle Routing Solver with Sparsity and Hierarchy." Forty-second International Conference on Machine Learning.

**Questions:**

Please see weaknesses

---

> ### Author Response · Authors · 2025-11-21
> **Response to Reviewer nRLf [1/4]**
>
> We sincerely appreciate the reviewer's careful scrutiny and valuable feedback. Below, we address each concern point by point. All revisions and additions made to the manuscript have been marked in blue to aid in your review.
> ## To Weakness 1:
> We thank the reviewer for this helpful feedback. We would like to clarify the relationship between these two sections. The original lines 203–211 define the fundamental architecture of the Combiner within the CoE architecture, whereas the original lines 263–275 describe the specific application of the knowledge sharing strategy to the Combiner. They address distinct aspects of the model design:
>
> Part 1 (the original lines 203–211): This section introduces the basic definition and functionality of the Combiner in the CoE architecture. As shown in Eq. (3) and Eq. (4), its role is to calculate weights $S_j(h)$ based on the input embedding $h$ and perform a weighted aggregation of the activated experts' outputs $E_j(h)$.
>
> Part 2 (the original lines 263–275): This section belongs to the "Multi-view knowledge sharing strategy" and specifically details "Combiner-view knowledge sharing." Here, we introduce a shared transformation layer $f_s$ before the Combiner. Consequently, the input to the Combiner is modified from the raw embedding $h$ to the processed representation $f_s(h)$ (as shown in Eq. 7).
>
> To clarify the connection between these sections, we have added a transitional sentence in lines 269–270: "Specifically, a shared transformation layer $f_s$ is applied to the input embedding $h$ before it reaches the combiners introduced in Section 4.1." This addition clarifies the logical connection between the base architecture and the knowledge sharing enhancement.
>
> ## To Weakness 2:
> Thank you for pointing this out. In the original submission, we briefly mentioned the rationale behind the distinction between the MD loss for $K = 2$ and $K > 2$—"The virtual expert simplifies computation for $K > 2$, guiding experts toward a consensus without pairwise comparisons." According to your suggestions, we would like to provide a more detailed explanation.
>
> The distinction is primarily driven by computational efficiency, while maintaining the same optimization goal of reducing variance among the expert outputs:
>
> When $K = 2$ (two active experts), the simplest and most direct way to encourage them to "learn from each other" or reach a "consensus" is by computing the MD loss as the Mean Squared Error (MSE) between their outputs:
> $${L}_\mathrm{md} = \text{MSE}(E_1(h), E_2(h)) = \text{Mean}(||E_1(h) - E_2(h)||^2)$$
>
> This method effectively minimizes the variance between them without needing auxiliary calculations.
>
> When $K > 2$ (e.g., $K = 5$), we introduce a "virtual expert" ( $E_{\text{avg}}(h) = \frac{1}{K} \sum_{i=1}^{K} E_i(h)$) to assist in calculating the MD loss. This simplifies the calculation from pairwise comparisons to comparisons with the average output of all experts:
>
> $$
> L_{\mathrm{md}} = \frac{1}{K} \sum_{i=1}^{K} \mathrm{MSE}(E_i(h), E_{\mathrm{avg}}(h))
> $$
>
>
> By using the "virtual expert", the complexity is reduced to $O(K)$, compared to the $O(K^2)$ complexity required for pairwise comparisons (which would involve $C(K, 2)=\frac{K(K-1)}{2}$ pairwise calculations). This reduction in computational complexity is crucial for efficiency.
>
> In both cases, the goal is the same: to minimize the variance between the expert outputs, thereby encouraging knowledge sharing among experts. We have revised Section 4.2 to more clearly articulate the computational trade-offs and the unified optimization goal.

---

> ### Author Response · Authors · 2025-11-21
> **Response to Reviewer nRLf [2/4]**
>
> ## To Weakness 3:
> Thank you for these insightful questions. We address your concerns in three parts: (1) related work on mutual distillation (MD), (2) the intrinsic mechanism that makes MD effective in CoEKS, and (3) how we reconcile knowledge sharing with expert specialization.
>
> **1. Related Work on MD**
>
> Similar concepts of mutual learning have been studied before [1]. Deep Mutual Learning [2] allowed models to learn from each other by computing cross-entropy loss between each pair of models. This process is supported by the "Multi-view" hypothesis [3], which shows that leveraging diverse feature views improves generalization.
>
> In CoEKS, MD serves as a similar feature enrichment mechanism, broadening each expert's perspective by exposing it to complementary features from peer experts without requiring them to process all data.
>
> **2. Underlying Mechanism Why MD Works in CoEKS**
>
> The adoption of MD is not merely a regularization technique. It addresses a fundamental limitation of expert-based models known as "Narrow Vision". Standard routing restricts individual experts to limited data subsets, causing them to miss potentially useful features captured by their peers. Our t-SNE analysis (Appendix F.2) illustrates this mechanism:
> * **With MD:** the t-SNE plots reveal a significant overlap in the embeddings of all experts, except for the capacity expert, within the lower layers (layer 1). This indicates that the lower layers capture shared, transferable representations, aligning with our design where knowledge sharing is applied at the lower layers. As we move to deeper layers, the clusters become clearly separated, showing that experts gradually specialize and align with their assigned constraints.
> * **Without MD:** Expert representations begin to separate already in the first layer, indicating that early specialization leads to the absence of transferable knowledge. This indicates the crucial importance of MD in building meaningful cross-task representations.
>
> Thus, MD provides a principled mechanism for enhancing generalizable representations without sacrificing specialization.
>
> **3. Resolving the Potential Conflict with Controlled Knowledge Sharing**
>
> We agree that unrestricted MD would undermine expert specialization. To ensure MD benefits generalization without compromising specialization, we implemented two critical design constraints:
>
>
> * **Restricted Location:** We apply MD only to the first layer of the encoder. Inspired by deep learning interpretability [4], we posit that lower layers capture general, transferable features (where consensus is beneficial), while higher layers develop task-specific logic.
> Our experimental results (Figures 4a-b) validate this. When MD is applied only to the first layer, OOD performance is best. However, when MD is applied across all layers, OOD performance significantly degrades (even worse than without MD), confirming your concern that unchecked MD can undermine specialization. Our specific design avoids this pitfall.
> * **Controlled Strength:** We set the distillation strength $\alpha$ to a very low value of 0.01. This ensures MD acts as a "gentle nudge" for sharing, not a hard constraint forcing homogeneity.
> As verified in Figures 4c-d, OOD performance reaches its optimum when $\alpha$ is set to 0.01. Larger values of $\alpha$, however, rapidly diminish the distinctiveness of the experts, impairing generalization ability.
>
> Furthermore, our ablation studies (Figures 3c–d) also indicate that completely removing MD significantly impairs OOD generalization, confirming that this controlled knowledge sharing mechanism is an essential component of CoEKS's effectiveness.

---

> ### Author Response · Authors · 2025-11-21
> **Response to Reviewer nRLf [3/4]**
>
> ## To Weakness 4:
> We thank the reviewer for pointing out the value of SHIELD as a relevant contemporary work, and we have updated the "Related Work" section to include a discussion of its method. However, we believe a direct empirical comparison is currently inappropriate due to key differences in research focus and unreleased SHIELD resources.
> 1. These two works are orthogonal: SHIELD enhances the decoder with sparsity (Mixture-of-Depths) and hierarchy (context-based clustering) to handle distribution complexity in Multi-Task Multi-Distribution VRP (MTMDVRP), while CoEKS introduces a constraint-specific combination-of-experts architecture with knowledge sharing into the encoder to address OOD constraint combinations and plug-in adaptation to new constraints in Multi-Task VRP (MTVRP), referred to as cross-task VRP in our paper. Incorporating a stronger MTVRP backbone such as CoEKS may further enhance SHIELD’s capability to handle MTMDVRP. From this perspective, our method can be viewed as a natural extension to SHIELD’s base model rather than a competing alternative.
> 2. Unreleased SHIELD Resources: Because SHIELD’s code and dataset are not publicly available, it is not possible to include a direct empirical comparison within the limited rebuttal period.
>
> Additionally, we have compared our method with CaDA [5], a contemporaneous work also presented at ICML 2025, which focuses on the MTVRP setting and aligns more closely with our research scope.
>
> Given these considerations, we believe our current comparisons are appropriate for the problem studied. In future work, we will explore integrating CoEKS with SHIELD to better address MTMDVRP.

---

> ### Author Response · Authors · 2025-11-21
> **Response to Reviewer nRLf [4/4]**
>
> ## To the overall weakness:
> We appreciate the reviewer’s comments. However, we respectfully clarify that CoEKS is not merely an "adaptation of the Mixture of Experts model with added regularization," but a novel and principled architecture specifically tailored for cross-task VRPs, introducing crucial distinctions that yield significant novelty and practical advantages.
>
> **Adaptation of Task-Level MoE with an MD Regularization**
>
> Existing MoE-based models (e.g., MVMoE [6] and ReLD-MoEL [7]), called node-level MoE models, route each node embedding independently via gating. This routing mechanism lacks constraint-level semantics and often leads to sparse and unstable expert utilization.
>
> Different from node-level MoE models, we construct a **task-level MoE with MD** baseline as follows:
> * Instances belonging to the same VRP variant (determined by a specific combination of constraints) are routed to fixed Top-K experts.
> * Expert selection remains learned via gating.
> * MD is applied among the activated experts.
>
> **Advantages of CoEKS over Task-Level MoE with MD**
>
> CoEKS introduces a principled and interpretable architecture specifically tailored for cross-task VRPs. Its key advantages are:
>
> * **Semantically grounded routing:** Unlike MoE gating, which lacks semantic alignment, CoEKS explicitly incorporates structured constraint priors when combining experts. This improves interpretability and efficiency.
> * **Modularity and plug-in scalability:** New constraints can be handled by adding and fine-tuning a dedicated expert without modifying the existing model. This structural modularity is validated in our scalability experiments (Tables 3 and 7), and offers practical benefits compared to end-to-end MoE methods.
> * **Complementary components:** Our knowledge sharing (KS) strategy is complementary to the CoE architecture. CoE ensures explicit constraint-specific specialization, while KS enriches features without undermining expert roles—a balance that is difficult to guarantee in a standard, unstructured task-level MoE.
> * **Stable and balanced expert usage:** CoEKS avoids the underutilization and imbalance issues common in MoE (due to over-selection of dominant experts), as routing is fixed by task semantics.
>
> **Experimental Verification**
>
> We demonstrate the superiority of CoEKS by comparing it with the task-level MoE with MD (adapted from ReLD-MoEL [7]). As shown in the table below, **CoEKS significantly outperforms task-level MoE with MD**, especially on OOD tasks with constraint combinations. These results highlight that CoEKS’s design is not an incremental modification, but a structurally innovative architecture whose complementary components are essential for achieving superior cross-task VRP generalization.
>
> **In-Distribution ($n=50$):**
>
> | Method \ Gap↓        | CVRP    | OVRP    | VRPB    | VRPL    | VRPTW   | OVRPTW  | OVRPL   | (ID Avg.)  |
> |--|-|-|-|-|-|-|-|-|
> Task-level MOE with MD| 0.9187% | 2.3783% | 2.5324% | 1.1909% | 2.2068% | 1.4912% | 2.3948% | 1.8733%
>  CoEKS                  | **0.8914%** | **2.1376%** | **2.4968%** | **1.1523%** | **2.0500%** | **1.3928%** | **2.1352%** | **1.7509%**
>
> **Out-of-Distribution ($n=50$):**
>
> | Method \ Gap↓        | OVRPB   | VRPBL   | VRPBTW  | VRPLTW  | OVRPBL  | OVRPBTW | OVRPLTW | VRPBLTW | OVRPBLTW | (OOD Avg.)  |
> |-|-|-|-|-|-|-|-|-|-|-|
>  Task-level MOE with MD | 8.1223% | 5.5019% | 11.3296% | 3.7191% | 6.1209% | 2.9201% | 2.5888% | 9.2930% | 5.4344%  | 6.1145%
>  CoEKS                  | **4.9127%** | **4.3871%** | **4.7473%**  | **3.2098%** | **2.9489%** | **2.5664%** | **1.6026%** | **3.7140%** | **2.7969%**  | **3.4317%**
>
> [1] Mode: A mixture-of-experts model with mutual distillation among the experts. AAAI, 2024.
> [2] Deep mutual learning. CVPR, 2018.
> [3] Towards understanding ensemble, knowledge distillation and self-distillation in deep learning. ICLR, 2023.
> [4] Deep transfer learning with joint adaptation networks. ICML, 2017.
> [5] CaDA: Cross-Problem Routing Solver with Constraint-Aware Dual-Attention. ICML, 2025.
> [6] Mvmoe: multi-task vehicle routing solver with mixture-of-experts. ICML, 2024.
> [7] Rethinking light decoder-based solvers for vehicle routing problems. ICLR, 2025.

---

### Author Response · Authors · 2025-11-30
**Global Response [2/2]**

## Global Response [2/2]:
## Reviewer `dA3D`
In the first-round response, we addressed the reviewer’s primary concern regarding the experimental results and received acknowledgment that the **work makes a significant contribution** and the **experimental performance fully meets ICLR** standards. The second round of discussions then focused on a remaining core concern.

**Core Concern**:
The scalability of COEKS in real-world scenarios with numerous constraints.

**Key Response:**
We address the reviewer’s concerns mainly from two aspects.(detailed in the second-round response to Weakness 3, on 28 Nov)
* Robust Practical Feasibility:
Memory overhead (including both storage and training) is negligible.
* Strategic Advantage & Fallback Options:
Expandable architecture (plug-in mechanism) is essential for complex real-world VRPs, addressing the limitation of fixed-capacity designs in existing methods. For resource-constrained scenarios, our validated "Shared Expert" fallback option ensures strong reliability while maintaining significant performance advantages over SOTA baselines.


## Reviewer `qhjy`
In the first-round, we successfully addressed the reviewer's primary concern regarding the architectural novelty of CoEKS, demonstrating it to be a principled redesign rather than a handcrafted MoE variant. The reviewer appreciated our detailed response and engaged actively in the discussion, specifically emphasizing that demonstrating the method’s zero-shot generalization to unseen constraints would **“greatly enhance the contribution of the paper.”** They pinpointed two remaining core concerns for further validation:

**Core Concern**:
1. One-to-one mapping suitability, memory risks, and optimization stability in real-world scenarios .
2. Zero-shot generalization to unseen constraints.

**Key Response:**
1. Scalability to Real-World Scenarios (detailed in the second-round response to Concern 1, on 28 Nov):
    * Expandable architecture (plug-in mechanism) is essential for complex real-world VRPs, addressing the limitation of fixed-capacity designs in existing methods. For resource-constrained scenarios, our validated "Shared Expert" fallback option ensures strong reliability while maintaining significant performance advantages over SOTA baselines.
    * Memory overhead (including both storage and training) is negligible.
    * The reliance on auxiliary constraints stems from the inherent difficulty of task distribution shifts, rather than from our methodological limitation, since all existing methods encounter similar convergence challenges. Once training stabilizes, CoEKS significantly outperforms existing methods, demonstrating its superiority.
2. Superior Zero-Shot Generalization (detailed in the second-round response to Concern 2 and 3, on 28 Nov):
Through an extensive evaluation over 24 new tasks, we confirm that CoEKS **achieves superior zero-shot generalization over all baselines**. The credibility of these results is ensured by aligning with established literature and through the public release of our code and checkpoints.
---
We kindly ask you to evaluate our submission in light of the above feedback and improvements made to our manuscript. We believe we have addressed all major concerns raised by the reviewers, which has significantly improved the clarity and reliability of our manuscript.

Thank you for your time and your valuable service to the research community.

---

### Author Response · Authors · 2025-11-30
**Global Response [1/2]**

## Global Response [1/2]:

Dear ACs, SACs, and PCs,

We thank all reviewers for their valuable feedback and their recognition of CoEKS’s contributions. Across the reviews, the strengths highlighted include:

1. The paper presents a clear and well-motivated exposition. (Reviewers `zFCW`, `dA3D`, `qhjy`, `nRLf`)
2. The conceptual idea is novel, intuitive, and well-founded. (Reviewers `nRLf`, `zFCW`, `dA3D`)
3. The experimental results are strong and comprehensive, demonstrating state-of-the-art performance. (Reviewers `zFCW`, `qhjy`, `dA3D`)
4. The work makes a significant contribution to the field. (Reviewers `dA3D`, `zFCW`)
5. The experimental design is rigorous and demonstrates universality across different backbones. (Reviewer `qhjy`)

In response to the reviewers’ comments, we have conducted extensive additional experiments, revised the manuscript accordingly, and made our code and trained checkpoints publicly available (https://anonymous.4open.science/r/CoEKS-B0D9/) to ensure reproducibility and transparency.

Below, we will summarize the reviewers’ core concerns and our key responses (with precise locations indicated).

## Reviewer `nRLf`
**Core Concern:**
1. Questioned the rationale/conflict of the mutual distillation (MD) loss.
2. Assumed that CoEKS was merely an adaptation of MoE with additional regularization.

**Key Response:**
1. Regarding MD:
    * Explained the MD loss's rationale (detailed in the response to Weakness 2).
    * Controlled MD and expert specialization are complementary rather than conflicting (detailed in the response to Weakness 3).
2. CoEKS's advantages over an adaptation of MoE with MD (detailed in the response to the overall weakness):
    * Clarify what the *Adaptation of Task-Level MoE with added regularization* refers to.
    * Explain the advantages of CoEKS over *Task-Level MoE with MD*.
    * Experimentally validate that our proposed architecture significantly outperforms task-level MoE with MD, especially on OOD tasks.


## Reviewer `zFcW`
Although the reviewer **highly appreciated the writing quality and motivation**, they raised a core concern.

**Core Concern**:
The performance might gain stemmed solely from increased parameter counts rather than from architectural innovation.

**Key Response:**
Performance gains stem from architectural advantages (detailed in the response to Weakness 1 and Question 1):
* Clarified the efficiency of sparse activation.
* Added new experiments introduced parameter-matched Dense-MLP variants and the parameter-matched SOTA sparse model (ReLD-MoEL-E5), demonstrating that CoEKS’s advantages stem from its architectural design rather than merely from increased parameters.

After reviewing the rebuttal and other reviews, the reviewer concluded: “I reviewed other rebuttals as well, and I did not find any major outstanding concerns. As my concerns have been fully addressed, I would be happy to recommend CoEKS for acceptance.” The reviewer **raised the score from 6 to 8 and strongly recommended acceptance.**

---

### Meta-Review · Area_Chair_2Jx4 · 2026-01-05

**Summary:**

Reviewers' concerns are mainly around the following points:

W1. Writing needs improvement

W2. Lacking detailed discussion on the rationale of the MD loss

W3. Missing important baseline

W4. The contribution is incremental

W5. The bump in performance might be simply due to having more parameters

W6. The method needs to activate all the parameters for problems

W7. The results are only incrementally better than previous work, and the proposed work is a small step forward

W8. The idea of adding a new expert for each type of constraint is unlikely to scale in practice

W9. Insufficient experiments to demonstrate advantage

**Reviewer Concerns:**

Most of the above concerns can be addressed by the authors' rebuttal. I do share the concerns raised by reviewer dA3D and qhjy that training an expert for each constraint seems not practical. While authors provided justification and new results in the rebuttal, I feel that this point deserves more in-depth analysis and discussion in the paper. Nevertheless, this paper offers an interesting method with valuable insights to the community. Therefore I recommend acceptance.

**Reviewer Scores:**

Three reviewers engaged in the discussion. Reviewer zFcW raised to 8, while reviewer dA3D and qhjy maintained their score. I feel that reviewer nRLf might increase his/her score. But the standpoint of dA3D is firm, and I think his/her evaluation would still be negative.

---

### Decision · Program_Chairs · 2026-01-26

Accept (Poster)